# Cancer immune control needs senescence induction by interferon-dependent cell cycle regulator pathways in tumours

Ellen Brenner[1,14], Barbara F. Schörg [2,14], Fatima Ahmetlić[3,4,14], Thomas Wieder[1], Franz Joachim Hilke[5], Nadine Simon[1], Christopher Schroeder[5], German Demidov [5], Tanja Riedel[3], Birgit Fehrenbacher[1], Martin Schaller[1], Andrea Forschner[1], Thomas Eigentler[1], Heike Niessner[1], Tobias Sinnberg [1], Katharina S. Böhm[1], Nadine Hömberg[3,4], Heidi Braumüller[1], Daniel Dauch[6,7], Stefan Zwirner[6], Lars Zender[6,7,8], Dominik Sonanini[2,6], Albert Geishauser[3,4], Jürgen Bauer[1], Martin Eichner[9], Katja J. Jarick [10], Andreas Beilhack[10], Saskia Biskup[8,11], Dennis Döcker[1,11], Dirk Schadendorf [7,12], Leticia Quintanilla-Martinez[8,13], Bernd J. Pichler[2,8], Manfred Kneilling[1,2,8], Ralph Mocikat[3,4] & Martin Röcken[1,7,8 ✉]

Immune checkpoint blockade (ICB)-based or natural cancer immune responses largely eliminate tumours. Yet, they require additional mechanisms to arrest those cancer cells that are not rejected. Cytokine-induced senescence (CIS) can stably arrest cancer cells, suggesting that interferon-dependent induction of senescence-inducing cell cycle regulators is needed to control those cancer cells that escape from killing. Here we report in two different cancers sensitive to T cell-mediated rejection, that deletion of the senescence-inducing cell cycle regulators p16$^{Ink4a}$/p19$^{Arf}$ (*Cdkn2a*) or p21$^{Cip1}$ (*Cdkn1a*) in the tumour cells abrogates both the natural and the ICB-induced cancer immune control. Also in humans, melanoma metastases that progressed rapidly during ICB have losses of senescence-inducing genes and amplifications of senescence inhibitors. Metastatic cells also resist CIS. Such genetic and functional alterations are infrequent in metastatic melanomas regressing during ICB. Thus, activation of tumour-intrinsic, senescence-inducing cell cycle regulators is required to stably arrest cancer cells that escape from eradication.

[1] Department of Dermatology, University of Tübingen, 72076 Tübingen, Germany. [2] Department of Preclinical Imaging and Radiopharmacy, Laboratory for Preclinical Imaging and Imaging Technology of the Werner Siemens-Foundation, University of Tübingen, 72076 Tübingen, Germany. [3] Institut für Molekulare Immunologie, Helmholtz-Zentrum München, 81377 Munich, Germany. [4] Eigenständige Forschungseinheit Translationale Molekulare Immunologie, Helmholtz Zentrum München, 81377 Munich, Germany. [5] Institute of Medical Genetics and Applied Genomics, University of Tübingen, 72076 Tübingen, Germany. [6] Department of Medical Oncology and Pneumology, University Hospital Tübingen, 72076 Tübingen, Germany. [7] German Cancer Research Consortium (DKTK), German Cancer Research Center (DKFZ), 69120 Heidelberg, Germany. [8] Cluster of Excellence iFIT (EXC 2180) "Image Guided and Functionally Instructed Tumor Therapies", 72076 Tübingen, Germany. [9] Institute of Clinical Epidemiology and Applied Biometry, University of Tübingen, 72076 Tübingen, Germany. [10] Department of Medicine II, Würzburg University, 97078 Würzburg, Germany. [11] Center for Genomics and Transcriptomics (CeGaT) GmbH and Practice for Human Genetics, 72076 Tübingen, Germany. [12] Department of Dermatology, University Hospital, West German Cancer Centre, University Duisburg-Essen, 45147 Essen, Germany. [13] Institute of Pathology, University of Tübingen, 72076 Tübingen, Germany. [14]These authors contributed equally: Ellen Brenner, Barbara F. Schörg, Fatima Ahmetlić. ✉email: mrocken@med.uni-tuebingen.de

mmune therapy with ICB and natural immune responses reveal that tumour-infiltrating cytotoxic T cells, natural killer cells and $T_H1$ cells can be activated to cause cancer regression and clearance of tumour cells through cytolysis, apoptosis and by activated macrophages in mice and in humans[1–10]. Nonetheless, cancers are often not completely eliminated, and aggregate or single cancer cells may persist in a controlled state, called tumour dormancy[2,11–17]. The failure of inducing tumour dormancy is a major cause of treatment resistance that may result from inappropriate immune activation, low tumour mutation burden, resistance to lysis or apoptosis, and other mechanisms[4,8,16,18–24]. More recent data associate melanoma progression and treatment resistance of cancers with functional losses of the IFN-JAK1-STAT1 signalling pathway[11,23,25–30]. IFN-γ has multiple effects on cancers. Previously, we have shown that the combined action of IFN-γ and TNF is capable of inducing a stable, senescence-like growth arrest in cancer cells that is called cytokine-induced senescence (CIS)[31]. As the IFN-JAK1-STAT1 signalling cascade activates two key inducers of senescence[14,31,32], p16[Ink4a] and p21[Cip1], here we analyse whether cancer immune control requires the IFN-γ-dependent induction of the tumour-intrinsic p16[Ink4a]-CDK4/6-Rb1 and MDM-p53-p21[Cip1] cell cycle regulation pathways to arrest those cancer cells that escape from cytotoxicity.

## Results

**Cancer immune control needs *Stat1*-dependent *Cdkn2a* activity.** In vitro activation of p16[Ink4a] and p21[Cip1] requires IFN-γ signalling in the tumour cells[31,32]. We therefore asked whether in vivo activation of p16[Ink4a], senescence induction and cancer immune control also require a functioning IFN-γ signalling cascade in the cancer cells. To investigate the role of the IFN-*Stat1*-dependent activation of *Cdkn2a*, what encodes alternatively spliced variants, including the structurally related CDK4 kinase inhibitor isoforms p16[Ink4a] and p19[Arf], we analysed the immune control of transplanted cancer cells isolated from RIP-Tag2 mice, where expression of the T antigen under the control of the rat insulin promoter (RIP) leads to pancreatic islet cancers (RT2-cancers)[25]. For this, we implanted either *Stat1*-positive or *Stat1*-negative RT2-cancer cells into syngeneic mice. Most *Stat1*-positive and *Stat1*-negative RT2-cancers (>80%) were rejected. As *Stat1*-positive and *Stat1*-negative RT2-cancers were similarly susceptible to lysis by cytotoxic CD8[+] T cells (Supplementary Fig. 1a), we depleted CD8 T cells with mAb before transplanting the cancer cells. Following CD8-depletion the tumours grew in vivo, and treatment was started when tumours reached >3 mm (Supplementary Fig. 1b). In sham-treated mice cancers reached a critical size within three weeks, and the tumours were analysed (Fig. 1a). The rapidly growing cancers from sham-treated mice had a proliferative Ki67[+]p16[Ink4a−] phenotype (Fig. 1b, c and Fig. 2a) that was negative for phosphorylated heterochromatin protein 1γ (S93) (pHP1γ) in senescence-associated heterochromatin foci (SAHF), H3K9me3 and senescence-associated β-galactosidase (SA-β-gal) (Fig. 2b–d, Supplementary Fig. 2a, b). To determine whether the immune system can control these cancers, we treated mice with a combination of anti-PD-L1 and anti-LAG-3 mAb[33]. In preclinical studies and human trials, dual blockade of LAG-3 and the PD-1/PD-L1 interaction generates a more efficient anti-cancer immunity in mice and in humans than blocking either molecule alone[34]. The *Stat1*-positive RT2-cancers became growth arrested or regressed (Fig. 1a). The residual cancer cells displayed a senescent p16[Ink4a+], Ki67[−] phenotype with an induction of p21[Cip1] in single cells (Fig. 1b, c and Fig. 2a). RT2-cancers were also positive for pHP1γ, H3K9me3 and SA-β-gal (Fig. 2b–d), and showed no ICB-induced double strand breaks as determined by γH2AX and DNA-PK staining (Supplementary Fig. 3a, b). In contrast, ICB did

not arrest *Stat1*-negative RT2-cancers that grew rapidly even when treated with ICB (Fig. 1a). *Stat1*-negative tumour cells were p16[Ink4a]-negative and expressed Ki67 (Fig. 1b, c) but neither pHP1γ, H3K9me3, nor SA-β-gal (Fig. 2b–d). In vitro data confirmed that RT2.*Stat1*[−/−]-cancers were selectively resistant to CIS but susceptible to apoptosis (Fig. 3a).

Immune activation with anti-PD-L1 and anti-LAG-3 requires the presence of PD-L1. *Stat1*-positive and *Stat1*-negative RT2-cancers expressed PD-L1 in the tumour tissue and on more than 10% of the isolated cancer cells (Supplementary Fig. 4a, b), showing that *Stat1*-negative RT2-cancers expressed the target of the anti-PD-L1 mAb. IFN-γ strongly increased PD-L1 and β2-microglobulin expression on RT2-cancer cells but not on RT2.*Stat1*[−/−]-cancer cells; β2-microglobulin[+] cells were found in sections of RT2.*Stat1*[−/−]-cancers, showing that IFN-γ-responsive host immune cells infiltrated the tumour microenvironment during ICB (Supplementary Fig. 4a–d).

*Stat1* was not required for tumour elimination by CD8[+] cells but for the induction of p16[Ink4a] by IFN-γ-producing immune cells and for an efficient cancer immune control. As p16[Ink4a] is needed for CIS in RT2-cancer cells[31], we asked whether cancer immune control needs the senescence-inducing cell cycle regulator p16[Ink4a] in the tumour cells. To address this question, we generated *Cdkn2a*-deficient RT2-cancer cell lines through in vitro and in vivo selection. Comparative genomic hybridisation (CGH) showed the loss of *Cdkn2a* on chromosome 4 (qC4.A) as the only genetic aberration common to all six cell lines (Supplementary Fig. 5a). PCR analysis confirmed the loss of *Cdkn2a* (Supplementary Fig. 5b); PCR also revealed that *Cdkn2a*[−/−]-RT2-cancer cells expressed Tag. RT2.*Cdkn2a*[−/−]-cancer cells also expressed PD-L1 and responded to IFN-γ (Supplementary Fig. 4a, b, d). While the parental RT2-cancer cells were susceptible to CIS and to apoptosis, the *Cdkn2a*-loss mutant cell lines were resistant to CIS but susceptible to apoptosis in vitro (Fig. 3b). To test the role of *Cdkn2a* for tumour immune control in vivo, we injected the tumour cell lines into syngeneic mice and again started treatment with ICB once tumours reached a diameter >3 mm. As we transferred polyclonal *Cdkn2a*-loss mutant cancer cell lines, the tumours grew with slightly different dynamics (Fig. 1a). ICB did not attenuate the growth of *Cdkn2a*-deficient RT2-cancers. Immune histology showed that ICB also failed to induce senescence in *Cdkn2a*-deficient RT2-cancers as they displayed a Ki67[+]p16[Ink4a−], pHP1γ[−], H3K9me3[−], SA-β-gal[−] phenotype (Fig. 1b, c and Fig. 2b–d).

To test whether the resistance to ICB specifically resulted from the p16[Ink4a]/p19[Arf] loss, and to exclude other potential confounders, we generated RT2-cancer cells from another diseased mouse and deleted p16[Ink4a]/p19[Arf] (g*Cdkn2a*) using CRISPR-Cas9[31]. In vitro, RT2-cancer cells transfected with either a control sgGFP or the g*Cdkn2a* construct grew with similar dynamics. Both cell types were sensitive to apoptosis. The CRISPR-Cas9 control cells expressed SA-β-gal when exposed to IFN-γ/TNF and did not restart their exponential growth following IFN-γ/TNF withdrawal, showing that they were susceptible to CIS (Fig. 3c). In contrast, IFN-γ/TNF did not induce SA-β-gal in the RT2.CRISPR-*Cdkn2a*-cancer cells, and the cells restarted exponentially growing after IFN-γ/TNF withdrawal, showing that RT2.CRISPR-*Cdkn2a*-cancer cells were resistant to CIS (Fig. 3c). To specifically analyse *Cdkn2a* in vivo, we injected the cells into the CD8-depleted mice. All CRISPR-Cas9 control cell lines were rejected, revealing that the cells were highly immunogenic. In contrast, 80% of the RT2.CRISPR-*Cdkn2a*-cancer cells grew in syngeneic mice (Supplementary Fig. 5c), demonstrating that natural immune responses required the senescence-inducing *Cdkn2a* gene to control these highly immunogenic cancer cells (Fig. 1a, upper panel). Even enhancement of the immune response with ICB did not arrest

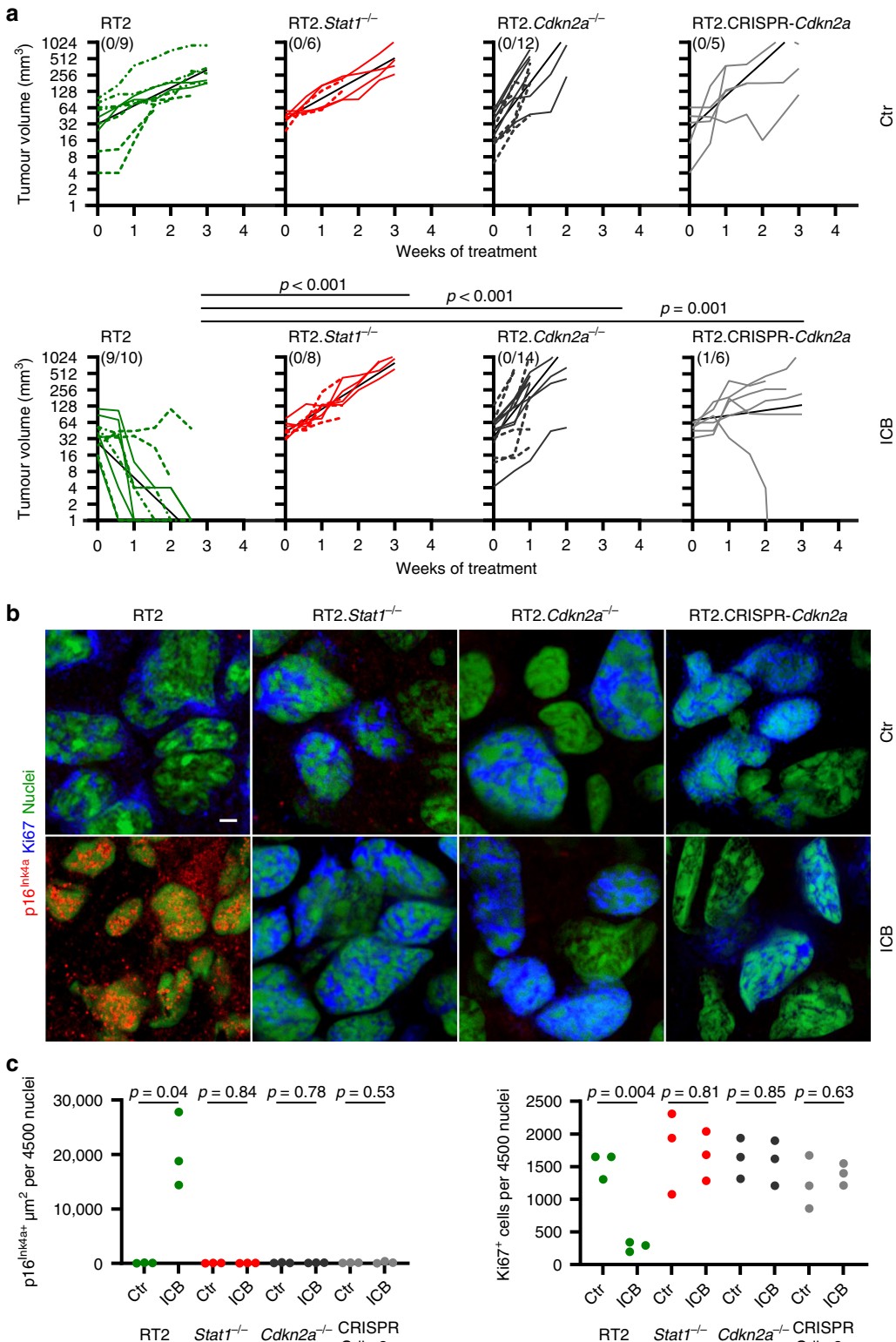

their growth in vivo (Fig. 1a, lower panel). RT2.CRISPR-*Cdkn2a*-cancer cells expressed PD-L1, β2-microglobulin (Supplementary Fig. 4a–d) and Tag. Immune histology revealed that the rapidly growing RT2.CRISPR-*Cdkn2a*-cancer cells had a Ki67+, p16^Ink4a−, pHP1γ−, SA-β-gal− proliferative phenotype, but were positive for H3K9me3 (Fig. 1b, c and Fig. 2b–d). Even though H3K9me3 is a marker of senescence, others have shown that CRISPR-Cas9 editing itself may also induce a non-specific H3K9

methylation that persists[35]. Importantly, ICB treatment increased neither H3K9me3, nor DNA-PK, nor γH2AX in RT2.CRISPR-*Cdkn2a*-cancer cells (Fig. 2c, Supplementary Fig. 3a, b).

CGH of 10 different cell lines revealed that the loss of the p16^Ink4a/p19^Arf-locus was the only common signature where the RT2.*Cdkn2a*^−/−- or the RT2.CRISPR-*Cdkn2a*-cancer cells differed from the parental or the CRISPR-Cas9 control cell lines (Supplementary Fig. 5a, b). We analysed a total of seven different

**Fig. 1 Stat1- and Cdkn2a-dependent immune control of transplanted RT2-cancers and induction of Ki67⁻p16^Ink4a+ senescent cancer cells. a** Individual follow-up of tumour volumes. CD8-depleted mice were subcutaneously (s.c.) engrafted with 1 × 10⁶ RT2-, RT2.$Stat1^{-/-}$-, RT2.$Cdkn2a^{-/-}$- or RT2.CRISPR-$Cdkn2a$-cancer cells. Treatment with isotype control mAbs (Ctr) or combined immune checkpoint inhibitors (ICB; anti-PD-L1 and anti-LAG-3) once per week was started when tumours were >3 mm in diameter. Cancer size was measured 2 times per week. Number of mice with regressing tumours and the total number of mice is given in parenthesis; RT2 Ctr $N = 9$, ICB $N = 10$, RT2.$Stat1^{-/-}$ Ctr $N = 6$, ICB $N = 8$, RT2.$Cdkn2a^{-/-}$ Ctr $N = 12$, ICB $N = 14$, RT2. CRISPR-$Cdkn2a$ Ctr $N = 5$, ICB $N = 10$. Each cell line was given a different lining. Black lines summarise the results for different treatment groups (as obtained from ANCOVA). $p$-values examine the question whether the treatment effect was different between two genotypes. Mice were killed either when tumours reached the critical diameter of 15–20 mm or ulcerated, or when mice developed signs of wasting. **b** Representative triple-staining for the senescence marker p16^Ink4a (red) and the proliferation marker Ki67 (blue) and for nuclei (green) of the s.c. tumour of individual mice treated as described in (**a**). Scale bar 2 μm. **c** Individual data points showing quantification of p16^Ink4a+ (left) or Ki67⁺ (right) cells. Each data point represents the total of three tumour slides measurements, tumours of three individual mice (described in **a**) were analysed. In **a** significance was tested by using unequal variances $t$-test, $p$-values examines the treatment effect, comparing the ICB-treated RT2-cancers with each ICB-treated knock-out group.

$Cdkn2a$-deficient RT2-cancer cell lines in vitro. All were resistant to CIS. Next, we analysed three of them in vivo, all were resistant to ICB. Thus, cytotoxic CD8⁺ T cells can directly reject $Cdkn2a$-deficient RT2-cancers, but senescence-induction through IFN-γ-mediated activation of the cell cycle regulators p16^Ink4a/p19^Arf was strictly needed to control those cancer cells that escape from cytotoxicity.

Cancers are frequently rejected by CD8 T cells. In addition, immune clearance of senescent cancer is mediated by IFN-γ-producing $T_H1$ cells and by IFN-γ-activated type I macrophages[25,31,36–39]. Indeed, the tumours controlled by ICB showed the established features of the $T_H1$-mediated clearance mechanism, as β2-microglobulin was strongly expressed (Supplementary Fig. 4c, d) and as the tumours were infiltrated by CD3⁺CD8⁻ T cells and F4/80⁺ and MHC class II⁺ activated macrophages (Supplementary Fig. 6a–c). We detected no CD8⁺ T cells and no CD49b⁺ NK cells (Supplementary Fig. 6d), what may be due to the fact that NK cells are primarily found in the spleen[40,41]. As ICB induced a similar immune infiltrate in RT2-cancers and in $Stat1$- and $Cdkn2a$-deficient RT2-cancers (Supplementary Fig. 6a–c), the data strongly suggest that senescence induction in the cancer cells was a prerequisite for tumour cell clearance by $T_H1$ cells and IFN-γ-activated type I macrophages[25,31,36–39].

**Cancer immune therapy and senescence induction require Stat1.** To determine whether IFN-γ signalling through $Stat1$ is also needed for the induction of p16^Ink4a, p21^Cip1, senescence and the control of endogenous cancers that are destroyed by strong T cell responses, we treated RT2 mice by the combination of anti-PD-L1 and anti-LAG-3 mAbs and adoptive T cell transfer (AT), with $T_H1$ cells specific for a tumour associated antigen (TAA) in the SV40-Tag protein[31] (Supplementary Fig. 7a). Combining anti-PD-L1/anti-LAG-3 mAbs with AT further enhances the therapeutic effect of ICB and largely eradicates all cancer cells[33]. We started the treatment once RT2 mice had a major cancer load, as documented by magnetic resonance imaging (Supplementary Fig. 7b). This immune therapy strongly decreased the islet size within 4 weeks, and functionally restored the blood glucose control (Fig. 4a–c). It largely destroyed the RT2-cancers but failed to eradicate all cancer cells (Fig. 4b, Supplementary Fig. 8a). Immune histology of residual RT2-cancers showed CD3⁺ cells, MHC class II⁺ and F4/80⁺ cells following ICB/AT treatment but only very few or no Foxp3⁺ regulatory T cells, CD8⁺ or CD49b⁺ cells (Fig. 5a, Supplementary Fig. 8a–c). The RT2-cancers expressed normal levels of Tag mRNA and protein (Fig. 5b, c) and expressed PD-L1 and β2-microglobulin protein (Supplementary Fig. 8d). The RT2-cancer cells showed a senescent phenotype as they expressed p16^Ink4a, p21^Cip1, H3K9me3, pHP1γ, and SA-β-gal but were Ki67⁻ (Fig. 4d, e and Fig. 6a–d). Staining for γH2AX and DNA-PK, markers that mainly indicate

double strand breaks, remained largely negative, confirming that CIS caused only minor DNA damage (Supplementary Fig. 9). Electron microscopy confirmed the accumulation of SA-β-gal in the cytoplasm of senescent tumour cells (Supplementary Fig. 10a–c). These residual RT2-cancer cells were also functionally senescent. When isolated and cultured for ≥5 passages in vitro they preserved their growth-arrested, SA-β-gal⁺ phenotype (Fig. 7a–c). RT2-cancer cells with such a senescent phenotype remain growth arrested for at least 3 months, even when transplanted into immune compromised NOD–SCIDIl2rγ^{-/-} mice, demonstrating the absence of cancer initiating cells[31].

Sham-treated RT2-mice, sham-treated $Stat1^{-/-}$ RT2 mice or $Stat1^{-/-}$ RT2 mice treated with ICB/AT died within 4 weeks, showed large tumours and failed to control their blood glucose levels (Fig. 4a–c). The tumours of these three groups of mice were strongly enriched in Ki67⁺ RT2-cancer cells that were negative for p16^Ink4a, p21^Cip1, H3K9me3, pHP1γ, and SA-β-gal (Fig. 4d, e and Fig. 6a–d). The isolated tumour cells also proliferated strongly when cultured in vitro and did not develop a senescent phenotype (Fig. 7a–c). While treatment with ICB/AT did not induce a senescent phenotype and did not attenuate the tumour growth in the RT2.$Stat1^{-/-}$ mice, the inflammatory infiltrate was similar to that of cancers in RT2.$Stat1^{+/+}$ mice (Supplementary Fig. 8a–c). Also three-dimensional imaging and FACS analyses revealed that the adoptively transferred T cells and dendritic cells infiltrated RT2-cancers of $Stat1^{-/-}$ or $Stat1^{+/+}$ mice with similar dynamics (Supplementary Fig. 11a–e); immune histology confirmed these findings. Islet tumours of RT2 and RT2.$Stat1^{-/-}$ mice had both CD3⁺ T-cells and MHC class II⁺, F4/80⁺ macrophages (Fig. 5a and Supplementary Fig. 8a, b). In islet tumours of RT2.$Stat1^{-/-}$ mice the T cell infiltrate was variable but there was no detectable size difference between tumour areas that were strongly or poorly infiltrated. At this advanced cancer stage, the combined ICB/AT therapy was superior to either ICB or AT monotherapy that provided intermediate results (Fig. 4a–e and Fig. 6a–d).

Importantly, the tumour cells surviving the combined ICB/AT immune therapy did not result from classical immune evasion mechanisms. In vitro, all RT2-cancer cells, including the RT2.$Stat1^{-/-}$-cancer cells were fully susceptible to T cell-mediated killing (Supplementary Fig. 1a) or apoptosis (Fig. 3a). RT2.$Stat1^{+/+}$-cancer cells and RT2.$Stat1^{-/-}$-cancer cells expressed normal levels of Tag (Fig. 5b, c), the antigen targeted by the immune therapy. RT2.$Stat1^{+/+}$-cancers and RT2.$Stat1^{-/-}$-cancers had normal baseline levels of PD-L1 and β2-microglobulin, and in RT2.$Stat1^{-/-}$-cancers β2-microglobulin was in the tumour microenvironment (Supplementary Fig. 4a–d and Supplementary Fig. 8d). Foxp3 regulatory T cells were not increased inside the tumours (Supplementary Fig. 8b). In consequence, the RT2.$Stat1^{-/-}$-cancers differed from RT2.$Stat1^{+/+}$-cancers selectively by their resistance to CIS (Fig. 3a).

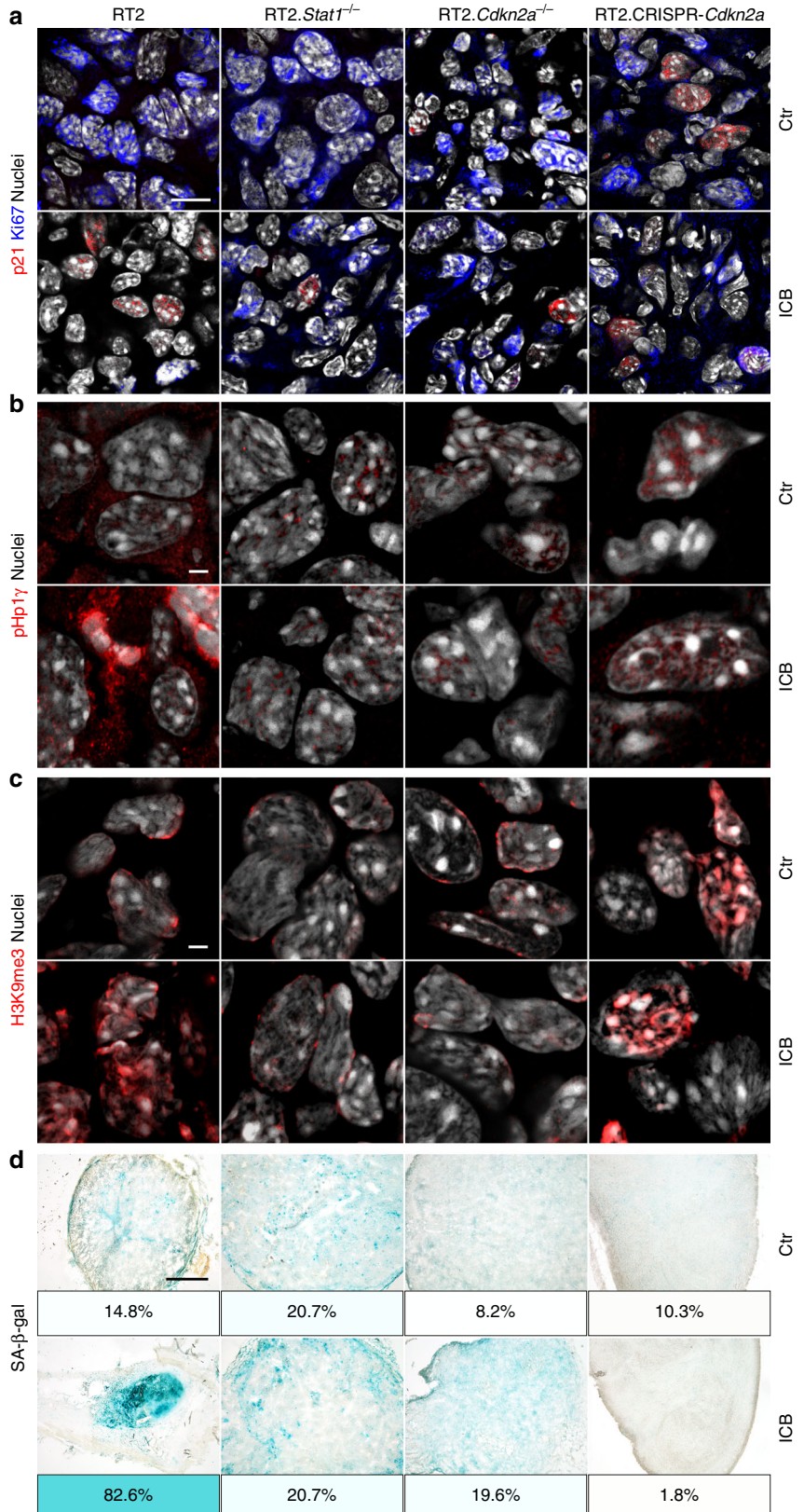

**Fig. 2 *Stat1*- and *Cdkn2a*-dependent induction of the senescence markers pHP1γ, H3K9me3, and of SA-β-gal in transplanted RT2-cancers.**
**a–d** Representative microscopic images of s.c. RT2-cancers from either RT2-, RT2.*Stat1*−/−-, RT2.*Cdkn2a*−/−- or from RT2.CRISPR-*Cdkn2a*-cancer cells. Mice were treated with isotype control mAbs (Ctr) or with immune checkpoint blockade (ICB, anti-PD-L1 and anti-LAG-3). Staining for the senescence marker p21[Cip1] (red), the proliferation marker Ki67 (blue) and for nuclei (white) (**a**), pHP1γ (red), nuclei (white) (**b**), H3K9me3 (red), nuclei (white) (**c**). SA-β-gal activity at pH5.5 and percentage of SA-β-gal positive tumour cells in each tumour (**d**). The colour evaluation and calculation of the SA-β-gal+ cells are described in Supplementary Fig. 2 and Methods (for **d**). Scale bars 10 μm (**a**), 2 μm (**b**, **c**), 1000 μm (**d**). Histology was performed in one to three representative tumours from Fig. 1c.

Even though the ICB/AT combination therapy largely destroyed the tumours, the therapy failed to eradicate all cancer cells. Together the data prove that *Stat1* is needed to activate p16[Ink4a] in vitro and in vivo and that *Stat1*-mediated activation of *Cdkn2a* is needed to induce senescence in cancer cells. In consequence, cancer immune control required *Stat1*-mediated activation of the cell cycle regulators like p16[Ink4a] to stably arrest the growth of those cancer cells that are not eliminated by cytotoxicity.

**Cancer immune control needs IFN-γ-dependent p21[Cip1] induction**. Following ICB *Stat1*-deficient cancers showed neither increased p16[Ink4a] nor p21[Cip1]. Thus, besides p16[Ink4a], cancer immune control may also require p21[Cip1] activation. SV40-Tag expression impairs p53 activation[42]. As p53 regulates mRNA and protein production of p21[Cip1], RT2-cancers are inappropriate to carefully investigate the role of p21[Cip1] for senescence induction in response to ICB. To test the need of senescence-inducing p21[Cip1] for cancer immune control and to determine whether senescence induction is also needed for tumours other than RT2-cancers, we studied the role of p21[Cip1] in the immune control of lymphomas. For this, we used λ-*MYC* mice, where a human *MYC* oncogene under the control of the immunoglobulin λ enhancer induces the development of endogenous B-cell lymphomas[43]. This also allows directly investigating the role of *p21*[Cip1] in the immune therapy of a naturally developing malignancy. Untreated mice died within <150 days (Fig. 8a) from Ki67[+]p16[Ink4a−], CD20[low] B-cell lymphomas that destroyed lymph nodes and spleen (Fig. 8b, c and Supplementary Fig. 12a). The Ki67[+] B cells were also negative for p21[Cip1], pHP1γ, H3K9me3 or SA-β-gal, showing that the tumours had a high proliferative capacity and were not senescent (Fig. 8b, c and Fig. 9a–c). Combined ICB with anti-CTLA-4 and anti-PD-1 mAb protected 18% (Fig. 8a) to 30% (Fig. 8d) of λ-*MYC* mice from lymphomas for at least 200 days. In a long-term experiment, 18% of the λ-*MYC* mice with a combined ICB therapy were still healthy at >250 days (Fig. 8a), a lifetime that has never been achieved by any other therapy in this mouse model. Treatment with either mAb alone did not rescue mice from lymphomas. Lymph nodes from healthy ICB-treated λ-*MYC* mice showed a normal architecture with normal B and T cell areas, expression of PD-L1 and β2-microglobulin, no T cell infiltration, normal CD20[+]λ-MYC[−] B cells, few CD161[+] cells, no signs of DNA double strand breaks and no destruction (Supplementary Fig. 12a–e). Even though we found no signs of immune destruction in the lymph nodes, depletion of pan-T cells with a mAb abrogated the ICB-mediated protection of λ-*MYC* mice, as all mice died from lymphomas by day 200 (Fig. 8d). Splenic dendritic cells of λ-*MYC* mice express the CTLA-4-targets CD80 and CD86 that are enhanced by IFN-γ[41]. Lymphoma-specific, regulatory T cells are also increased in the spleen. Yet, depletion of regulatory T cells delays lymphoma development and death only for a short time[40].

Nodal B cells from ICB-treated, healthy λ-*MYC* mice were Ki67[−] but strongly expressed nuclear p16[Ink4a] and p21[Cip1] (Fig. 8b, c), pHP1γ, H3K9me3 and SA-β-gal (Fig. 9a–c), displaying a senescent phenotype[24,31]. This suggests that ICB-mediated protection from B-cell lymphomas included mechanisms other than cancer cell killing. To test whether ICB-mediated lymphoma prevention required the senescence-inducing p21[Cip1], we generated syngeneic λ-*MYC*.p21[Cip1−/−] mice. While ICB protected 18–30% of λ-*MYC* mice from lymphoma and death (Fig. 8a, d), all ICB-treated λ-*MYC*.p21[Cip1−/−] mice died from lymphomas between day 100 and 210 (Fig. 8e). Histology revealed lymph node destruction by Ki67[+], CD20[low] B cells that

were devoid of the senescence markers p16[Ink4a], p21[Cip1], pHP1γ, H3K9me3, or SA-β-gal (Fig. 8b, c, Fig. 9a–c and Supplementary Fig. 12a).

ICB-mediated p21[Cip1] induction and lymphoma prevention required IFN-γ, as mice receiving ICB in the presence of anti-IFN-γ mAb died from CD20[low] B-cell lymphomas as fast as untreated mice (Fig. 8a). B cells were Ki67[+] and negative for p21[Cip1], p16[Ink4a], pHP1γ, H3K9me3 and SA-β-gal (Fig. 8b, c and Fig. 9a–c), showing that IFN-γ was needed to activate the cell cycle regulators p16[Ink4a] and p21[Cip1] that, in turn, were required for senescence-induction in vivo. This proves that the senescence-inducing cell cycle regulator p21[Cip1] was strictly required to prevent the transition of pre-malignant B cells into B-cell lymphomas in ICB-treated λ-*MYC* mice, and that ICB-mediated p21[Cip1] activation was IFN-γ-dependent.

**Impaired senescence pathways in melanomas resistant to ICB**. ICB is an approved standard of care therapy for metastatic melanoma and some other cancers, and efficient in about 40% of patients with metastatic melanoma. Another 40% are non-responder patients that mostly progress rapidly despite ICB therapy[6,8,30,44]. Based on the experimental data, we asked whether cell cycle regulator genes that control senescence induction were also needed for cancer immune control in humans. To address this, we compared the genetic alterations by targeted panel sequencing of 30 melanoma biopsies of consecutive non-responder patients, where metastases progressed within <3 months of ICB with the genetic alterations in melanoma biopsies of 12 responder patients, where metastases regressed during ICB ≥1 year. For each patient we identified the tumour-specific alterations by paired sequencing of tumour and normal tissue. In agreement with published data, the biopsy material of responder patients had a significantly higher tumour mutational burden than that of non-responder patients[8,10,45] (Fig. 10a). We used a panel that includes a total of 678 established cancer-associated genes[46] (Supplementary Data Tables 1–3). To determine whether CIS was needed to protect from tumour progression, we explicitly focussed only on 19 genes of the panel that encode for molecules that are also involved in the IFN-γ-mediated activation of the senescence signalling pathways that we identified in our murine experiments. This panel approach may have missed very rare fusion/translocation alterations of the genes investigated, but it would be difficult to identify such rare fusion/translocation alterations also with an exome-based approach, as the panel allowed the deep sequencing necessary to find the DNA alterations in less than 10% of all cells of the melanoma metastases.

We focused on somatic alterations, namely copy number variants (CNVs) and single nucleotide variants (SNVs) in key cell cycle control genes (*CCND1/2/3*; *CDKN2A/B/C*, *CDK4/6*, *CCNE1*, *CDKN1A/B*, *RB1*, *TP53*, *MDM2/4*), as well as *JAK1,2,3* and *MYC*. Both groups had a similar distribution of genetic aberrations in *JAK1,2,3*, *MYC* and the cell cycle control genes when all somatic alterations (SNVs and CNVs) were included (Fig. 10b). In contrast, comparing especially the number of strong amplification (≥ 3fold amplification) and of fully inactivating mutations (homozygous deletions and loss of heterozygosity (LOH)), we observed significant differences between the two groups. Melanomas of non-responder patients had significantly more fully inactivating mutations of senescence-inducing cell cycle genes (*CDKN2A/B/C*; *CDKN1A/B*; *RB1*; *TP53*; *JAK1/2/3*) or ≥ 3fold amplifications of genes promoting cell cycle progression (*CCND1/2/3*; *CDK4/6*; *CCNE1*; *MDM2/4*; *MYC*) than the biopsy material of melanomas from responder patients (Fig. 10c, Supplementary Fig. 13 and 14a, b).

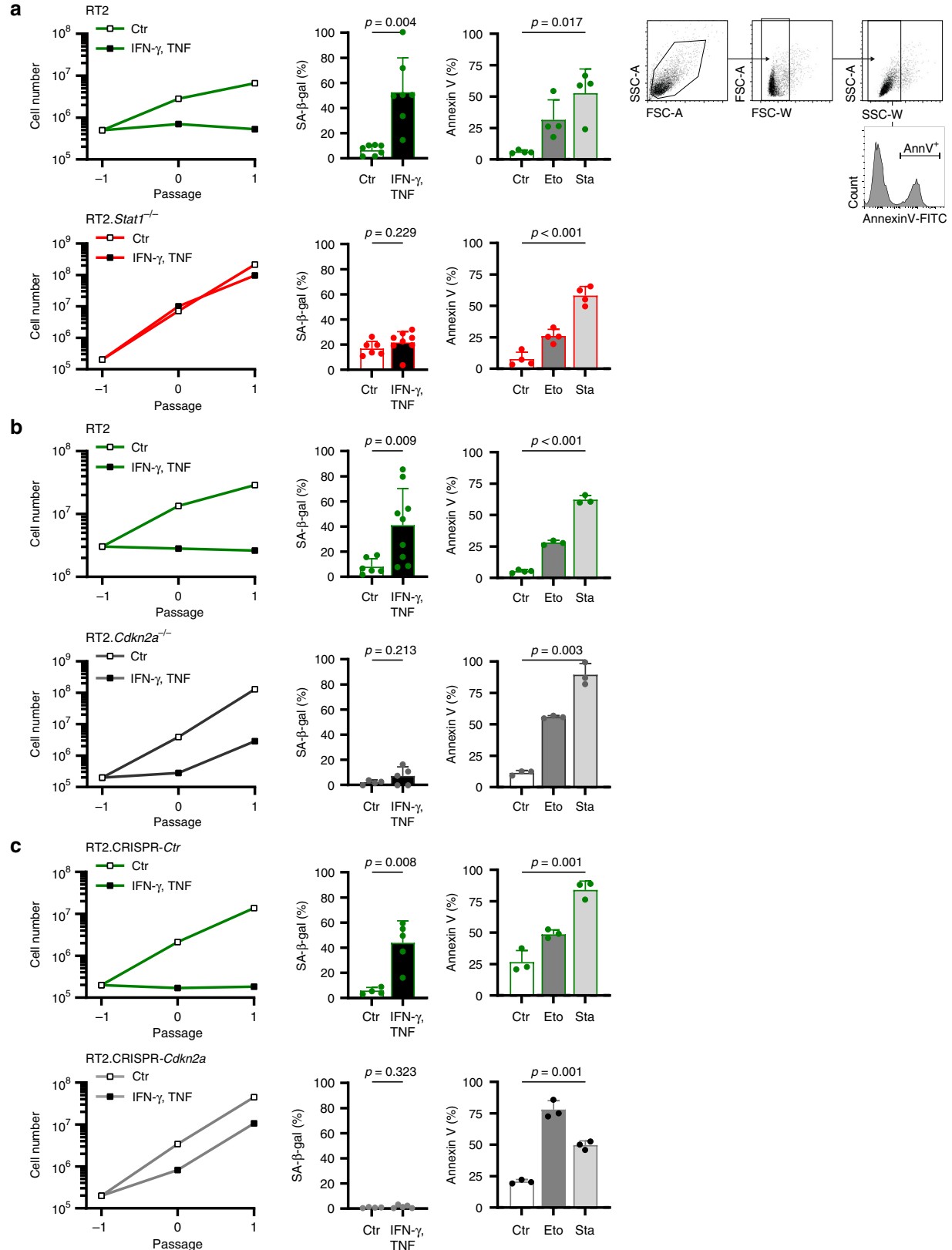

The genetic data were supported by functional analyses in vitro. Melanoma lines that we developed from biopsies of non-responder patients grew with similar dynamics as those derived from biopsies of responder patients and were susceptible to apoptosis (Fig. 10d). Yet, these cell lines were resistant to CIS. In contrast, melanoma lines that we derived from biopsy material of responder patients were susceptible to both, CIS and drug-induced apoptosis. TNF and IFN-γ are established inducers of annexin V and apoptosis in multiple cancer cells including melanoma cell lines[47]. Importantly, they induced senescence only in selected lines, such as those that could be derived from biopsy material of responder patients (Fig. 10e).

**Fig. 3 Stat1- and Cdkn2a-dependent induction of CIS in RT2-cancer cells, but Stat1- and Cdkn2a-independent induction of apoptosis. a–c** Assays were performed with RT2-cancer cells (**a**, **b**), RT2.$Stat1^{−/−}$-cancer cells (**a**), $Cdkn2a$-deficient RT2-cancer cells (**b**), or RT2.CRISPR-Ctr- control, or RT2.CRISPR-$Cdkn2a$-cancer cells (**c**). For the senescence growth assay, cells were cultured either with medium (Ctr) or with medium containing 100 ng ml$^{−1}$ IFN-γ and 10 ng ml$^{−1}$ TNF for 96 h, washed and then cultured with medium for another 3–4 days. One representative out of 3 independent experiments was given. SA-β-gal activity was determined after 96 h of culture with medium (Ctr) or with medium containing 100 ng ml$^{−1}$ IFN-γ and 10 ng ml$^{−1}$ TNF, data show the mean with SD, RT2 Ctr $n = 7$, CIS $n = 7$, RT2.$Stat1^{−/−}$ Ctr $n = 6$, CIS $n = 8$ (**a**), RT2 Ctr $n = 6$, CIS $n = 9$, RT2.$Cdkn2a^{−/−}$ Ctr $n = 4$, CIS $n = 5$ (**b**), RT2.CRISPR-Ctr Ctr $n = 4$, CIS $n = 5$, RT2.CRISPR-$Cdkn2a$ Ctr $n = 4$, CIS $n = 5$ (**c**). For apoptosis induction, cells were exposed to either medium (Ctr) or etoposide (Eto, 100 μM) or staurosporine (Sta, 0.5 μM) for 24 h and then stained for annexin V. Positive cells were detected by flow cytometry data show the mean with SD (**a** (including gating strategy), **b**, **c**), RT2 $n = 4$, RT2.$Stat1^{−/−}$ $n = 4$ (**a**), RT2 Ctr $n = 4$, Eto, Stau $n = 3$, RT2.$Cdkn2a^{−/−}$ $n = 3$ (**b**), RT2.CRISPR-Ctr and RT2.CRISPR-$Cdkn2a$ $n = 3$ (**c**). Significance tested by using unequal variances $t$-test.

## Discussion

Our results show that cancer control strictly requires the activation of tumour-intrinsic, senescence-inducing cell cycle regulators by the immune system to stably arrest those cancer cells that escape from eradication, in mice and in humans. Even though elimination of cancer cells is a primary therapeutic goal, cancer cell reduction by cytotoxicity is frequently incomplete and insufficient for permanent cancer control. About 90% of all cancer-related deaths result from metastases arising from reawakened, dormant cancer cells that survive chemo- or immune-therapies[2,48–51], often months to decades after treatment of the primary tumour[15]. The data here unravel that IFN-γ/STAT1-dependent activation of the senescence-inducing cell cycle regulators p16$^{Ink4a}$/p19$^{Arf}$ and p21$^{Cip1}$ is needed to keep those cells senescent that escape from ICB-induced cell death, and that metastases resistant to immune therapies grow, when senescence-inducing signalling pathways become interrupted. This concept is supported by recent clinical data. Thus, melanoma metastases regress and may even clinically disappear upon ICB therapy, but can restart growing when mutations abrogate the IFN-γ-signalling pathway[26]. Various observations suggest that immune control of cancers requires, in addition to cancer cell killing, IFN-dependent activation of cancer-intrinsic senescence-inducing cell cycle regulators[13,14,31,52,53], to stably arrest those cancer cells that escape from cytotoxicity.

Even though senescence occurs following cancer therapy[54], the role of senescence in the development of metastases and the response to chemo- and immune-therapy has not been resolved[52,53,55]. In mice, loss of $p16^{Ink4a}$ promotes the development of melanoma metastases in mice[56] and loss of the $Suv39h1$ or $p53$ genes in senescent cells may transform these cells in highly aggressive, cancer initiating cells[24]. In contrast, in humans the bi-allelic loss of $CDKN2A$ has only been shown to promote melanoma invasion. Until now, it was not possible to identify specific gene mutations, like the loss of $CDKN2A$ that promote the development of melanoma metastases in humans. This has only been demonstrated in mice[56–59]. The missing identification of metastases-promoting genes may be due to the fact that the IFN-γ-mediated activation of the MDM2-p53-p21$^{Cip1}$ senescence pathway can in part compensate the loss of $CDKN2A$[55]. In line with this, we found gene amplifications of $MDM2$ and $MDM4$ in 26% and of $CDK4/6$ in 13% of the melanoma biopsies of patients that did not respond to ICB. These amplifications were absent in melanoma biopsies of patients responding to ICB. More than half of the metastases of non-responder patients had at least one defect in the IFN-dependent senescence-signalling pathway. Both, p16$^{Ink4a}$ and p21$^{Cip1}$ induce senescence by inhibiting CDK4/6, and most genes associated with the resistance of melanoma metastases to ICB were up-stream of CDK4/6. Therefore, combining ICB with CDK4/6 inhibitors is a promising strategy to turn metastases with these types of defects in the senescence pathway from metastases not responding to ICB into metastases responding to ICB.

While senescence is established as a barrier against cancer development[60], and while it is needed to prevent the rapid regrowth of cancers that escape from therapy- or immune-mediated lysis or apoptosis, the exact role of senescence in halting the progression of established cancers is still under debate. Some data suggest that the senescence-associated secretory phenotype (SASP) may even exert harmful effects[54]. Moreover, senescence induction requires subsequent clearance of the residual cancer cells by $T_H1$ cells and macrophages to protect from cancer progression in a model of hepatocellular carcinoma[36]. In contrast, other data strongly suggest that senescence is a critical barrier that protects better against the progression of cancers than the attempt to clear all cancer cells[61].

It remains open whether senescence in established cancers generally protects against cancer progression or whether this depends on the mode of senescence induction. TNF and IFN-γ are major inducers of apoptosis in cancer cells but some cells escape from TNF- and IFN-γ-induced apoptosis[47]. We have previously shown that TNF and IFN-γ induced senescence in cells escaping from TNF/IFN-γ-mediated apoptosis[31], and here we showed that CIS protected from the regrowth of cancer cells resistant to the elimination by natural or ICB-enhanced cytotoxicity. In RT2-cancers, CIS establishes a stable type of senescence where the SASP does not promote cancer progression for prolonged periods of time, as the senescent RT2-cancer cells remain growth arrested for extended periods of time, even when transplanted into severely immune compromised mice[31]. Importantly, CIS requires the combined action of TNF and IFN-γ, as immune therapies of cancers in the absence of either IFN-γ, of Stat1- or of Tnfr1-signalling strongly accelerate the transformation and growth of cancer cells[19,25].

This is especially relevant in the context of our data showing that melanoma metastases that fail responding to ICB therapies and that grow very fast despite ICB, frequently show functionally relevant gene aberrations in IFN-γ-regulated, cancer-intrinsic senescence-inducing cell cycle regulators. Even though IFN-γ induces a broad spectrum of tumour-protective mechanisms, the data here proved that IFN-γ-dependent senescence induction is a key mechanism required to protect against those cancer cells that escape from cytotoxicity.

## Methods

**Animals.** C3HeB/FeJ mice were purchased from The Jackson Laboratory (Bar Harbor, ME, USA). Syngeneic transgenic TCR2 mice[31,62] express a T cell receptor (TCR) specific for Tag peptide 362-384 on CD4$^+$ T cells, RIP-Tag2 (RT2) mice express the T antigen under control of the rat insulin promotor (RIP) that leads to pancreatic islet cancers (RT2-cancers)[31,62] and double transgenic RT2.$Stat1^{−/−}$ mice (backcross of 129S6/SvEv-$Stat1^{tm1Rds}$ mice[31]) were provided by Taconic and backcrossed to C3HeB/FeJ. OT-I mice (C57BL/6-Tg(TcraTcrb)1100Mjb/J, and NSG (NOD.$Cg$-$Prkdc^{scid}$ $Il2rg^{tm1Wjl}$/SzJ) mice were from The Jackson Laboratory. λ-MYC mice and double transgenic λ-MYC.p21$^{−/−}$ mice (both C57BL/6 background) express a human $MYC$ oncogene under the control of the immunoglobulin λ enhancer and develop an endogenous B-cell lymphoma[43]. Mice with C3HeB/FeJ background, OT-I mice and NSG-mice were bred in the animal facility Tübingen. λ-MYC and λ-MYC.p21$^{−/−}$ mice were bred in the animal facility Munich. All animals were bred under specific pathogen-free conditions. Animal

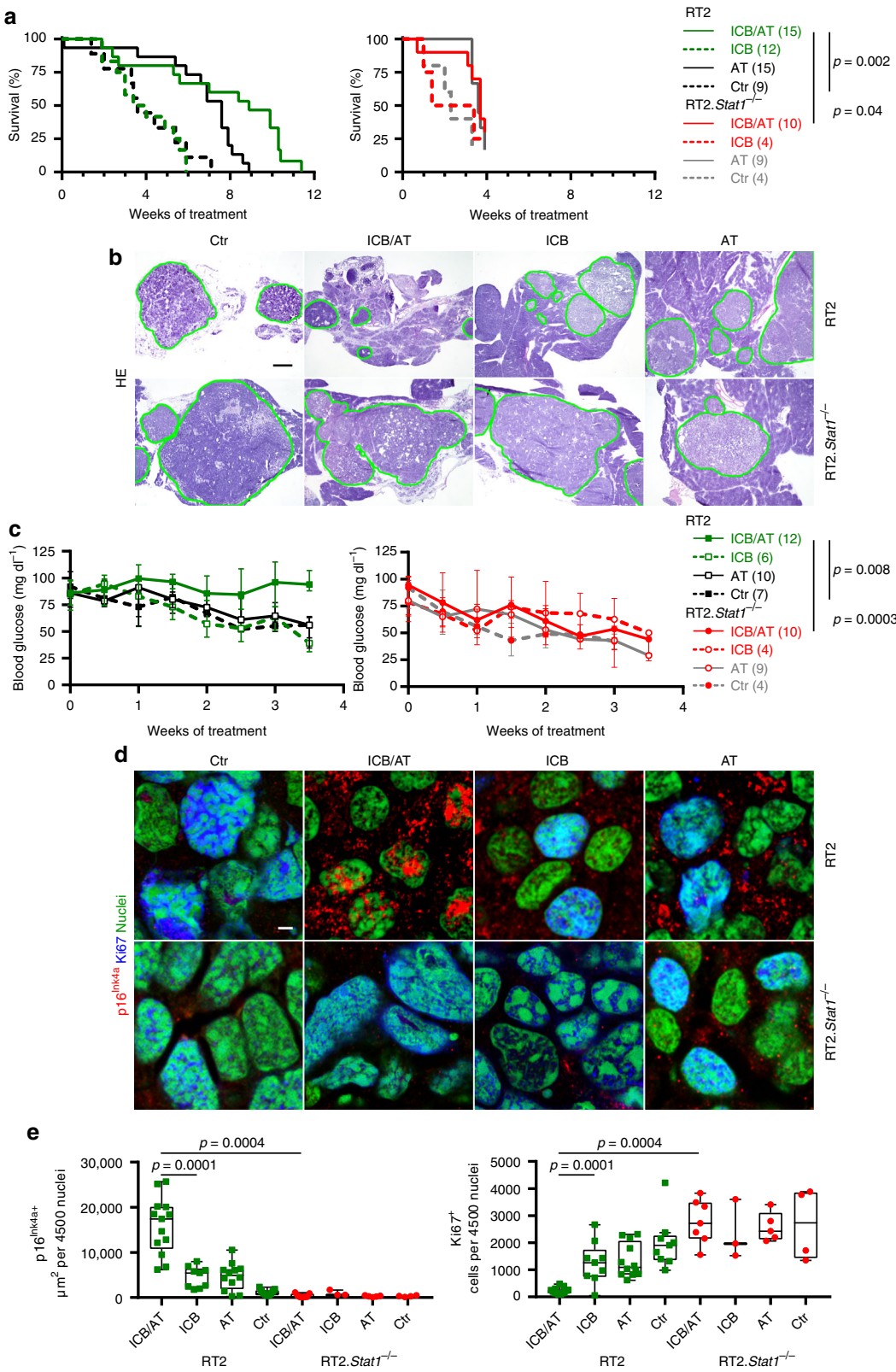

experiments were in accordance with animal welfare regulations and ethical approval was obtained by the local authorities (Regierung von Oberbayern and Regierungspräsidium Tübingen).

**Treatment of RT2-cancers in C3HeB/FeJ mice.** A total of $1 \times 10^6$ cells (in 100 µl NaCl) RT2-, RT2.Stat1[−/−]-, RT2.Cdkn2a[−/−]- or RT2.CRISPR-Cdkn2a-cancer cells were s.c. transplanted, three days after depletion of CD8 cells with 100 µg anti-CD8

mAb[39] (Rm-CD8-2 AK, Core Facility mAb, Helmholtz-Zentrum München). CD8 depletion was repeated every ten days till the tumour lesion became palpable. Once tumours were ≥3 mm, mice were treated with anti-PD-L1 mAb and anti-LAG-3 mAb once per week (initially 500 µg, then 200 µg). Ctr mice received isotype-control mAbs (clone LTF-2 and HRPN, Bio X Cell). Tumour growth was monitored two times per week using a calliper and blood glucose was measured twice a week using an Accu-Check sensor for up to 8 weeks. Cancer size was measured two times per week. Mice were sacrificed after 4 treatment cycles, when the tumour

**Fig. 4 *Stat1*-dependent immune control of endogenous RT2-cancers and induction of Ki67⁻p16^Ink4a+ senescent cancer cells. a** Survival curves of RT2 or RT2.*Stat1*⁻/⁻ mice treated with either isotype control mAbs (Ctr, RT2 N = 9, RT2.*Stat1*⁻/⁻ N = 4), with adoptive transfer of T antigen-specific CD4⁺ T_H1 cells (AT, RT2 N = 15, RT2.*Stat1*⁻/⁻ N = 9), with immune checkpoint inhibitors (ICB, RT2 N = 12, RT2.*Stat1*⁻/⁻ N = 4; anti-PD-L1 and anti-LAG-3) or with ICB/AT (RT2 N = 15, RT2.*Stat1*⁻/⁻ N = 10). For each AT experiment we generated a non-cytotoxic Tag-specific T_H1 cell line. **b** Cancer size shown by representative hematoxylin and eosin staining of pancreas of RT2 (upper panel) or RT2.*Stat1*⁻/⁻ mice (lower panel) after four weeks of treatment. Mice were treated as described in **a**. Green lines depict the size of RT2-cancers after treatment. After 4 weeks of treatment ICB/AT-treated mice (N = 6) had a two-fold smaller islet size than Ctr (N = 7). **c** Time course of blood glucose levels as surrogate marker for the growth of the insulin-producing tumours of RT2 or RT2.*Stat1*⁻/⁻ mice (median ± interquartile range) treated as described in (**a** Ctr, RT2 N = 7, RT2.*Stat1*⁻/⁻ N = 4; AT, RT2 N = 10, RT2.*Stat1*⁻/⁻ N = 9; ICB, RT2 N = 6, RT2.*Stat1*⁻/⁻ N = 4, ICB/AT, RT2 N = 12, RT2.*Stat1*⁻/⁻ N = 10). The mice were sacrificed after 4 weeks of treatment for ex vivo analysis. **d, e** Representative triple-staining for the senescence marker p16^Ink4a (red) and the proliferation marker Ki67 (blue) and for nuclei (green) (**d**), **e** showing quantification of p16^Ink4a+ (left) or Ki67⁺ (right) cancer cells of individual mice, treated as described in **a**, each data point represents the total of three tumour slides measurements (Ctr, RT2 N = 9, RT2.*Stat1*⁻/⁻ N = 4; AT, RT2 N = 12, RT2.*Stat1*⁻/⁻ N = 5; ICB, RT2 N = 9, RT2.*Stat1*⁻/⁻ N = 3, ICB/AT, RT2 N = 13, RT2.*Stat1*⁻/⁻ N = 7), box plots show the median with 25th and 75th interquartile range (IQR), and whiskers indicate 1.5 × IQR. Significance tested by using Log Rank test (**a**, left), Fishers exact test (**a**, right), or two-tailed Mann-Whitney test (**c**, **e**). RT2.*Stat1*⁻/⁻ mice have been censored after 3.7 weeks of treatment. Scale bars 200 μm (**b**) or 2 μm (**d**). Number of mice is given in parenthesis.

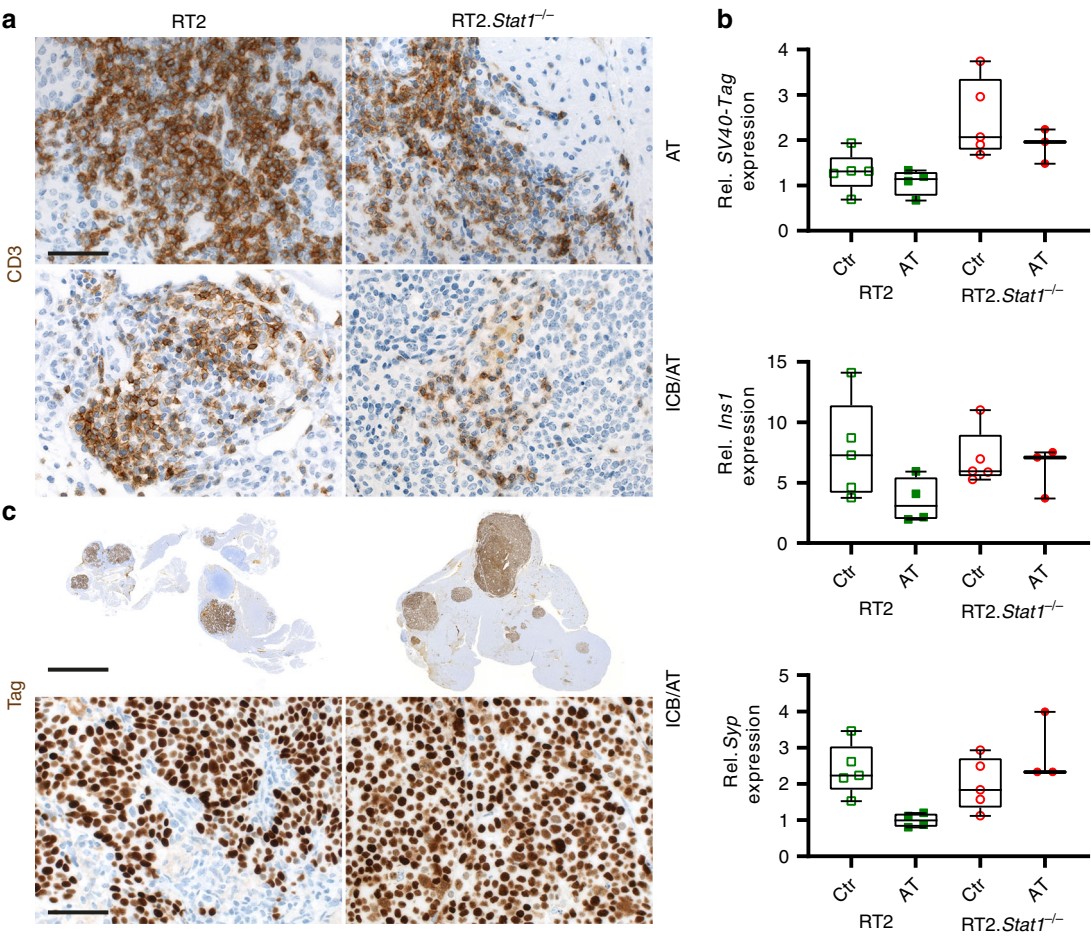

**Fig. 5 Infiltration of either RT2 or RT2.*Stat1*⁻/⁻ cancers by CD3⁺ T cells, mRNA expression of tumour associated antigens (TAA) and protein expression of the SV40-Tag tumour antigen recognised by the transferred TAA-specific T_H1 cells. a** Representative CD3 immune histochemistry images of RT2-cancers from either RT2 or RT2.*Stat1*⁻/⁻ mice treated with adoptive transfer of T antigen-specific CD4⁺ T_H1 cells (AT) or immune checkpoint blockade combined with AT (ICB/AT), scale bar 20 μm. **b** Relative expression of *SV40-Tag*, insulin (*Ins1*) or of synaptophysin (*Syp*) in single tumours isolated from RT2 or RT2.*Stat1*⁻/⁻ mice after control (Ctr) treatment with NaCl or AT. Gene expression was analysed using *Actb* and *Eef1a1* as references. Each point represents one tumour from one mouse (Ctr, RT2 N = 5, RT2.*Stat1*⁻/⁻ N = 5; AT, RT2 N = 4, RT2.*Stat1*⁻/⁻ N = 3), box plots show the median with 25th and 75th interquartile range (IQR), and whiskers indicate 1.5 × IQR. **c** SV40-Tag immune histochemistry images of RT2-cancers were from either RT2 or RT2.*Stat1*⁻/⁻ mice after ICB/AT therapy. Scale bar upper panel 400 μm, lower panel 20 μm.

diameter reached >15 mm, or ulcerated or when blood glucose dropped the second time below 30 mg dl⁻¹. Tumours were collected for ex vivo analysis. The tumour volume was calculated as ellipsoid $V = \frac{3}{4}\pi abc$; $a$, $b$, $c$ are the semi-major axis $\left(\frac{lenght}{2}\right)$, $\left(\frac{width}{2}\right)$, with the assumption that the tumour height is equal to the short tumour side, defined as width $b = c$.

**Treatment of RT2 or RT2.*Stat1*⁻/⁻ mice**. Ten- to eleven-week-old old female RIP-Tag2 (RT2)[31,62] or RT2.*Stat1*⁻/⁻ mice were irradiated with 2 Gy one day before the first i.p. transfer of 1 × 10⁷ tumour antigen-specific T_H1 cells. Tag-specific T_H1 cells were generated from spleen and lymph node cells of TCR2 mice. CD4⁺ T cells were enriched by positive selection over magnetic microbeads. To generate Tag-T_H1 cells CD4⁺ T cells were stimulated with Tag

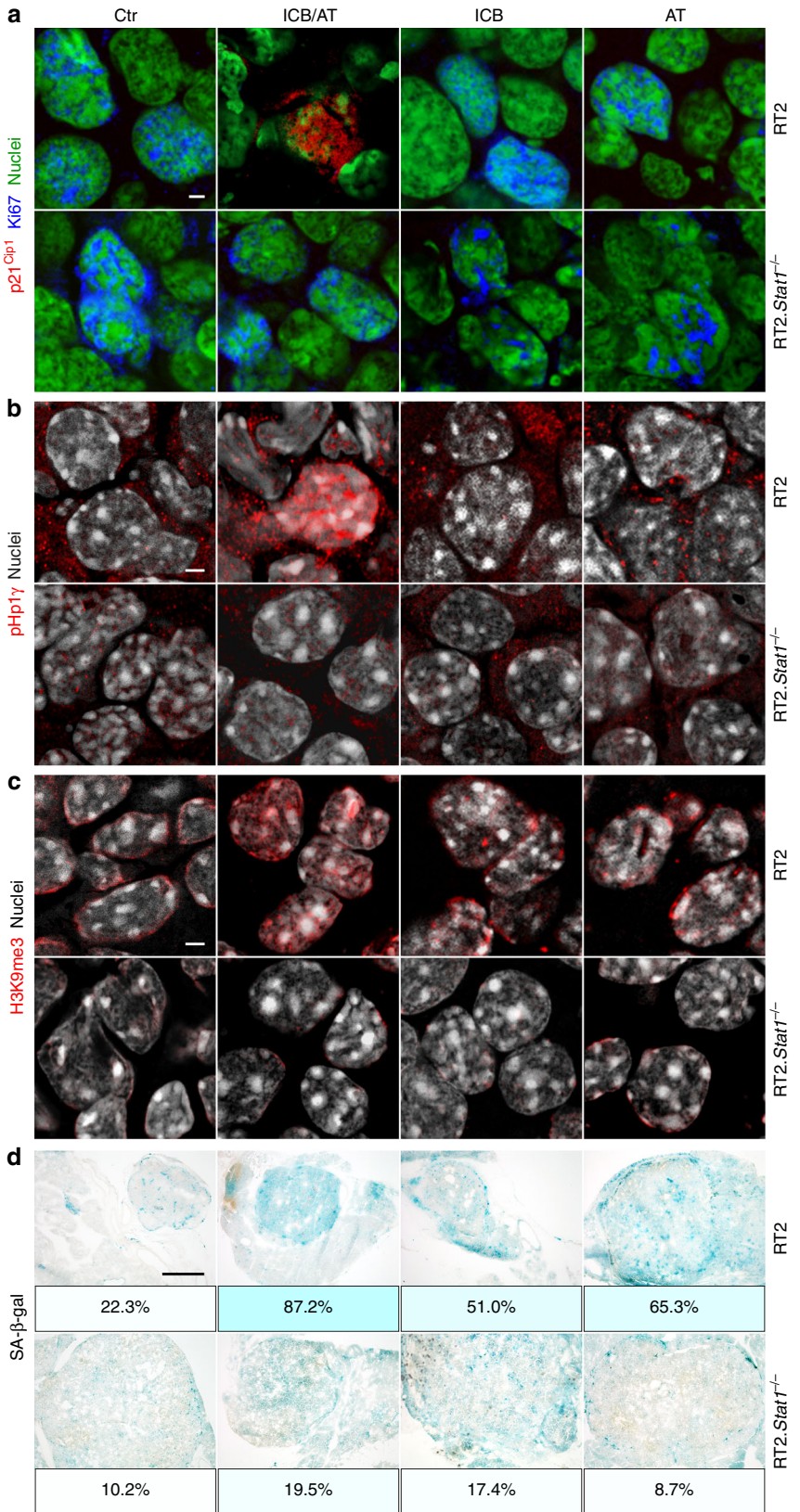

peptide 362–384 (EMC Microcollections), CpG-DNA 1668 (Eurofins MWG Operon) and 11B11 mAb (anti-IL-4; in-house production) and cultured in the presence of irradiated (30 Gy), syngeneic, T-cell-depleted antigen presenting cells (APC)[39]. After 3–4 days, Tag-$T_H1$ cells were expanded in the presence of 50 U IL-2 (hrIL-2). After cultivation for another 8 days, the Tag-$T_H1$ cells were used for adoptive transfer[31]. Cell transfer was applied once weekly. Anti-PD-L1 mAb (clone 10F.9G2, Bio X Cell) and anti-LAG-3 mAb (clone C9B7W, Bio X

Cell) were i.p. injected twice per week (initially 500 µg each, afterwards 200 µg). Ctr mice received isotype-matched control antibodies (clone LTF-2 and HRPN, Bio X Cell) and PBS. Blood glucose was measured twice per week using the HemoCue Glucose 201+ System (HemoCue). Treatment was ended either after 4 treatment cycles for ex vivo analysis of tumour tissue, or when the blood glucose of the mice dropped twice below 30 mg dl$^{-1}$ or when disease reached evidence; mice had no food restriction.

**Fig. 6 *Stat1*-dependent induction of the nuclear senescence markers p21^Cip1, HP1γ, H3K9me3, and of SA-β-gal in RT2-cancers by combined ICB/AT therapy.** Representative fresh frozen cryostat sections of RT2-cancers from either RT2 or RT2.*Stat1*$^{-/-}$ mice with established cancers were treated with either isotype control mAbs (Ctr), with adoptive transfer of TAA-specific T$_H$1 cells (AT), with immune checkpoint inhibitors (ICB; anti-PD-L1 and anti-LAG-3), or with ICB/AT. **a** p21^Cip1 (red), Ki67 (blue), nuclei (green). **b** HP1γ (red), nuclei (white). **c** H3K9me3 (red), nuclei (white). **d** Representative microscopic images of SA-β-gal activity at pH5.5 and percentage of SA-β-gal positive tumour cells in each tumour. The colour evaluation and calculation of the SA-β-gal$^+$ cells are described in Supplementary Fig. 2 and Methods. Scale bars 2 μm (**a–c**), 1000 μm (**d**). Histology was performed in one to three representative tumours from Fig. 4e.

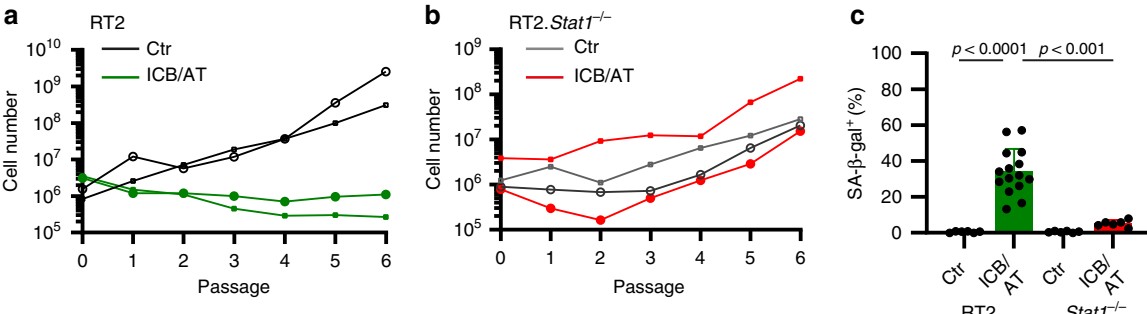

**Fig. 7 *Stat1*-dependent induction of senescence in RT2-cancers by combined ICB/AT therapy. a, b** Either RT2 mice (**a**) or RT2.*Stat1*$^{-/-}$ mice (**b**) with established cancers were either treated with isotype control mAbs (Ctr) or treated with immune checkpoint blockade (anti-PD-L1 and anti-LAG-3) and adoptive transfer of TAA-specific T$_H$1 cells (ICB/AT). After four weeks of treatment cells were isolated from the RT2-cancers and cultured in medium. Cells were counted at each passage. **c** After the passage ≥5, we determined the percentage of the SA-β-gal$^+$ cells. Data show the mean with SD; Ctr, RT2 $n = 6$, RT2.*Stat1*$^{-/-}$ $n = 6$; ICB/AT, RT2 $n = 15$, RT2.*Stat1*$^{-/-}$ $n = 6$. Staining for synaptophysin confirmed that the cultured cells were RT2-cancer cells. Significance was tested by using unequal variances *t*-test.

**Treatment of λ-*MYC* mice**. λ-*MYC* mice and double transgenic λ-*MYC*.*p21*$^{-/-}$ mice received intraperitoneal (i.p.) injections of 100 μg anti-CTLA-4 mAb (clone UC10-4B9, BioLegend) and 100 μg anti-PD-1 mAb (clone J43, Bio X Cell) (ICB) two to four times every ten days, starting at day 55 after birth. Control mice (Ctr) received no treatment. For IFN-γ neutralisation (ICB/anti-IFN-γ mAb), mice were additionally treated with an initial dose of 500 μg on day 54 and later with 300 μg anti-IFN-γ mAb (clone XMG1.2[25], Core Facility mAb, Helmholtz-Zentrum München) 6 h prior to the anti-CTLA-4/ PD-1 mAbs injection. Administration of 100 μg of the IFN-γ-neutralising mAb was continued at ten day intervals until the mice developed lymphomas. T cells were depleted by i.p. injection of 1 mg anti-pan-T-cell mAb MmTC[63] at day 54. Treatment was repeated every 3 to 4 days using doses of 400 μg. Mice were sacrificed as soon as tumours became clinically visible.

**Immunofluorescence staining**. Fresh frozen 5 μm serial cryosections of lymph nodes from λ-*MYC* mice, whole pancreas from RT2 and RT2.*Stat1*$^{-/-}$ mice or isolated RT2-, *Stat1*-, or *p16*$^{Ink4a}$-deficient RT2-cancer cells were fixed with perjodate-lysine-paraformaldehyde. Sections were blocked using donkey serum and stained with primary antibodies using rabbit-anti-p16$^{Ink4a}$ (clone CDKN2A, catalogue number AHP1488, dilution 1:100 Bio-Rad/Serotec), rat-anti-Ki67 (clone SolA15, catalogue number 14-5698-82, dilution 1:100, Thermo Fisher Scientific/ eBioscience), rat-anti-p21^Cip1 (clone HUGO291, catalogue number, dilution 1:100, Abcam), goat-anti-CD20 (clone M-20, catalogue number sc-7735, dilution 1:50, Santa Cruz), rabbit-anti-CD3 (clone SP7, catalogue number C1597R06, dilution 1:100, DCS), rabbit-anti-pHP1γ phospho S93 (aa 50-150, catalogue number ab45270, dilution 1:50, Abcam), rabbit-anti-H3K9me3 (aa 1-100, catalogue number ab8898, dilution 1:100, Abcam), rat-anti-PD-L1 (clone MIH6, catalogue number ab80276, dilution 1:50, Abcam) to use a mAb that recognises a different PD-L1 epitope than the applied in vivo mAb from clone 10F.9G2, rabbit-anti-beta 2 microglobulin antibody (clone EP2978Y, catalogue number ab75853, dilution 1:50, Abcam), goat-anti-CD3ε (clone M-20, catalogue number sc-1127, dilution 1:20, Santa Cruz), rat-anti-CD8a (clone 53-6.7, catalogue number 14-0081, dilution 1:1000, eBioscience), rat-anti-F4/80 (clone BM8, catalogue number 14-4801-82, dilution 1:50, eBioscience), rat-anti-MHC Class II antigen I Ak,d,b,q,r (ER-TR3) (catalogue number NB100-64961, dilution 1:200 Novusbio), armenian hamster-anti-CD49b (Integrin alpha 2) (catalogue number 14-0491 dilution 1:50, eBioscience), rabbit-anti-gamma H2A.X (phospho S139) (aa 100-200, catalogue number ab2893, dilution 1:200, abcam), goat-anti-DNA PKcs (aa 4078-4128, catalogue number ab168854, dilution 1:500, abcam), rabbit-anti-synaptophysin (aa 41-62, catalogue number NB300-653, dilution 1:200, Novusbio), rabbit-anti-CD161 antibody (clone ERP21236, catalogue number ab234107, dilution 1:100, abcam), rabbit-anti-Foxp3 (aa 43-100, catalogue number NB 100-39002, dilution 1:200, Novusbio). Bound antibodies were visualised using donkey-anti-rabbit-Cy3 (catalogue number 711-166-152, dilution 1:500 Dianova), donkey-anti-rat-Alexa 647 (catalogue number 712-606-153, dilution 1:500 Dianova), donkey-anti-rat-Cy3

(catalogue number 712-166-153, dilution 1:500, Dianova), donkey-anti-goat-Cy3 (catalogue number 705-166-147, dilution 1:500, Dianova) and donkey-anti-rabbit-Alexa 488 catalogue number 711-546-152, dilution 1:500, Dianova). For nuclear staining, Yopro (catalogue number Y 3603, dilution 1:1000, Invitrogen) or DAPI (catalogue number D9542, dilution 1:2000 Sigma) was used. Sections were analysed using a LSM 800 confocal laser scanning microscope (Zeiss Oberkochen). Images were processed with the software ZEN 2.3 (blue edition) and the Image Analysis Module.

**Quantification of immunofluorescence**. Tumour areas of images derived from three individual stained cryosections per tissue sample were analysed. For quantification of the Ki67 staining, 4500 Yopro stained nuclei were counted for Ki67$^+$ cells (1500 per slide, three serial slides per mouse). For quantification of p16$^{Ink4a}$, the area (μm$^2$) of the nuclear p16$^{Ink4a}$ signal derived from 4500 Yopro stained nuclei (1500 per slide three serial slides per mouse) was measured.

**Hematoxylin and eosin staining**. Hematoxylin and eosin staining of the serial cryosections was performed according to standard procedures.

**Immunohistochemistry staining**. Immunohistochemistry was performed on an automated immunostainer (Ventana Medical Systems) according to the company's protocols for open procedures with slight modifications. 5 μm sections were stained with rabbit-anti-human/mouse-CD3 (clone SP7, catalogue number CI597C01, dilution 1:50, DCS-diagnostics) and PAB101 (Tag-specific mouse IgG2a mAb; dilution 1:200; in-house production).

**SA-β-gal detection**. 20 μm serial cryosections were fixed in 2% formaldehyde/ 0.25% glutaraldehyde and washed in PBS/MgCl$_2$. Slides were incubated in X-gal (5-bromo-4-chloro-3-indolyl-beta-D-galactopyranoside) staining solution (1 mg/mL X-Gal, 1 mM MgCl$_2$, 5 mM K3Fe(CN)6, 5 mM K4Fe(CN)6 in PBS, pH 4.0, 5.5, and 7.0) up to 10 h at 37 °C.[64] The stained slides were rinsed in PBS/MgCl$_2$ and analysed using a Nikon Eclipse 80i microscope; magnification ×4.

**SA-β-gal positive percentage quotation with Adobe Photoshop CS6**. All images were analysed with the "White Balance Tool" to obtain the same white background, followed by "Quick Mask Mode" to identify only the tumour area. Using this area the arithmetic mean of the blue-green values was obtained by the filter "Blur Average Tool". The "Eye Dropper Tool" was used to identify the red, green and blue (RGB) colour code to get the corresponding colour field (Supplementary Fig. 2a). Afterwards the tumour area was reselected using the previous created "Quick Mask Mode Layer". The pixels outside the tumour area were deleted by inversing the selection. The number of pixels in the histogram correlates with the tumour area. "The Posterize Tool" separated the different tonal values, in

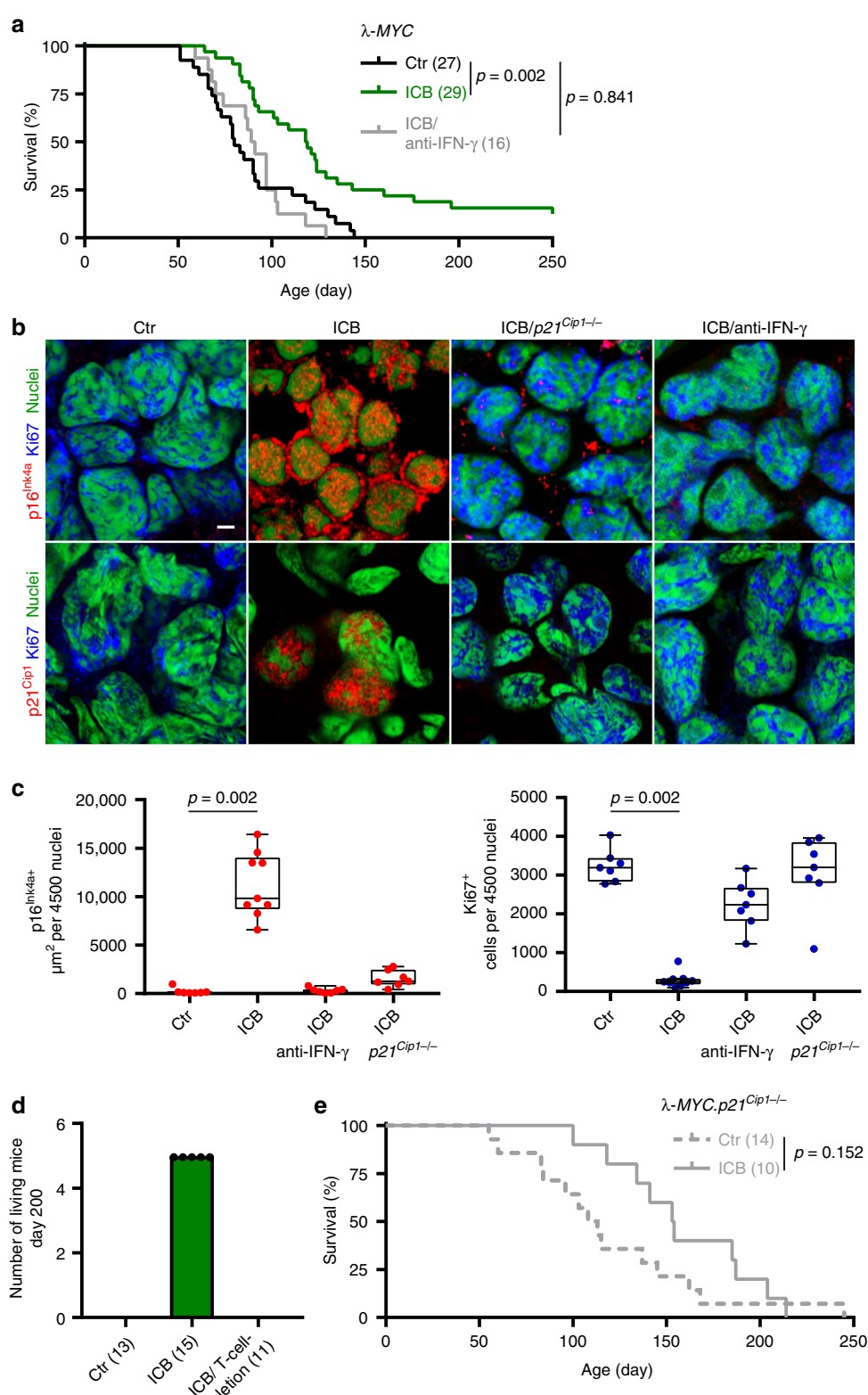

our case blue-green. "The Magic Wand Tool" was used to select and delete the white pixels (Supplementary Fig. 2b). The number of pixels correlates with the area of SA-β-gal stained tumour cells. The SA-β-gal stained tumour cells in Fig. 2d, Fig. 6d and Fig. 9c were calculated and given in percent (blue pixel of total pixels of the tumour area).

**SA-β-gal activity for electron microscopy**. Small tissue samples (semi-thin 0.5 μm and for electron microscopy ultra-thin 20 nm, respectively) were fixed in fixation solution (0.25% Glutaraldehyde in 2% PFA) and washed in PBS/MgCl₂

solution. X-gal staining solution was added for 12 h. Samples were washed in PBS/MgCl₂ solution afterwards followed by Karnovsky fixation. Samples were embedded in glycid ether for electron microscopic analysis[64].

**Electron microscopy**. SA-β-gal stained cryosections were fixed with Karnovsky's fixative for 24 h. Post-fixation was based on 1% osmium tetroxide containing 1.5% K-ferrocyanide in cacodylate buffer. After following the standard methods, blocks were embedded in glycide ether and cut using an ultra-microtome (Ultracut, Reichert, Vienna, Austria). Ultra-thin sections (30 nm) were mounted on copper

**Fig. 8 IFN-γ and *p21^Cip1*-dependent immune control of λ-*MYC*-induced lymphomas by ICB therapy. a** Survival curves of control (Ctr $N = 27$), or immune checkpoint inhibitor (ICB; anti-CTLA-4 and anti-PD-1 $N = 29$) treated λ-*MYC* mice, or of λ-*MYC* mice treated with ICB and anti-IFN-γ ($N = 16$). **b, c** Triple-staining for the senescence marker p16$^{Ink4a}$ (red, upper panel) or p21$^{Cip1}$ (red, lower panel), the proliferation marker Ki67 (blue), and for nuclei (green); scale bar 2 μm; representative pictures from Fig. 8c (**b**). Box plots with individual data points representing p16$^{Ink4a}$+ (**c**, left) or Ki67$^+$ nuclei (**c**, right) of B cells from Ctr- ($N = 7$), ICB- ($N = 9$), ICB and anti-IFN-γ-treated λ-*MYC* mice ($N = 7$), or ICB-treated λ-*MYC.p21^{Cip1−/−}* mice ($N = 7$). Lymph nodes were isolated at similar ages. Each point represents triplicates from one mouse, box plot show the median with 25th and 75th interquartile range (IQR), and whiskers indicate 1.5 × IQR. **d** λ-*MYC* mice from the individual treatment groups living at day 200, Ctr $N = 13$, ICB $N = 15$, ICB and T-cell-depletion $N = 11$. **e** Survival curves of either control (Ctr, $N = 14$) or ICB-treated λ-*MYC.p21^{Cip1−/−}* mice ($N = 10$). Significance tested by using Log Rank test (**a**, **e**), two-tailed Mann-Whitney test (**c**).

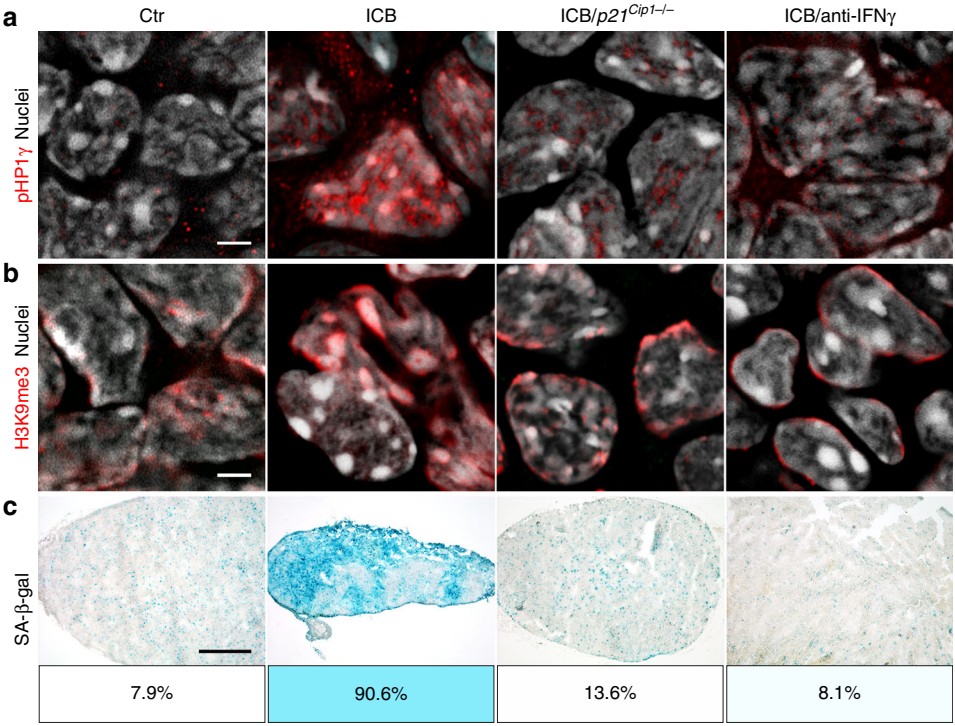

**Fig. 9 IFN-γ and *p21^Cip1*-dependent senescence induction in B cells of λ-*MYC* mice during ICB. a–c** Fresh frozen cryostat sections of representative lymph nodes of λ-*MYC* or λ-*MYC.p21^{Cip1−/−}* mice. λ-*MYC* mice were controls (Ctr) or treated with anti-CTLA-4 and anti-PD-1 mAbs (ICB) or anti-CTLA-4, anti-PD-1 and anti-IFN-γ mAbs (ICB/anti-IFN-γ), or λ-*MYC.p21^{Cip1−/−}* mice were treated with ICB (ICB/p21$^{Cip1−/−}$). pHP1γ (red), nuclei (white) (**a**). H3K9me3 (red), nuclei (white) (**b**). Representative microscopic images of SA-β-gal activity at pH5.5 and percentage of SA-β-gal positive tumour cells in each tumour (**c**). The colour evaluation and calculation of the SA-β-gal$^+$ cells are described in Supplementary Fig. 2 and Methods. Scale bars 2 μm (**a**, **b**), 1000 μm (**c**). The SA-β-gal data are representative for three individual tumours. Immune fluorescence was performed in one to two representative tumours from Fig. 8c.

grids and analysed using a Zeiss LIBRA 120 transmission electron microscope (Carl Zeiss, Oberkochen, Germany) operating at 120 kV.

**Cell culture.** Adherent primary murine RT2-cancer cells (described above), B16-F10 melanoma cells (catalogue number CRL-6475, ATCC) and B16-OVA melanoma cells (gift from R. Dutton, Trudeau Institute, New York, USA) were cultured in Dulbecco's Modified Eagle's Medium (DMEM) supplemented with 10% foetal calf serum (FCS), nonessential amino acids, sodium pyruvate, antibiotics, and 50 μM 2-mercaptoethanol at 37 °C and 7.5% $CO_2$. The murine melanoma cell lines B16 and B16-OVA were cultured in DMEM medium, containing 10% FCS and penicillin/streptomycin (100 U ml$^{−1}$; all from Biochrom AG), at 37 °C and 5% $CO_2$. The human patient-derived xenograft (PDX) cell lines were cultured in RPMI-1640 medium supplemented with 10% FCS, nonessential amino acids, sodium pyruvate, antibiotics, and 5 μg ml$^{−1}$ Plasmocin (Invitrogen) to treat infections of clinical samples, at 37 °C and 5.0% $CO_2$. Cell line supernatants were tested with a mycoplasma detection PCR Venor GeM (Minerva Biolabs GmbH) regularly every 4 weeks.

**Generation of primary murine RT2- or RT2.*Stat1*^−/−-cell lines.** RT2-cancers were isolated from the pancreas of female RT2 or RT2.*Stat1*^−/− mice by intraductal injection of collagenase (1 mg ml$^{−1}$, Serva, Heidelberg, Germany)[25,31]. After injection, the pancreata were harvested, digested in collagenase solution at 37 °C for 10 min, and then mechanically disrupted. Whole encapsulated tumours were separated under a dissection microscope (Leica Microsystems) and further

processed for immunofluorescence microscopy, immunohistochemistry or gene expression analysis. Alternatively, single tumour cells were obtained by incubation of the tumours in 0.05% trypsin/EDTA solution (Invitrogen) at 37 °C for 10 min. After incubation, RT2-cancer cells were seeded onto tissue culture plates.

**Generation of primary p16^Ink4a^-deficient RT2-cell lines.** CDKN2a-loss variants on chromosome 4, qC4.A were generated by random selection from CIS- or ICB-resistant RT2-cancer cells.

**Generation of CRISPR-mediated deletion of *Cdkn2a* in primary RT2-cell lines.** gRNAs targeting Exon 2 (position 120-142 and 125-157) of murine *Cdkn2a* (sequence from Ensembl genome browser 97) were designed using CRISPRdirect[65] (g*Cdkn2a*_1: 5′-gcgtcgtggtggtcgcacagg-3′, g*Cdkn2a*_2: 5′-gacacgctggtggtgctgcac-3′). As a control a previously described gRNA targeting GFP gRNA targeting GFP (sgRNA target site sequence: 5′-gggcgaggagctgttcaccg-3′)[66] was used. The DNA oligos were ordered from Sigma Aldrich. Equimolar amounts of complementary DNA oligos were annealed and phosphorylated in T4 ligation buffer (NEB). The reaction was performed at 37 °C for 30 min, 95 °C for 5 min, followed by a gradual temperature reduction to 25 °C (5 °C/min). The vector pSpCas9(BB)-2A-GFP (pX458; Addgene plasmid ID: 48138) was digested with the *BbsI* restriction enzyme (NEB) and dephosphorylated using Calf Intestinal Alkaline Phosphatase (NEB). The ligation reaction (NEB) was performed overnight. The plasmids were sequenced (Microsynth Seqlab) with a hU6 sequencing primer (LKO.1 5′: gactat-catatgcttaccgt. For transfection, plasmid DNA was generated by using the Qiagen

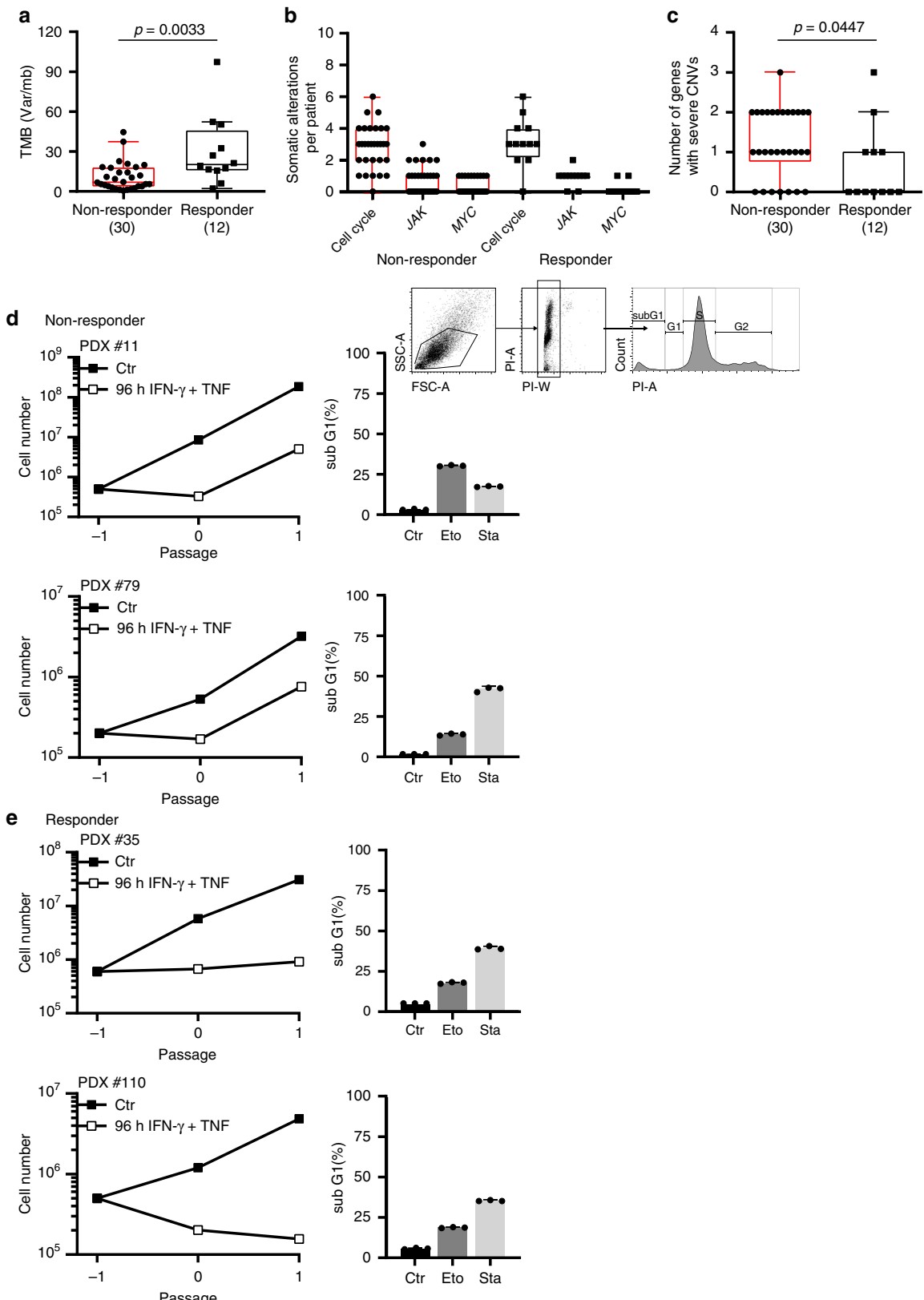

EndoFreeMaxi Kit.[66] The primary RT2-cell lines were transfected using the Quiagene Effectene Transfection Reagent.

**Generation of primary human melanoma cell lines.** Melanoma tissue was cut into small pieces, digested in HBBS (w/o $Ca^{2+}$ and $Mg^{2+}$) with 0.05% collagenase, 0.1% hyaluronidase and 0.15% dispase, at 37 °C for 1 h and filtered through a cell strainer (100 μm mesh). The melanoma cell suspension was implanted with Matrigel (Corning Life Sciences) subcutaneously in NSG (NOD.Cg-Prkdc$^{scid}$ Il2rg$^{tm1Wjl}$/SzJ) mice, patient derived xenografts (PDX). Tumour grafts were harvested when they reached a diameter of 10 to 15 mm, digested as above and the single cells were taken into culture using RPMI1640 with Hepes and L-Glutamine, containing 10% foetal bovine serum, 1% penicillin-streptomycin at 37 °C with 5% $CO_2$ and 95% humidity. For cryo-preservation cell pellets were resuspended in Biofreeze medium (Biochrom/Merck)

**Fig. 10 Loss of senescence-inducing and amplification of senescence-inhibiting cell cycle regulator genes in melanoma metastases of patients resistant to ICB. a–c** Sequencing data from metastases of non-responder patients ($N = 30$) versus metastases of responder patients ($N = 12$). In non-responder patients disease progressed within 3 months of ICB therapy. In responder patients metastases regressed ≥1 year of ICB therapy. Tumour mutational burden (TMB, copy number variations) (**a**). Number of tumour-specific alterations in 19 genes of the cell cycle, *JAK* or *MYC* pathway (**b**). Number of genes with homozygous deletions in cycle inhibitors or amplifications ≥ 3fold in cell cycle promoters. Genes analysed: *CDKN2A/B/C, CDKN1A/B, RB1, TP53, JAK1/2/3, CCND1/2/3, CDK4/6, CCNE1, MDM2/4, MYC*. We performed paired panel sequencing of the tumour DNA and of normal DNA to identify tumour-specific alterations. Box plots show the median with 25th and 75th interquartile range (IQR), and whiskers indicate 1.5 × IQR (**c**). **d, e** Growth curves of patient derived melanoma lines from two metastases of non-responder patients (**d**) or from two metastases of responder patients (**e**). Cells were cultured for the senescence assay either with medium (Ctr) or with medium containing 100 ng ml$^{-1}$ IFN-γ and 10 ng ml$^{-1}$ TNF for 96 h, washed and then cultured with medium for another 4–6 days. **d, e** Apoptosis assay with melanoma cells derived from two non-responder patients (**d**, including gating strategy) or two responder patients (**e**) were exposed to either medium (Ctr) or etoposide (Eto, 100 μM) or staurosporine (Sta, 0.5 μM) for 24 h for apoptosis induction and then stained with propidium iodide. SubG$_1$ cells were detected by flow cytometry analysis; data show the geometric mean with geometric SD, $n = 3$. Significance tested by using two-tailed Mann-Whitney test (**a–c**).

---

and 1 ml per cryotube of the cell suspension was frozen for short-term storage at −80 °C and for long-term storage in liquid nitrogen.

**Senescence assay.** RT2-, RT2.*Stat1*$^{-/-}$, RT2.*p16*$^{Ink4a}$-, RT2-CRISPER-control- or RT-CRISPER-*p16*$^{Ink4a}$-cancer cells or human melanoma-derived (PDX) cells were cultured with medium containing 100 ng ml$^{-1}$IFN-γ and 10 ng ml$^{-1}$ TNF or exclusively with medium (Ctr) for 96 h, washed, then cultured with medium for another 3–4 days (till the Ctr cells reach confluence) and counted[31].

**SA-β-galactosidase activity assay in vitro.** After treatment with IFN-γ and TNF, RT2-cancer cells were fixed for 15 min at room temperature, and then stained for 16 h at 37 °C using the β-galactosidase staining kit (United States Biological)[31]. In addition, cell nuclei were stained with 4′,6-diamidin-2-phenylindole (DAPI; Invitrogen). SA-β-gal-positive and -negative cells were counted using a Zeiss Axiovert 200 microscope equipped with Visiview software and analysed by using ImageJ software (NIH).

**Apoptosis assay.** For apoptosis assay, cancer cells were exposed to etoposide (Eto, 100 μM, Bristol-Meyers-Squibb, ETOPOPHOS), staurosporine (Sta, 0.5 μM, Bio-Vision) or medium (Ctr) for 24 h and stained for propidium iodide and annexin V. Annexin V-positive and subG1 cells were detected and quantified by flow cytometry. Analysis was performed on a LSRII cytometer (BD Bioscience) and analysed by FlowJo software version 10.

**Chromium release assay.** CD8$^+$ cytotoxic T cells (CTL) were generated from spleen and lymph node cells of OT I-transgenic mice. CD8$^+$ T cells were enriched by positive selection over magnetic microbeads. To generate CTL cells CD8$^+$ T cells were cultured in the presence of irradiated (30 Gy), syngeneic, T-cell-depleted antigen presenting cells (APC)[39] in the presence of IL-2 (10 U ml$^{-1}$), IL-12 (5 ng ml$^{-1}$), and anti-IL-4 Ab (10 μg ml$^{-1}$).[31] $2.5 \times 10^6$ target cells were labelled with 250 μCi (9.25 MBq)$^{51}$NaCr (Hartmann Analytic) at 37 °C for 1.5 h, washed and plated into microtitre round bottom plates at $1 \times 10^4$ cells per well. CD8$^+$ effector cells were added to target cells in the ratio 40 to 1 and incubated at 37 °C for 4 h. B16-OVA cells were used as positive controls, B16-F10 melanoma cells were used as negative controls. Spontaneous release in the absence of effector cells was <30% of the maximal release induced by 1% Triton X-100. After incubation, 50 μl supernatant per well was mixed with 200 μl scintillation cocktail (Ultima Gold, PerkinElmer) and measured in a liquid scintillation counter (MicroBeta, PerkinElmer).

**Patients and specimen collection.** Patients with metastatic melanoma were treated with either anti-PD-1 mAb or combined anti-CTLA-4 and anti-PD-1 mAbs. Non-responder patients were 30 patients (43.3% female, 61.5 median age, 22–89 age range) that progressed during the first 3 month of therapy. Responder patients were 12 patients (33.3% female, 56.5 median age, 27–75 age range) that had either a partial (>30%) or complete tumour regression over more than 1 year. Tumour biopsies were compared to healthy tissue from the safety margins as control tissue. Ethical approval was obtained from the Ethics Committee Tübingen. All patients had signed the written informed consent form for research analyses. The study was carried out in accordance with the Declaration of Helsinki and good clinical practice.

**Sequencing.** All patient samples were analysed using a hybridisation-based custom gene panel. Since the patient samples were collected in clinical routine, three different versions of the panel were used (ssSCv2, ssSCv3 and ssSCv4). The number of target genes was increased from one version to the next starting from 337 genes on ssSCv2, to 678 genes on version ssSCv3 and 693 on the current version ssSCv4. The panels were designed to detect somatic mutations (SNVs), small insertions and deletions (INDELs), copy number alterations (CNAs)

and selected structural rearrangements Supplementary Data Tables 1–3 (custom gene panel ssSCv2 ssSCv3 ssSCv4). The library preparation and in solution capture of the target region was performed using the Agilent SureSelectXT and SureSelectXT$^{HS}$ reagent kit (Agilent, Santa Clara, CA). DNA from tumour (FFPE) and matched normal controls (blood) were sequenced in parallel on a Illumina NextSeq500 using 75 bp paired-end reads. The tumour samples were sequenced to an average sequencing depth of coverage of 511x and the normal control samples to 521×, respectively. An in-house developed pipeline, called "megSAP" was used for data analysis [https://github.com/imgag/megSAP, version: 0.1-733-g19bde95 and 0.1-751-g1c381e5]. The sequencing reads were aligned to the human genome reference sequence (GRCh37) using BWA (vers. 0.7.15)[67]. Variants were called using Strelka2 (vers. 2.7.1) and annotated with SNPeff/SnpSift (vers. 4.3i)[68,69]. The overall mutational rate was as calculated using the formula:

$$\left[ \frac{\left( \frac{Somatic - Known - Tumorgenes}{Target size} \times Genome\ size \right) + Tumorgenes}{Genome\ size} \right].$$ For validity and clinical relevance, an

allele fraction of ≥ 5% (i.e. ≥10% affected tumour cell fraction) was required for reported mutations (SNVs, INDELs). Copy number alterations (CNAs) were identified using ClinCNV [https://doi.org/10.1101/837971, https://doi.org/10.1101/837971v1], a method for multi-sample CNV detection using targeted or whole-genome NGS data. The method consists of four steps: (i) quantification of reads per target region, (ii) normalisation by GC-content, library size and median-coverage within a cohort of samples sequenced with the same NGS panel, (iii) calculation of log2-fold changes between tumour and normal sample, (iv) segmentation and CNV calling. Using log2-fold changes ClinCNV estimates statistical models for different copy number states per region (conditioned by tumour sample purity) and reports a likelihood for each statistical model, assuming that the majority of samples are diploid at a focal region. The log-likelihood of the diploid model is subtracted from alternative models, resulting in positive likelihoods for true alternative copy number states. Finally, maximum segments of contiguous regions with positive log-likelihood ratios are identified in an iterative manner. Segments consisting of at least three regions with log-likelihood ratio ≥ 40 and CN state ≤ 1.5 or ≥ 3 are reported as CNVs. Quality control (QC) parameters were collected during all analysis steps[70].

**Comparative genomic hybridisation (CGH) array.** DNA was isolated from RT2-cancer cells or reference spleen (wildtype male) tissue with the DNeasy Blood & Tissue Kit (Qiagen) according to the manufacturer's instructions. DNA was labelled using the SureTag Complete DNA Labelling Kit (Agilent Technologies) and hybridised on an Agilent Mouse Genome CGH Microarray, 2 × 105 K (Agilent Technologies), and the image was analysed using Feature Extraction 10.5.1.1 and Agilent Genomic Workbench Lite Edition 6.5 with Genome Reference Consortium Mouse Build 38.

**Quantitative PCR.** RT2-cancer cells were harvested by trypsin digestion and snap frozen in liquid nitrogen. RNA was prepared using the Nucleospin RNA Mini kit (Macherey-Nagel); cells were lysed using Tris(2-carboxyethyl)phosphine (TCEP)-containing RL1 buffer, followed by DNase digestion (Invitrogen). RNA quality was controlled by agarose gel electrophoresis and by OD600 measurements using a photometer (Eppendorf AG). Complementary DNA was prepared using the iScript cDNA synthesis kit (Bio-Rad Laboratories). Quantitative PCR was performed with SybrGreen using a LightCycler 480 (Roche). Gene expression was analysed using qbase software (Biogazelle) based on the delta-delta-CT-method and reference genes were evaluated using geNorm (feature of the q base software, Biogazelle). The following primers were used: *SV40-Tag* sense 5′-tcc act cca caa ttc tgc tct-3′, antisense 5′-ttg ctt ctt atg tta att tgg tac aga-3; *Cdkn2a* sense 5′-ttg ccc atc atc atc acct-3′, antisense 5′-ggg ttt tct tgg tga agt tcg-3′; *Actb* sense 5′-cta agg cca acc gtg aaa ag-3′, antisense 5′ acc aga ggc ata cag gga ca 3′; *Eef1a1* sense 5′ aca cgt aga ttc cgg caa gt 3′, antisense 5′ agg agc cct tcc cca tctc 3′.

**Magnetic resonance imaging**. Magnetic resonance imaging (MRI) was performed with ten-week-old RT2 mice under 1.5% isoflurane anaesthesia using a 7 T small animal magnetic resonance scanner (ClinScan, Bruker Biospin MRI, Ettlingen, Germany) equipped with quadrature mouse whole-body coil with an inner diameter of 35 mm. For detection of the β-cancers and general anatomic information, a T2-weighted 3D turbo spin-echo-sequence (TE/TR 205/3,000 ms, image matrix of 160 × 256, slice thickness 0.22 mm) was used. Respiration was monitored and used for triggering MRI data acquisition. The acquired MRI data was visualised using Inveon Research Workplace software (Siemens Preclinical Solutions, Knoxville, TN, USA).

**Light sheet fluorescence microscopy**. Five-week-old RT2 or RT2.$Stat1^{-/-}$ mice were treated with TAA-$T_H$1 cells or NaCl as described. Mice were injected i.v. with 50 μg of Alexa Fluor 700 anti-mouse CD4 mAb clone GK1.5 and Alexa Fluor 647 anti-mouse CD11b mAb clone M1/70 (BioLegend), 48 h after the second treatment. After 2 h, organs were harvested and fixed in 4% paraformaldehyde/PBS solution at 4 °C for 8 h. Tissue was dehydrated at room temperature using increasing concentrations of ethanol (30, 50, 70, 80, 90%) for 2 h each and in 100% ethanol at 4 °C overnight. Tissues were incubated in n-hexane for 2 h and then cleared using two parts benzyl benzoate and one part benzyl alcohol (Sigma-Aldrich) three times for 30 min. Air exposure was strictly avoided during this step. The samples were then visualised and analysed using a custom-built laser scanning light sheet microscope using a high NA 20× magnification and reconstructed using the IMARIS software (Bitplane).

**Flow cytometry**. Five-week-old RT2 or RT2.$Stat1^{-/-}$ mice were irradiated with 2 Gy one day before the first i.p. transfer of $1 \times 10^7$ TAA-$T_H$1 cells or NaCl[25]. 48 h after the second treatment mice were sacrificed. The pancreatic lymph node was separated from the pancreas tissue of each mouse, the pancreas was homogenised via a 200 μm cell strainer in DMEM media containing 20% FCS at 4 °C, ery-throcytes were lysed with ACK lysis buffer (Cambrex), and samples were stained for 30 min at 8 °C with fluorochrome-conjugated antibodies (anti-mouse CD4-Pacific Blue, clone GK1.5, catalogue number 100428; anti-mouse CD8a-PE-Cy7, clone 53–6.7, catalogue number 100722; anti-mouse CD45.2-APC-Cy7, clone 104 catalogue number 101224; anti-mouse-CD11c-APC, clone N418, catalogue number 117310; anti-mouse CD11b-Pacific blue, clone M1/70, catalogue number 101224; or corresponding isotype controls (dilution 1:150, BioLegend)). Cells were separated again via a 50 μm cell strainer. Flow cytometry was performed using a FACS Aria and analysed with DIVA software (Becton Dickinson). Alternatively, flow cytometry analyses with $T_H$1 cells or RT2-cancer cells which have been stained with fluorochrome-conjugated antibodies (anti-mouse-PE-IFN-γ, clone XMG1.2, catalogue number 505807; anti-mouse-PE-TNF-α, clone MP6-XT22, catalogue number 506306; anti-mouse-PE IL-4, clone 11B11, catalogue number 504104; anti-mouse- PE β2-microglobulin, clone A16041A, catalogue number 154503; anti-mouse PE/Cy7 CD274 (B7-H1, PD-L1), clone 10F.9G2, catalogue number 124313 (dilution 1:250, BioLegend)) were performed on a LSRII cytometer (BD Bioscience) and analysed by FlowJo software version 10.

**Statistical analysis**. The experiments were not randomised. The investigators were not blinded to allocation during the experiments or outcome assessment. No power calculations were used, but sample sizes were selected on the basis of previous experiments; in vitro results were based on three independent experiments to guarantee reproducibility of findings. The statistics software JMP version 12.2.0 (SAS Institute) and GraphPad Prism version 6 (GraphPadSoftware, California, USA) were used for statistical analyses and for the generation of diagrams. To address the question whether the treatment effect was different between two genotypes, the decadic logarithms of tumour volumes were analysed in ANCOVAs, using the nominal factors "mouseID" (nested under "treatment" and "genotype"), "treatment" and "genotype" as well as the combination "treatment" and "genotype"; "time" was used as continuous factor; finally, the combinations of "mouse ID" and "time", "treatment" and "time", "genotype" and time" and the most important combination "treatment" and "genotype" and "time" were used. For purpose of normalisation, decadic logarithms of tumour volume were used in the analyses; zero observations were replaced by half the minimum of positive values before calculating the logarithms. Group comparisons were made with non-parametric, unpaired, two-tailed Mann-Whitney (Wilcoxon) tests or parametric, unpaired, two-tailed t test with Welch's correction for unequal variances. Log-rank test was used for the comparison of survival from λ-MYC mice, RT2 mice and tumour latency curves. Because of disparate censoring between RT2 mice and RT2.$Stat1^{-/-}$ mice Fisher's exact tests was used to compare the survival (as indicated in the text). N refers to the number of patients, mice or samples and cell lines from different mice, respectively.

## Data availability

Primary generated cancer cells are available from the authors. The sequencing data from patient samples have been deposited in the EGA database under the accession code EGAS00001004151. Comparative genome hybridisation array data have been deposited in the NCBI GEO database under the accession code GSE142192. All the other data supporting the findings of this study are available within the article and its supplementary information files and from the corresponding author upon request.

Source data for Fig. 1a, c and 3a–c and 4a, c, e and 5b and 7a–c and 8a, c, d, e and 10a–e and Supplementary Figs. 1a and 11c–e are available as a source data file. A reporting summary for this article is available as a Supplementary Information file.

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

## Acknowledgements

The excellent technical assistance of S. Weidemann, V. Galinat, E. Müller-Hermelink, A. Odon, R. Nordin, S. Riel and T. Schneider, C. Grimmel is gratefully acknowledged. The authors thank H.G. Rammensee, O. Rieß, K. Ghoreschi, J. Brück, B. Bauer, M. Rentschler for helpful discussions, and T. Haug for technical support in the chromium release assay. Furthermore, the authors thank M. Hagemann, F. Liebel, J. Teutsch and M. Möschter for help in animal experiments. S. Kaesler for providing B16-OVA cells. This work is part of the doctoral thesis of E.B., B.F.S., F.A., T.R., F.J.H., G.D. and N.S. Wilhelm Sander-Stiftung (2012.056.3), Deutsche Krebshilfe (application numbers 109037, 110662, 110664, 70112332 and 70112337), Werner Reichenbach Stiftung and the Deutsche Forschungsgemeinschaft (SFB 685, SFB TRR 156/2, RO 764/14-1; RO 764/15-1, SFB 773, Wi 1279/4-1). Cluster of Excellence iFIT (EXC 2180) "Image-Guided and Functionally Instructed Tumor Therapies", University of Tübingen, Germany, funded by the Deutsche Forschungsgemeinschaft (DFG, German Research Foundation) under Germany's Excellence Strategy - EXC 2180 – 390900677.

## Author contributions

M.R. developed the concept, together with E.B., T.W., and with R.M., F.A., T.R. (Fig. 8) and M.K., B.F.S. (Fig. 4) M.R. planned and designed the experiments. E.B. performed or experimentally supported most experiments and analysed most data with the help of T.W. or B.F. (except Fig. 8a, c, d, e; Fig. 10a–c; Supplementary Fig. 7b and Supplementary Fig. 13a); B.F.S. and D.So. performed the experiments Fig. 4a–c. F.A., T.R., N.H., and A. G. performed the experiments Fig. 8a, d, e. F.J.H., C.S., and G.D. did the genomic analysis of melanoma patients, initiated by S.B. and D.D. B.F. and M.Scha. established and carried out fluorescence and electron microscopy. A.F. and T.E. recruited patients and provided patient material. H.N. and T.S. isolated primary melanoma cells. N.S. and J.B. performed and interpreted comparative genomic hybridisation arrays. L.Z., D.Dau. and S.Z. generated gRNAs against Cdkn2a for CRISPR gene-editing technology and supervised the knock-out experiments. H.B. did the chromium release assay, L.Q-M. the immunohistochemistry, K.S.B. provided cell lines and data for reviewers. B.F.S. and B.J.P. established and carried out magnetic resonance imaging. M.E. did the statistical analyses (Fig. 1a, 4a and c). K.J.J. and A.B. performed two-dimensional light sheet fluorescence microscopy. E.B., T.W., L.Z., B.P., D.Sch., R.M., M.K. and M.R. discussed and interpreted the data. M.R. wrote the paper. All authors discussed the results and commented on the manuscript, and agreed with its content.

## Competing interests

M.R., E.B. and T.W. are inventors of an European pending patent Number 13 826 993.1: "Use of active substance combinations for tumour senescence" and an United States patent Number US 10,046,029 B2: "Method of inducing senescence in tumour cells by administrating TNF-A in combination with IFN-A of IFN-Y". As chairperson M.R. has to sign all contracts of the department. M.R. receives grants for research projects or travel support and has the indicated shares. The authors B.P and M.K have research contracts with Imaging modality suppliers, and are members of the "ImmuneImage" consortium within the Horizon2020 IMI programme of the European Union. F.J.H reports institutional research support from Novartis. C.S. reports research support from Illumina and institutional research support from Novartis. All remaining authors declare no competing interests.
