## [Peer Review File · Nature Communications]

Reviewers' comments:

Reviewer #1 (Remarks to the Author):

This is a timely article providing novel highly relevant information to understand the relationship between interferon signaling, cellular senescence and response or resistance to immune checkpoint blockade therapy. Together with the series of recent articles reporting on the immune effects of CDK4/6 inhibitor therapy and the synergy with immune checkpoint blockade therapy, they provide new understanding on the biology of cell cycle control, cellular senescence and immunotherapy. The new article builds upon prior work from this group reporting on the induction of cellular senescence with interferon gamma and TNF α . This article is well developed, with elegant hypothesis testing and compelling results based on relevant mouse and cellular modeling.

Major comments:

Why did the authors change the mouse model when testing the role of p21?

It would have been desirable to perform the analysis of human biopsy samples from patients treated with anti-PD-1 therapy using whole exome sequencing of tumors compared to normal genomic DNA to better analyze nonsynonymous mutations and other genomic alterations.

Minor comments:

The authors use terms as "responder patient had significantly higher tumor mutational burden" to refer to analyses done in biopsies of patients who responded to therapy. It would be desirable to refer to cancer biopsies and not patients when referring to analyses done in cancer biopsies, and make the patient the noun of the sentence as opposed to an adjective to response to therapy.

Reviewer #2 (Remarks to the Author):

Brenner-E,... ..Röcken-M, Cancer immune control needs intratumoral senescence-inducing p21CIP1 and p16INK4a pathways

Submitted to Nature Communications

In this manuscript, Röcken and colleagues link cell-based anti-cancer immunity – beyond its well-established cytotoxic mode of action – to induction of cellular senescence, and demonstrate, in turn, that defective target cell senescence (e.g. by deletion of the senescence mediators p16INK4a or p21CIP1) impairs natural or immune checkpoint blockade (ICB)-exerted tumor control.

This is an interesting manuscript on a timely, not to say hot topic. However, central, essential components of the experimental setup remain entirely unaddressed and put the investigation at risk to come to fundamentally flawed conclusions. With these points clarified experimentally (which is a work-intensive endeavor, I'm aware), the story might very well be suitable for publication in Nature Communications. Please see detailed comments below.

Major concerns and comments

1. What is the expression status of PD-L1 and PD-L2 on non-rejected Stat1-proficient vs. Stat1-deficient RT2 cancers that emerged after transplantation? Where is the evidence that Stat1-

deficient RT2 tumors protect themselves from cytotoxicity via a PD-L1 inhibition-sensitive immune checkpoint blockade (ICB) principle (the specific kind of ICB applied is not even mentioned in the main text or legend)? If the anti-PD-L1 antibody had no target principle to interact with on Stat1-deficient RT2 cells, no cell-based immunity would be re-established, and no impact on tumor growth or senescence should be expected. This is indeed likely to be the case, since the authors apparently did not observe quantitative, senescence-independent cytolysis of the Stat1-deficient RT2 tumors following exposure to ICB.

2. The authors should provide evidence that the ICB treatment actually works via a cell-based immune attack.

3. Unfortunately, the authors did not manage to include p16INK4a-proficient CRISPR control RT2 cancers to compare those as "matched pairs" to their CRISPR-p16INK4a-deleted counterparts. Hence, it remains somewhat questionable whether these p16INK4a-loss derivatives (either spontaneously or CRISPR-mediated p16INK4a-deficient) only differ in their p16INK4a expression status – or reflect further mutated/altered offspring of the parental tumor cells that may, for instance, now critically differ in their expression profile of immune checkpoint mediators beyond PD-L1.

4. In the context of Fig. 2, the authors refer to an adoptive T-cell therapy against endogenous tumors in RT2 mice: do these T antigen-specific CD4+ TH1 cells possess direct cytotoxic activity? Are they clonal, expanded as a cell line, or how were they raised?

5. What is the reason for incomplete eradication of the tumor load? Is it clonal heterogeneity of the target cells regarding the target antigen, i.e. the Large T antigen in the adoptive T-cell experiment (if the T-cells were raised against Large T and clonally expanded), with some cells escaping by antigen loss (and presumable gain of genetic lesions that substitute for the Large T oncogene)? If there is antigen loss, why should the antigen-specific T-cells still interact with the target cells (as a prerequisite to induce, at least, senescence)?

What is the cellular mechanism that redirects a cell-mediated cytotoxic principle (is it perforin/granzyme B? Or FasL?) in some cells now towards a senescence inducer?

Or, is it signaling heterogeneity on the target cell side, leading to an inability to die (how is death via cytotoxic granules actually prevented?); hence, allowing senescence to occur as a secondary, delayed onset backup effector mechanism?

Or, is it a stochastic process, where senescence might be the T-cell-induced target cell state by chance, hence competing with apoptosis. If so, senescence would potentially work as a resistance mechanism towards apoptotic cell death, and, vice versa, target cells with a senescent defect should rather be prone to complete cytolysis.

Of note, in subsequent experiments using adoptive T-cell transfer in RT2.Stat1^{-/-} mice, it would be important to demonstrate that tumors arising in this genetic context are not (even more?) prone to Large T antigen loss variant subclones.

These critical questions remain to be addressed here!

6. What is the long-term fate of T-cell-related senescence induction in RT2 target cells? Do they persist for extended periods of time in situ, do they occasionally re-enter the cell-cycle, do they promote growth of neither killed, nor senescent neighbor cancer cells via the senescence-associated secretome, or do they even turn into cancer stem cells, as recently reported? Or, is there secondary clearance by innate immune cells?

7. When viewed as single factors, p16INK4a alone or p21CIP1 alone are, in many cellular contexts,

not known to operate as senescence-essential mediators; hence, their single-gene inactivation should not compromise senescence. This might be different if deletions at the INK4a/ARF locus ablate both p16INK4a and p19ARF.

This notion is important with respect to the presented absence of senescence signs in myc-driven lymphomas lacking p21CIP1 (Fig. 3B) – where it remains unclear whether there actually happened a T-cell attack that failed to induce senescence due to target cell inability.

Minor concerns

1. How many independent primary RT2 tumors were used, e.g. in Fig. 1 and subsequent experiments? Some plots present an overlay of individual tumor curves over time, others represent either only one sample or lack error bars. Immunofluorescence photomicrographs largely lack quantification throughout the figures, and if indicated, numbers/percentages lack standard deviation (e.g. Fig. S1C).
2. According to the labelling, Fig. S1C presents RT2 cells that lack p16INK4a due to CRISPR editing, but the main text confusingly refers to spontaneous p16INK4a-deleted tumors selected for in vitro and in vivo.
3. Those Fig. S1C p16INK4a-deficient RT2 tumors seem to respond to ICB with H3K9me3 reactivity, similar to p16INK4a-proficient tumors – which is different from the interpretation stated in the text.
4. What do the authors mean by “electron microscopy confirmed the nuclear accumulation of SA-beta-gal in the senescent tumor cells”? SA-beta-gal is an enzymatic assay, which does not work anymore after fixation with glutaraldehyde. Moreover, SA-beta-gal staining is lysosomal, not nuclear (it is not clear to me what Fig. S5E – in the cytoplasm, not the nucleus – actually detected). And: what was the idea behind the ultrastructural analysis in addition to conventional enzymatic SA-beta-gal staining?
5. Fig. 3E is a comparison of ICB vs. control/mock treatment, and does, unlike stated in the text, not show that “these mice [myc.p21CIP1-/-] developed lymphomas with slower dynamics than Myc controls”. If this would be the case, it would be an unexpected observation, and different from the published literature (at least with respect to the Eu-myc transgenic line ± CIP1). Again, what is the mechanistic basis for an ICB therapy (here: anti-PD1 plus anti-CTLA-A4) to unleash blunted cellular immunity against these endogenous myc-driven lymphomas? What are the immune checkpoint ligands expressed on the lymphoma cells conferring susceptibility to ICB here, in this model?
6. The melanoma data are interesting, but, again, cause and consequence are not necessarily clear. It is no surprise, as repeatedly reported in the literature, that manifest melanomas possess senescence-inactivating gene lesions, either to bypass or to escape Braf- (and, to some extent, Ras-)driven senescence. Equipped with those lesions (e.g. INK4a/ARF loss), these tumors not only can't senesce anymore, they will also present with a variety of more aggressive growth properties (due to shutdown of both the Rb and the p53 axis). Whether ICB is less efficient due to a blunted cell-related senescence response, as claimed here, or simply due to a reduced susceptibility to die to any pro-apoptotic trigger, or, alternatively or in combination, to more aggressively proliferate, remains to be shown.

Reviewer #3 (Remarks to the Author):

The manuscript from Brenner et al. describes an interesting role for cellular senescence in favoring cancer eradication in response to immune-checkpoint blockade (ICB). By employing a pancreatic cancer cells murine model, the authors report that a senescent program, characterized by SA β -Gal, p16 and SAHF induction, is activated in response to ICB and in a STAT1-dependent manner. Similarly, p21-associated senescence was required for the immune control of murine lymphomas. Likewise, melanoma patients that did not respond to ICB treatments, displayed a selective loss or inactivating mutations of senescence associated genes further supporting the role of senescence in mediating a stronger cancer immune control.

The study is well done and informative and will be of real value to researchers interested in senescence and cancer immunotherapy for both solid and hematologic malignancies. The writing is clear. The statistical analysis is solid and supports the authors' conclusions.

I do suggest a few experiments and some controls that would be nice to have to make the connection between senescence and immune crosstalk even stronger.

Major Points:

-in Figure 1C,D: the authors performed in vitro treatment of RT2 WT or p16 null cells with IFN γ and TNF. Both cells underwent cell cycle arrest during treatment, however, p16 null cells escaped from senescence and kept proliferating. The authors should complement the analysis of cell number with SA- β -Gal assays to verify the establishment of senescence immediately after treatment in WT or p16 null (as well as induction of p21 and possibly DDR activation).

-in Figure 1F: I noticed some variability in the control treated group between animals with RT2-CRISPR-p16 cells. What is the explanation for this variability? Did the authors verify the loss of p16 in these samples by surveyor/NHEJ molecular assays? The authors should also rule out the possibility that their CRISPR-mediated KO does not impact on the expression levels of other genes in the INK4a locus (e.g. p19/ARF?).

-Figure 2/Suppl. Figure 2: is there DDR activation in tumor cells induced into senescence in response to ICB or ICB/AT treatment? IHC stainings for γ H2AX and pATM would be an important addition to the manuscript.

Figure 2/3: In the RT2 model the authors should complement the analysis on senescence in tissue section with an in depth characterization of immune-infiltrates. Cellular stainings for cytotoxic T cells, Tregs, NK and macrophages will be very informative in all treated groups.

Figure 3d,e: there is discrepancy in the two graphs regarding the number of alive mice at 200 days post treatment (0 in panel d; 1 in panel e)

General notes:

- I understand if there is a hard limit of characters for this manuscript format, but I feel that the introduction is very short. I would also recommend the authors to expand on the discussion section. The authors should have in mind the very recent discovery by the lab of C. Schmitt on the possible reprogramming of cancer cells induced into senescence by antineoplastic treatments (Milanovic et al. Nature 2018; Lee et al. Nature Cell Biology 2019). Of note, while the authors focused on the cell-autonomous aspects of cellular senescence, they cannot ignore that activation of SASP from senescent cells may be reinforcing senescence in the short-term but contribute via chronic inflammation to cancer relapse and senescent escape in the long-term (Demaria et al. Cancer Discovery, 2017). The likelihood of this phenomenon is particularly high in elderly subjects where a compromised immune system may delay senescent cells eradication.

Reviewer #4 (Remarks to the Author):

The manuscript by Brenner et al., describes how a p16 and p21-dependent cancer cell senescence program may promote checkpoint blockade (CB) and adoptive cell transfer (ACT) mediated immune regulation in syngeneic solid and liquid tumors. The authors suggest that cellular senescence, driven by cell-cycle regulators, is a fundamental mechanism by which T cells prevent cancer-cell escape from immunosurveillance. In addition, they provide correlative evidence that induction of cancer-cell senescence, through activation of p16 and p21, is dependent on IFN γ -STAT1 paracrine signaling from immune cell IFN γ . While these findings are intriguing, they are largely correlative in nature and lack a mechanistic basis. Additional biochemical, in vitro, and in vivo experiments are necessary to substantiate the authors claims. Furthermore, the authors fail to provide convincing evidence that IFN γ -dependent activation of STAT1 in cancer cells is responsible for the observed senescence phenotype.

In its current state, the data is too premature for publication. The following major points should be considered.

1. The mechanism associated with tumor rejection in mice receiving immune checkpoint blockade (ICB) is unclear. The authors indicate that CD8 depletion is necessary to prevent RT2 tumor rejection in wildtype treatment naïve mice. Data illustrating the efficacy of depletion and the presence or absence of CD8 T cells at the time of ICB is critical for understanding the significance of the findings in Figure 1. If CD8 T cells have been effectively depleted what is the proposed mechanism of tumor rejection upon ICB? If CD4-dependent mechanisms are proposed, data illustrating altered Treg or Th1 biology associated with rejection need to be shown. Furthermore, ICB works in T cell depleted mice in Figure 1.a, but not in Figure 3.d. The authors need to address the question of the proposed mechanism for ICB efficacy upon T cell depletion in Figure 1.a, that is not occurring in Figure 3.d.
2. The rationale for the different immunotherapies used and their treatment regimens applied in the different experiments is unclear.
3. The lack of statistical comparison between the tumor volumes illustrated in the spider plots in Figure 1 confuse data interpretation. Group differences that are significant should be illustrated, and tumor sizes should be represented as volume. While the individual tumor growth curves illustrate group variance and highlight outliers, growth curves with means and standard error of mean, and accompanying significance indication, should be provided to allow for direct comparison of tumor sizes between groups. Mean data from the different groups should be presented on a single plot illustrating tumor volumes. Additionally, it appears that p16 loss is much more aggressive than STAT1 loss in these cancer cells (Figure 1.b,e), suggesting alternative mechanism than the one proposed by the authors. However, it is difficult to discern as no statistics or direct comparison can be made with the plots provided.
4. For Figure 1.b, the lack of p16 induction in a STAT1 dependent manner needs to be validated with biochemical evidence. The authors should at least show quantification of IF. In addition, qPCR for p16 transcript should be shown. Furthermore, β 2M is also a STAT1 target gene that is critical for immune response post ICB. The lack of ICB in Stat1 $^{-/-}$ RT2 cells could be a result of β 2M loss in these cells. STAT1-dependent efficacy of immune responses to solid tumors is not novel. The authors need to clearly demonstrate that STAT1 activation in these cancer cells results in the transcription of p16, and that this is required for sensitization to ICB.
5. The authors claim CRISPR-Cas9 p16-deficient cancer cell model is similar to the p16-deficient cells derived from tumors. However, based on the data in Figure S1 there are clear differences between these two model systems. Additionally, the methods for the establishment of the CRISPR

cell line needs substantial elaboration.

6. For Figure 1 and 4, the lack of cytotoxicity upon TNF and IFN γ treatment needs to be shown. Cell number is not an adequate measure of cytostatic effect of treatment. It is further necessary to provide Annexin V staining for the cells after treatment and before the "wash" step.

7. Data in Figure 3 is largely phenomenological, with no clear indication of mechanism. IFN γ has effects on a variety of cell types including myeloid and stromal cells within the TME. The lack of p16 induction in anti-IFN γ treated mice is correlative. The authors need to provide direct evidence that this is functioning in a manner dependent on IFNR signaling in cancer cells. It needs to be clarified if anti-IFN γ treatment has the same effect in IFNR $^{-/-}$ RT2 cells.

8. In vitro and biochemical validation are needed to substantiate the claims that the effect of ICB observed is at least partially explained by IFNR-dependent activation of STAT1 in cancer cells that leads to p16 and/or p21 activation.

Answers to the comments of the reviewers:

We thank the reviewers for their comments that will help us to improve our manuscript. In the text, all changes are labelled in red. In the responses to the reviewers we indicate the side and the paragraph, or the figures, where their points have been addressed.

In the answer to the reviewers we underlined the part where we indicate the page and paragraph or figure with the experiments or comments, performed in response to the reviewer.

Reviewer #1

Major comments:

1. Question

Why did the authors change the mouse model when testing the role of p21?

Answer

This had three reasons:

To determine whether the data we obtained with immune checkpoint blockade (ICB) were restricted to the missing p16^{INK4a}/p19^{Arf} in RIP-Tag tumours or whether they were of more general validity, we had to analyse whether ICB requires cytokine-induced senescence (CIS) also the second key-signalling pathway for senescence induction, the p21^{Cip1}- signalling pathway. If so, the data would be of broader validity. Moreover, if shown in a second, entirely different tumour, this would broaden the general validity of the experiments (page 8, 3rd paragraph).

Importantly, the p53 signalling is altered in RT2-tumours (Casanovas O et al. Oncogene 2005). Therefore we could not properly analyse the role of p21^{Cip1} in RT2-cancers and needed the λ -MYC mice to carefully analyse its role in ICB induced tumour cell senescence and therapy (page 8, 3rd paragraph).

Moreover, the λ -MYC mice allowed us to investigate the role of CIS and p21^{Cip1} in endogenously developing tumours without the need of cancer cell transplantation (page 8, 3rd paragraph).

2. Question:

It would have been desirable to perform the analysis of human biopsy samples from patients treated with anti-PD-1 therapy using whole exome sequencing of tumors compared to normal genomic DNA to better analyze nonsynonymous mutations and other genomic alterations.

Answers

The answer comprises two independent aspects:

A. Comparing the tumour genome with normal genomic DNA.

Here we have to apologize that we were not precise enough in the methods. For each patient we performed paired panel sequencing of the tumour DNA and normal DNA to identify tumour-specific alterations (somatic single nucleotide variants and somatic copy

number alterations). This is now clearly described in the main text (page 9, 4th paragraph till page 10, 1st paragraph).

B. The reviewer suggests using whole exome sequencing to better analyse non-synonymous mutations and other genomic alterations.

We fully agree with the comment of the reviewer, but the goal of the experiment was not to identify new, senescence associated mutations, but to ask whether mutations in the pathway of those interferon-regulated genes that had been identified as relevant for successful immune checkpoint blockade (ICB) in mice would also affect in humans the response of melanoma metastases to ICB.

Here, we used a panel targeting the coding and flanking (+/-20 bases) region of up to 693 cancer relevant genes that are well defined as genes relevant for the process of carcinogenesis, cancer treatment and prognosis (PMID: 29592813). From these 693 genes (Supplementary table 1-3), we explicitly selected and analysed only those 19 genes that encode for molecules that are also involved in the IFN- γ -mediated activation of the senescence signalling pathways that we identified in our murine experiments.

To ask whether genes of the senescence-inducing signalling pathways are more frequently mutated in metastases not responding to ICB than in melanoma metastases responding to ICB. We explicitly analysed somatic SNVs and CNVs leading to a complete loss or strong activation. We selected only those mutations, where a complete loss or a strong amplification could alter the gene function significantly. In our approach, we may have missed the very rare fusions/translocations of the genes investigated. However, it may have been difficult to identify those also with the exome-based approach. The panel approach with 693 cancer-associated genes allowed us a deep sequencing necessary to identify tumour-specific alterations in less than 10% of all cells.

Thus the goal of this approach was not to identify new genes, but to determine whether the genetic aberrations in signalling pathways that we identified as important for defined functional defects in the CIS were also associated with a poor response to ICB in humans.

The data indeed showed that mutations only in genes involved in IFN- γ -mediated activation of the senescence signalling pathways were significantly associated with a fast progression of the metastases during ICB (Fig. 10b, c). SNVs and CNVs did not differ between the metastases of the two patient groups. Moreover, functional analyses of selected melanoma cell lines that we could raise from such metastases confirmed that only the rapidly progressing metastases were resistant to CIS.

We addressed these important points on page 10, 1st and 2nd paragraph, and in the figure legend 10.

Minor comments

3. Question:

The authors use terms as “responder patient had significantly higher tumor mutational burden” to refer to analyses done in biopsies of patients who responded to therapy. It would be desirable to refer to cancer biopsies and not patients when referring to analyses done in cancer biopsies, and make the patient the noun of the sentence as opposed to an adjective to response to therapy.

Answer

We fully agree with the reviewer – and changed it into: Biopsy material from metastases (or melanoma metastases) from (non-) responder patients in the manuscript and the figure legends.

We corrected this on page 9, 3rd paragraph, on the entire page 10 and in the legends of Fig. 10 and Supplementary Figures 12, 13.

Reviewer #2

Major concerns and comments

1. Question

What is the expression status of PD-L1 and PD-L2 on non-rejected Stat1-proficient vs. Stat1-deficient RT2 cancers that emerged after transplantation? Where is the evidence that Stat1-deficient RT2 tumors protect themselves from cytotoxicity via a PD-L1 inhibition-sensitive immune checkpoint blockade (ICB) principle (the specific kind of ICB applied is not even mentioned in the main text or legend)? If the anti-PD-L1 antibody had no target principle to interact with on Stat1-deficient RT2 cells, no cell-based immunity would be re-established, and no impact on tumor growth or senescence should be expected. This is indeed likely to be the case, since the authors apparently did not observe quantitative, senescence-independent cytolysis of the Stat1-deficient RT2 tumors following exposure to ICB.

Answer

In the new figures (Supplementary Figure 2a, b and new Supplementary Figure 7d) we now show the expression of PD-L1 by tumours and the tumour microenvironment of STAT1^{+/+} and on STAT1^{-/-} RT2 cancer cells. We did both, immune histology and FACS analysis (New Supplementary Figure 2a, b and new Supplementary Figure 7d) of PD-L1. As expected, IFN- γ further increased PD-L1 expression only in STAT1^{+/+} cells (New Supplementary Figure 2b). The anti-PD-L1 mAb thus had a clear target in all experiments. In STAT1-deficient mice it was detectable in mice that received adoptive T cell transfer (those cells are STAT1^{+/+}).

The analyses revealed that > 10% of the cells analysed expressed PD-L1. In a previous study, melanomas with 5 % of the tissue expressing PD-L1 were considered as PD-L1 positive (Postow MA et al. *New Engl J Med* 2015). This was addressed in the text page 4, 1st paragraph, page 7, 4th paragraph.

We focused on PD-L1 as the role of PD-L2 is still questionable and studies on the role of PD-L2 expression and the therapeutic effects of anti-PD-L2 mAb are still inconclusive (Tanegashima T et al. *Clin Cancer Res* 2019).

2. Question

The authors should provide evidence that the ICB treatment actually works via a cell-based immune attack.

Answer

In Fig. 8d (new; old Fig. 3d) we showed that the therapy was strictly depended on CD3⁺ T cells, as deletion of all T cells with a pan- anti-T cell mAb completely abrogated the therapeutic effect of ICB.

3. Question

Unfortunately, the authors did not manage to include p16^{INK4a}-proficient CRISPR control RT2 cancers to compare those as “matched pairs” to their CRISPR-p16^{INK4a}-deleted counterparts. Hence, it remains somewhat questionable whether these p16^{INK4a}-loss derivatives (either spontaneously or CRISPR-mediated p16^{INK4a}-deficient) only differ in their p16^{INK4a} expression status – or reflect further mutated/alterd offspring of the parental tumor cells that may, for instance, now critically differ in their expression profile of immune checkpoint mediators beyond PD-L1.

Answer

We included p16^{INK4a}/p19^{Arf}-proficient CRISPR control RT2 cancers. We analysed these tumours and show that they were susceptible to CIS (Fig. 3c). In vivo 100% of the transplanted tumours were rejected even in the CD8-depleted mice (Supplementary Figure 3c). In contrast, 80% of the p16^{INK4a}/p19^{Arf} deficient RT2 cancers did grow.

A. The p16^{INK4a}/p19^{Arf}-proficient CRISPR control is obligatory, in order to *exclude that the p16^{INK4a}/p19^{Arf} knock-out and not the CRISPR construct itself or the cancer cell transfection cause the unresponsiveness to ICB and CIS*. This was the answer the experiment should provide.

B. As even only 80% of the p16^{INK4a}/p19^{Arf} knock-out tumour cells grew in vivo (Supplementary Figure 3c), the data underline a) that the CRISPR construct was highly immunogenic and b) that those tumour cells that were not completely deleted required p16^{INK4a}/p19^{Arf} to be controlled.

C. We performed comparative genomic hybridization (CGH) of the parental tumours, the p16^{INK4a}/p19^{Arf}-proficient CRISPR control RT2 cancers and the either spontaneously or CRISPR-mediated p16^{INK4a}/p19^{Arf}-deficient RT2 cancers. The analyses of all seven p16-deficient tumour cell lines (loss variants by selection or by CRISPR Cas9) showed that all had a slightly different pattern. The loss of the p16^{INK4a}/p19^{Arf} locus was the only CGH motif, common to all p16^{INK4a}/p19^{Arf} deficient RT2 cancer lines. This was the only locus where all p16^{INK4a}/p19^{Arf} deficient RT2 cancer lines differed from the control cancer lines. We now show in addition the CGH of the CRISPR transfected cells (New Supplementary Figure 3a).

D. The concerns that the transfection of a cell line might also affect a unique unknown gene that is not affected in the control transfected cells can not be excluded. Even if the CRISPR Cas9 control would have grown in vivo and responded to ICB – such a possibility would not have been excluded.

To address this problem, we analysed in addition six different newly generated CDKN2a deficient RT2 cancer lines and the CRISPR-mediated p16^{INK4a}/p19^{Arf}-deficient RT2 cancers. All seven CDKN2a deficient RT2 cancer lines provided the same response pattern. It is very unlikely that all seven CDKN2a deficient RT2 cancer lines failed to respond to ICB because of a common mutated/alterd gene that was different from the parental tumour cell lines. We addressed this on page 5, 2nd paragraph.

4. Question

In the context of Fig. 2, the authors refer to an adoptive T-cell therapy against endogenous tumors in RT2 mice: do these T antigen-specific CD4⁺ TH1 cells possess direct cytotoxic activity? Are they clonal, expanded as a cell line, or how were they raised?

Answer

The responses to these questions were carefully elaborated in previous publications (Muller-Hermelink N et al. Cancer Cell 2008, Braumuller H. et al. Nature 2013). Therefore, here we clarify the procedure in the figure legend and cite the sites where the results were described in the manuscripts above.

We had carefully analysed the cytotoxic capacity of these T antigen-specific TH1 cells. We exposed RT2-cancer cells to these TH1 cells and found no detectable cytotoxic activity (as compared to CTL). This is shown in Braumuller H et al. Nature 2013 (Suppl. Fig. 3c, d). [red acted].

The Tag-specific TH1 cells were cell lines, derived from TCR-transgenic mice freshly generated for each adoptive transfer from lymph nodes and spleen. Briefly, freshly isolated CD4 T cells from TCR transgenic mice were stimulated with Tag peptide, anti-IL-4 and IL-2 and then expanded with IL-2 to generate Tag specific TH1 cells. The cell lines were periodically tested and we always generated IFN- γ ⁺, TNF⁺ and IL-4⁻ Tag specific TH1 cells. The method was now clarified in the legend to Fig. 4, where we also gave the reference for

the generation of the Tag-specific TH1 cells.

5. Question

What is the reason for incomplete eradication of the tumor load? Is it clonal heterogeneity of the target cells regarding the target antigen, i.e. the Large T antigen in the adoptive T-cell experiment (if the T-cells were raised against Large T and clonally expanded), with some cells escaping by antigen loss (and presumable gain of genetic lesions that substitute for the Large T oncogene)? If there is antigen loss, why should the antigen-specific T-cells still interact with the target cells (as a prerequisite to induce, at least, senescence)?

Answer

We analysed antigen expression as suggested by the reviewer.

The incomplete eradication of the RT2-cancers was not due to the proposed classic tumour evasion phenomena like clonal heterogeneity of the target cells regarding the target antigen. At the end of the experiment, the tumour cells of either treated or untreated mice had similar levels of Tag mRNA expression and were strongly positive for Tag protein, the target antigen (New Fig. 5b, c).

We addressed this in the text on page 6, 2nd paragraph.

6. Question

What is the cellular mechanism that redirects a cell-mediated cytotoxic principle (is it perforin/granzyme B? Or FasL?) in some cells now towards a senescence inducer?

Or, is it signaling heterogeneity on the target cell side, leading to an inability to die (how is death via cytotoxic granules actually prevented?); hence, allowing senescence to occur as a secondary, delayed onset backup effector mechanism?

Or, is it a stochastic process, where senescence might be the T-cell-induced target cell state by

chance, hence competing with apoptosis. If so, senescence would potentially work as a resistance mechanism towards apoptotic cell death, and, vice versa, target cells with a senescent defect should rather be prone to complete cytolysis.

Answer

In Braumüller et al. 2013, we showed that CD8⁺ cytotoxic T cells (CTL) lyse cells by a cell-mediated toxic principle, while CD4⁺ Tag-T_H1 cells do not [redacted].

We therefore excluded cytotoxic events of CD4⁺ Tag-T_H1 cells mediated by granzyme (see also answer to Question 4).

The data by others (Chang J et al. Nat Med 2016) and preliminary data from us strongly suggest that the third hypothesis of the reviewer should be the correct one. It is known that senescent cancer cells have a defect in apoptosis induction. The Campisi group has shown that this is due to increased levels of Bcl-2 (Chang J et al. Nat Med 2016). [redacted]

Importantly, stimulation of cancer cells with TNF and IFN- γ frequently induces apoptosis (Kroemer G et al. Annu Rev Immunol 2013). Unpublished data with tumour cells confirm this: we found that at higher concentrations TNF and IFN- γ rather induce apoptosis and at lower concentrations senescence.

In the context of the manuscript, the following observation is important: In some cancers TNF and IFN- γ induce only apoptosis and the surviving cancer cells restart proliferation, sometimes very rapidly. In other cancers TNF and IFN- γ induce apoptosis and senescence, and the surviving cancer cells remain growth arrested because of CIS. Senescence can also be induced in cancer cells that have a defect in apoptosis induction (Braumüller H et al. Nature 2013; Rentschler M et al Cell Physiol Biochem 2018).

As these data are already published, we address this important aspect in the discussion

section, page 12, 3rd paragraph.

We hope that the reviewer accepts that we show the pro- and anti-apoptotic effects of CIS only as confidential data to the reviewer only. This is an entirely new and complex research topic that requires large analyses. They cannot be presented in the context of this focus as we feel that they would deviate from the message of the manuscript.

7. Question

Of note, in subsequent experiments using adoptive T-cell transfer in RT2.Stat1^{-/-} mice, it would be important to demonstrate that tumors arising in this genetic context are not (even more?) prone to Large T antigen loss variant subclones.

Answer

In the new Fig. 5b, c we showed that the tumours growing in RT2.Stat1^{-/-} mice normally express both, Tag mRNA and protein. Therefore the tumours growing in Stat1^{-/-} mice did not

result from Large T antigen loss variant subclones. We addressed this also in the text on page 6, 2nd paragraph.

Reviewer remark:

These critical questions remain to be addressed here!

Answer

We hope that we have adequately addressed each point with new data.

8. Question

What is the long-term fate of T-cell-related senescence induction in RT2 target cells? Do they persist for extended periods of time in situ, do they occasionally re-enter the cell-cycle, do they promote growth of neither killed, nor senescent neighbor cancer cells via the senescence-associated secretome, or do they even turn into cancer stem cells, as recently reported? Or, is there secondary clearance by innate immune cells?

Answer

We addressed this question by further analysing the phenotype of RT2-cancer cells of the growth arrested, senescent cancers from RT2 mice, where we treated established RT2 cancers with ICB and T_H1 cells, and then isolated and cultured the RT2-cancer cells (New Fig. 7a, b). When isolated *ex vivo* and then cultured, *exclusively the senescent RT2-cancer cells failed to re-enter* cell cycle when cultured for prolonged periods of time. When analysed after ≥ 5 passages only these growth arrested RT2-cancer cells also expressed SA- β -Gal (New Fig. 7c).

The further growth behaviour of such growth arrested RT2-cancer cells has extensively been addressed in a previous manuscript (Braumuller, Wieder et al. Nature 2013). To ask whether such explanted and cultured senescent RT2-cancer cells remain also senescent *in vivo*, we retransferred the living but senescent cancer cells into T, B, and NK cell deficient NSG mice. We showed that such long term cultured senescent cancer cells remained living but stably growth arrested for another 7 weeks in those NSG mice (Braumuller H et al. Nature 2013, Fig. 4).

[redacted]

As these senescent cancer cell populations do neither grow for 6 passages *ex vivo* (New Fig. 7a), nor for seven weeks *in vivo* (New Fig. 7a; Braumuller H et al. Nature 2013, Fig. 4), and as we found no increase in genes associated with the cancer initiating phenotype, there is no evidence neither for cancer initiating cells nor for cancer promotion by the SASP of senescent RT2-cancer cells after CIS.

The key data were summarized in the new Fig. 7a-c in the text page 7, 2nd paragraph and 5th paragraph, and in the discussion page 11, 3rd paragraph. The question concerning the stemness

was introduced into the discussion (page 11, 3rd paragraph).

9. Question

When viewed as single factors, p16INK4a alone or p21CIP1 alone are, in many cellular contexts, not known to operate as senescence-essential mediators; hence, their single-gene inactivation should not compromise senescence. This might be different if deletions at the INK4a/ARF locus ablate both p16INK4a and p19ARF.

Answer

Here we have to apologize that we were not precise enough in our material and methods section: we did simultaneously delete p16^{INK4a} and p19^{ARF}. We now corrected this by naming the construct CRISPR-p16^{INK4a/-}-p19^{ARF/-} throughout the text (beginning with page 5, 1st paragraph) and the figure legends (Fig. 2, 3, Supplementary Figure 2, 3, 5). In the original version we only referred to the construct by a reference, where we had previously published this construct (reference 33). We agree that it is clearer to describe it again. We also introduced this into the material and methods section (page 32, 4th paragraph).

10. Question

This notion is important with respect to the presented absence of senescence signs in myc-driven lymphomas lacking p21^{CIP1} (Fig. 3B) – where it remains unclear whether there actually happened a T-cell attack that failed to induce senescence due to target cell inability.

Answer

In the λ MYC-driven tumours we demonstrated that the therapy was strictly dependent on T cells as the deletion of all T cells with a pan anti-T cell mAb abrogated both, senescence induction, expression of p21^{CIP1} and the therapeutic efficacy of the immune therapy (New Fig. 8d; in the text end of page 8).

Minor concerns

11. Question

How many independent primary RT2 tumors were used, e.g. in Fig. 1 and subsequent experiments? Some plots present an overlay of individual tumor curves over time, others represent either only one sample or lack error bars. Immunofluorescence photomicrographs largely lack quantification throughout the figures, and if indicated, numbers/percentages lack standard deviation (e.g. Fig. S1C). ??

Answer

A. In the new Fig. 1, we addressed this question giving each cell line a different lining (RT2: 3 different cell lines; 19 tumour bearing mice. RT2.Stat1^{-/-}, 2 different cell lines; 14 tumour bearing mice. p16^{Ink4a/-} 2 different cell lines; 26 tumour bearing mice, CRISPR-p16^{Ink4a/-}-p19^{Arf/-} 1 cell line; 11 tumour bearing mice). The in the new Fig. 3, we performed *in vitro* CIS treatment on 6 different cell lines as shown in the Figure.

B. Figure 1 was changed as proposed by the reviewer. We now calculated the tumour volumes and provided the statistics. (Please note that we give a summary curve with the slope of the tumour growth inside the spider plot, as instructed by the statistics department; ANCOVA).

We clarified this in the legend to Fig. 1a

The statistics for (Fig. 1a, c, Fig. 3a-c, Fig. 4a, c, e, Fig. 7c, Fig. 8a, c, e, Fig. 10a, c) were now introduced into the figure and the types of analysis used in the figure legend.

C. We now provided for all experiments the quantification, numbers/percentages of the immunofluorescence of p16^{INK4a} and Ki67 (Fig. 1c, Fig. 4e, Fig. 8c). One representative image was given for each parameter, each data point was the summary of three analyses per mouse, and three mice have been analysed.

Thus, we statistically confirmed with two RT2-cancer models and one λ MYC model that ICB induced senescence using five of the most important senescence associated

parameters: ICB induced a stable growth arrest *in vivo*, after ICB the senescent RT2-cancer cells remained growth arrested when cultured *in vitro*, had increased numbers of SA- β -Gal⁺ cancer cells, expressed p16^{INK4a} and were deficient in Ki67. We further statistically prove that ICB failed to induce senescence and the senescence phenotype in RT2-cancers that were deficient in either Stat1 or p16^{INK4a}, or in preneoplastic B cells of λ MYC mice deficient in p21^{Cip1}.

As the proof of concept was done, we provided for the additional 39 supportive immune histology parameters that we analysed in up to eight cancer conditions one representative figure. The analyses were performed on one to two ‘representative’ tumours given in Fig. 1c, Fig. 4e, or Fig. 8d (where the statistics had now been performed). This provides 150 immune histology pictures, many of them with two to three parameters. Immune histological analysis of the three different experimental cancer models and settings (ICB therapy of transplanted RT2-cancers, ICB/AT therapy of endogenously growing RT2-cancers, and ICB therapy of λ -MYC lymphomas) always confirmed that ICB induced not only p16INK4a but also the other senescence associated surrogate marker and that the induction was dependent on: IFN- γ , STAT1, CDKN2a and p21. This underlines the high reproducibility.

12. Question

According to the labelling, Fig. S1C presents RT2 cells that lack p16INK4a due to CRISPR editing, but the main text confusingly refers to spontaneous p16INK4a-deleted tumors selected for *in vitro* and *in vivo*.

Answer

The old Fig. S1C (new Fig. 3b, c) showed in the middle group tumours with spontaneous p16^{INK4a}-deletion and in the last two lines CRISPR controls or p16^{INK4a}-deficient tumours due to CRISPR editing. We corrected this point in the new Fig. 3c, the figure legend 3 and the main text (beginning of page 5).

13. Question

Those Fig. S1C p16INK4a-deficient RT2 tumors seem to respond to ICB with H3K9me3 reactivity, similar to p16INK4a-proficient tumors – which is different from the interpretation stated in the text.

Answer

A recent manuscript in Nat Commun showed that CRISPR editing causes a DNA damage response that directly induces H3K9me3 (Natale et al. Nat Commun 2017). This was addressed on page 5, beginning 2nd paragraph, and we cited the manuscript.

14. Question

What do the authors mean by “electron microscopy confirmed the nuclear accumulation of SA-beta-gal in the senescent tumor cells”? SA-beta-gal is an enzymatic assay, which does not work anymore after fixation with glutaraldehyde. Moreover, SA-beta-gal staining is lysosomal, not nuclear (it is not clear to me what Fig. S5E – in the cytoplasm, not the nucleus – actually detected). And: what was the idea behind the ultrastructural analysis in addition to

conventional enzymatic SA-beta-gal staining?

Answer

Feil and co-worker established the electron microscopic analysis of SA- β -gal in senescent cells after fixation with glutaraldehyde (Feil S et al. *Circ Res* 2014). We used this technique to demonstrate SA- β -gal in senescent RT2-cancers. This technique allows a precise demonstration of SA- β -gal in the cytoplasm – we addressed this precisely in the figure legend (Supplementary Figure 9) and cited the reference in the methods.

We have to apologize the error in the figure legend of the previous manuscript.

15. Question

Fig. 3E is a comparison of ICB vs. control/mock treatment, and does, unlike stated in the text, not show that “these mice [*myc.p21CIP1*-/-] developed lymphomas with slower dynamics than *Myc* controls”. If this would be the case, it would be an unexpected observation, and different from the published literature (at least with respect to the *Eu-myc* transgenic line \pm *CIP1*). Again, what is the mechanistic basis for an ICB therapy (here: anti-PD1 plus anti-CTLA-A4) to unleash blunted cellular immunity against these endogenous *myc*-driven lymphomas? What are the immune checkpoint ligands expressed on the lymphoma cells conferring susceptibility to ICB here, in this model?

Answer

A) The difference was not significant; we deleted this sentence. Thank you for the comment.

B) We now showed (New Supplementary Figure 11b) that the lymph nodes in the λ *MYC* mice expressed PD-L1 the ligand for PD-1. We have previously shown that CD80 and CD86 are expressed on dendritic cells in λ *MYC* mice. They are primarily found in the spleen (Naujoks M et al. *Cancer Immunol Immunother*, 2014). Thus, CTLA had a target. Here we showed that only the combined ICB therapy (anti-CTLA4 and anti-PD1) provided long-term survivors (Fig. 8a; text page 8, 3rd paragraph)

C) Here we showed that the therapeutic effect of the ICB was strictly dependent on T cells. ICB did not prolong the life of T cell depleted λ *MYC* mice beyond day 200 (Fig. 8d). In the text we addressed this on page 9, 1st paragraph.

16. Question

The melanoma data are interesting, but, again, cause and consequence are not necessarily clear. It is no surprise, as repeatedly reported in the literature, that manifest melanomas possess senescence-inactivating gene lesions, either to bypass or to escape *Braf*- (and, to some extent, *Ras*-)driven senescence. Equipped with those lesions (e.g. *INK4a/ARF* loss), these tumors not only can't senesce anymore, they will also present with a variety of more aggressive growth properties (due to shutdown of both the *Rb* and the *p53* axis). Whether ICB is less efficient due to a blunted cell-related senescence response, as claimed here, or simply due to a reduced susceptibility to die to any pro-apoptotic trigger, or, alternatively or in combination, to more aggressively proliferate, remains to be shown.

Answer

Indeed, it was repeatedly reported in the literature that the loss of senescence controlling genes, like the *BRAF*^{V600E} mutation and especially the loss of *CDKN2a* are key factors for the

development of melanomas. Yet, in humans the loss of p16^{INK4a} was only shown to be relevant for the transition of nevi into invasive melanomas (Shain AH et al. *Cancer Cell* 2018; Zeng H et al. *Cancer Cell* 2018).

The association between the CDKN2a loss and the formation of metastases has only been described in mice (Krimpenfort P et al. *Nature* 2001, Lee S & Schmitt CA *Nat Cell Biol* 2019). Surprisingly, in a recent study Bastian and co-workers summarized that such an association or correlation has not yet been shown in humans (Shain AH et al. *Cancer Cell* 2018; Zeng H et al. *Cancer Cell* 2018). It is not known whether genes regulating the senescence pathway may cause melanoma metastases. In consequence, it is also unknown, whether such changes would affect the responsiveness of melanoma metastases to ICB. On the contrary, the only ‘senescence-associated’ molecule analysed so far in the context of ICB is the Braf^{V600E} mutation. This Braf^{V600E} mutation does not attenuate the response of melanoma metastases to ICB (Postow MA et al. *N Engl J Med* 2015). Interestingly, the only report showing that acquisition of defects in the IFN-signalling cascade results in treatment resistance, did even not address the possibility that the treatment resistance might result from defects in the senescence-signalling pathway (Zaretsky JM et al. *N Engl J Med* 2016).

Here we analysed, whether ICB is less efficient due to a blunted cell-related senescence response. Indeed the data showed that resistance of melanoma metastases to ICB was *significantly* associated with severe function altering mutations in a small panel of classical senescence-associated genes.

We did *not* analyse, whether ICB is less efficient due to a reduced susceptibility to die to any pro-apoptotic trigger. This is an interesting and important, but different question. For this question a genetic analysis would analyse a different panel of genes. Many of the genes that we analysed, like CDK4 and CDK6, that had ≥ 4 fold amplifications in 13% of the metastases, regulate senescence but not apoptosis.

To test the validity of our data we performed the positive and negative controls. Thus, metastases of responder patients had a significant increase in the total mutational burden (positive control), while the frequency of CNVs or SNVs did not differ between the two groups (negative control).

In consequence, our manuscript is the first one showing that function altering defects in the IFN-regulated senescence pathway were significantly associated with resistance to ICB in humans.

We provided the information in the manuscript in the new Fig. 10a-c and in the new Supplementary Figure 12 and discussed them in the revised version (page 11, 2st and 3nd paragraph).

To functionally ask whether the defects in the IFN-regulated senescence pathway of melanoma metastases from patients not responding to ICB may also impair CIS, we generated cell lines.

Cell lines that we could develop from biopsies of melanoma metastases with a major genetic defect in the IFN-regulated senescence pathway proliferated with the same dynamics as those derived from responder patients. They were normally susceptible to apoptosis induction. But, they were entirely resistant to CIS. Thus, the cells did not proliferate more aggressively *in vitro* and were not resistant to apoptosis induction (Fig. 10d, e).

In contrast, cell lines that we developed from biopsies of metastases of patients responding to ICB proliferated with the same dynamics as those from non-responder patients, and were also susceptible apoptosis induction. But, they were, in addition, susceptible to CIS (Fig. 10d, e).

Thus, the data prove that also in humans genetic defects that severely impair the IFN-regulated senescence pathway were associated with an unresponsiveness of melanoma metastases to ICB in humans. Many other resistance mechanisms do exist; yet these other resistance mechanisms were normally related to a defect in dying or to uncontrolled tumour cell proliferation. This was now more clearly addressed in the page 10, 2nd paragraph, in page 11 1st paragraph and the following discussion on page 11.

Reviewer #3

Major Points:

1. Question

-in Figure 1C,D: the authors performed in vitro treatment of RT2 WT or p16 null cells with IFN γ and TNF. Both cells underwent cell cycle arrest during treatment, however, p16 null cells escaped from senescence and kept proliferating. The authors should complement the analysis of cell number with SA- β -Gal assays to verify the establishment of senescence immediately after treatment in WT or p16 null (as well as induction of p21 and possibly DDR activation).

Answer

We now performed SA- β -Gal assays in WT or p16 null cancer cell lines. This was now shown in the new Fig. 3a-c. DDR activation was studied on tumour samples isolated from ICB treated mice. We did not find signs for DDR as neither DNA-PK nor γ H2AX were increased in the tumour cells analysed (Supplementary Fig. 5a, b; text page 6 top).

[redacted]

2. Question

-in Figure 1F: I noticed some variability in the control treated group between animals with RT2-CRISPR-p16 cells. What is the explanation for this variability? Did the authors verify the loss of p16 in these samples by surveyor/NHEJ molecular assays? The authors should also rule out the possibility that their CRISPR-mediated KO does not impact on the expression levels of other genes in the INK4a locus (e.g. p19/ARF?).

Answer

A. We used polyclonal cell lines that were, in addition, derived from different primary tumours and mice. Polyclonal cell lines frequently show a variable growth pattern (Fig. 1a). Yet, all cell lines of each group showed exactly the same response pattern, what supports the validity and stability of the system. We addressed this aspect in the manuscript on page 4, 2nd paragraph.

B. We checked the p16 knock-out by PCR (New Supplementary Figure 3b). We also showed that other important tumour-associated mRNA (SV40-Tag) levels and SV40-Tag protein remained unaffected (New Figure 5b, c).

C. Here we have to apologize that we were not precise enough in our material and methods section: we did simultaneously delete p16^{INK4a} and p19^{ARF}. We now corrected this naming CRISPR-p16^{INK4a^{-/-}}-p19^{ARF^{-/-}} throughout the text (beginning page 5), the figure legend (Fig. 2, 3, Supplementary Figure 2, 3, 5). In the original version we only referred to the construct by a reference, where we had previously published this construct (reference 33). We agree that it is clearer to describe it again. The exact description of the CRISPR construct is now in the materials and methods, page 32.

3. Question

-Figure 2/Suppl. Figure 2: is there DDR activation in tumor cells induced into senescence in response to ICB or ICB/AT treatment? IHC stainings for γ H2AX and pATM would be an important addition to the manuscript.

Answer

We performed the suggested analyses. For the pATM no appropriate antibody was available for staining the multiple antibodies; we therefore stained the sections with an anti-DNA-PK antibody, as DNA-PK is a direct inducer of pATM. As these two molecules are closely associated with double strand breaks, they remained largely negative. Only γ H2AX was positive in single cells of the ICB/AT treated RT2 mice. This was now shown in the new Supplementary Fig. 8 and addressed in the text on page 6, bottom.

[redacted]

4. Question

Figure 2/3: In the RT2 model the authors should complement the analysis on senescence in tissue section with an in depth characterization of immune-infiltrates. Cellular stainings for cytotoxic T cells, Tregs, NK and macrophages will be very informative in all treated groups.

Answer

We performed the suggested characterization of the immune infiltrate, for the transplanted RT2-cancers (where no CD8 cells can be found because of the treatment with the anti-CD8 mAb), for the endogenously growing RT2-cancers, and for the λ MYC tumours (that were the former Fig. 3).

For the transplanted cancers, Supplementary Figure 4:

- a) anti-CD3 (was positive) and anti-CD8 (was negative)
- b) F4/80 positive macrophages (was positive)
- c) MHC class II cells (macrophages; was positive)
- d) CD49⁺ NK cells (was negative)

For the endogenously growing RT2-cancers, Supplementary Figure 7

- a) anti-CD3 (was positive) and MHC class II (was positive)
- b) F4/80 positive macrophages (was positive). Foxp3⁺ regulatory cells (almost negative, very few, see inset)

c) CD8⁺ T cells or CD49b⁺ NK cells (was negative)

For the endogenously growing λ MYC tumours, Supplementary Figure 11

a) anti-CD3 and CD20 (positive and normal structure only in healthy, ICB treated mice)

c) MHC class II (slightly positive)

d) CD8⁺ T cells (slightly positive) or CD161⁺ NK cells (was negative; slightly positive only in healthy, ICB treated mice)

The data were introduced into the text, where we characterised the single tumours.

5. Question

Figure 3d,e: there is discrepancy in the two graphs regarding the number of alive mice at 200 days post treatment (0 in panel d; 1 in panel e)

Answer

It is correct that figures 3d and 3e showed discrete differences in the number of mice alive at 200 days. As the figure are from two different experiments, they should be slightly different. As the data were very close, they underline the high reproducibility of these experiments.

General notes:

6. Question

- I understand if there is a hard limit of characters for this manuscript format, but I feel that the introduction is very short. I would also recommend the authors to expand on the discussion section. The authors should have in mind the very recent discovery by the lab of C. Schmitt on the possible reprogramming of cancer cells induced into senescence by antineoplastic treatments (Milanovic et al. Nature 2018; Lee et al. Nature Cell Biology 2019). Of note, while the authors focused on the cell-autonomous aspects of cellular senescence, they cannot ignore that activation of SASP from senescent cells may be reinforcing senescence in the short-term but contribute via chronic inflammation to cancer relapse and senescent escape in the long-term (Demaria et al. Cancer Discovery, 2017). The likelihood of this phenomenon is particularly high in elderly subjects where a compromised immune system may delay senescent cells eradication.

Answer

We thank the reviewer for this comment. As we now have the possibility to extend the number of words. We now address these aspects in the new discussion section on page 11, 3rd paragraph, especially the role of p16^{INK4a} in preventing the transformation of senescent cells into cancer initiating cells that do no longer respond to therapies as worked out by Milanovic et al. Nature 2018 and carefully discussed by Lee et al. Nature Cell Biology 2019.

Reviewer #4

The manuscript by Brenner et al., describes how a p16 and p21-dependent cancer cell senescence program may promote checkpoint blockade (CB) and adoptive cell transfer (ACT) mediated immune regulation in syngeneic solid and liquid tumors. The authors suggest that cellular senescence, driven by cell-cycle regulators, is a fundamental mechanism by which T cells prevent cancer-cell escape from immunosurveillance. In addition, they provide correlative evidence that induction of cancer-cell senescence, through activation of p16 and p21, is dependent on IFNR-STAT1 paracrine signaling from immune cell IFN γ . While these findings are intriguing, they are largely correlative in nature and lack a mechanistic basis. Additional biochemical, in vitro, and in vivo experiments are necessary to substantiate the authors claims. Furthermore, the authors fail to provide convincing evidence that IFNR-dependent activation of STAT1 in cancer cells is responsible for the observed senescence phenotype. In its current state, the data is too premature for publication.

Answer

We thank the reviewer for his comments. Probably we were not clear enough in describing the goal of our experiments – for this we have to apologize.

The manuscript has one key question: does cancer immune control require the p16^{Ink4a}p19^{Arf}-dependent signalling pathway or the p21^{Cip}- dependent signalling pathway to control those cancers that are not killed by cytotoxic T cells, and are these signalling pathway needed to induce senescence in those tumour cells that survive the killing by T cells and activated macrophages.

For this question we provided the proof of concept.

A) We provided proof of concept that loss mutants and knock-out tumours of p16^{Ink4a}p19^{Arf} that escape from killing cannot be controlled by immune check blockade (ICB) and fail to become senescent (Fig. 1, 2, 3).

B) We provided proof of concept that p21^{Cip} knock-out tumours that escape from killing cannot be controlled by ICB and fail to become senescent (Fig. 8).

C) We provided proof of concept that ICB required STAT1 and sufficient amounts of IFN- γ to activate p16^{Ink4a}p19^{Arf} and p21^{Cip} in tumour cells (Fig. 1, 4, 8).

Various IFNR knock-out (IFNR.KO) mutant mice show very different responses to infections with *leishmania* major or viruses (for review: van den Broek M et al. Immunol Rev 1995). Therefore the manuscript here focussed on type II IFN and the downstream effector STAT1. We did not analyse the role of the various IFNR.KO mutant variants. Because of the complexity of the INFR-signalling pathway, that signals also through pathways other than STAT1, analysing the role of the INFR would be a different question.

To address the reviewers' question, we therefore [redacted]

The following major points should be considered.

1. Question

The mechanism associated with tumor rejection in mice receiving immune checkpoint blockade (ICB) is unclear. The authors indicate that CD8 depletion is necessary to prevent RT2 tumor rejection in wildtype treatment naïve mice. Data illustrating the efficacy of depletion and the presence or absence of CD8 T cells at the time of ICB is critical for understanding the significance of the findings in Figure 1. If CD8 T cells have been effectively depleted what is the proposed mechanism of tumor rejection upon ICB? If CD4-dependent mechanisms are proposed, data illustrating altered Treg or Th1 biology associated with rejection need to be shown. Furthermore, ICB works in T cell depleted mice in Figure 1.a, but not in Figure 3.d. The authors need to address the question of the proposed mechanism for ICB efficacy upon T cell depletion in Figure 1.a, that is not occurring in Figure 3.d.

Answer 1

A. In Fig. 1 we depleted the CD8 T cells as the transplanted tumours did not grow in fully immune competent mice (0/20 what is the real number). Depletion of CD8 T cells with mAb at the time of transplantation is a protocol commonly used in tumour immunology to enable the initial tumour growth. [redacted] We addressed this point in the main text and especially in the legend Fig 1 and Supplementary Fig. 1a.

B. In the new Supplementary Figure 4a, we show that CD4 T cells ($CD8^-CD3^+$) infiltrate the tumours and that this is associated with the infiltration of activated macrophages (new Supplementary Figure 4b, c) and the increased expression of $\beta 2$ -microglobulin, and of PD-L1 (new Supplementary Figure 2a-d). Activation of macrophages, induction of PD-L1 and especially of $\beta 2$ -microglobulin in tissues results from IFN- γ -producing T_H1 cells (as no CD8 cells were present; NK cells were almost undetectable).

R Zinkernagel and co-workers were among the first showing that CTL, perforin and granzyme are required for the control of many cancers. On the other side, a significant number of cancers requires, instead of CTL, activated T_H1 cells, killing by activated macrophages and dendritic cells (van den Broeck M et al. J Exp Med 1996). Similar data were published by others (e.g. Kang T et al. Nature 2011).

In line with this, the RT2-cancers can be controlled by IFN- γ -producing T_H1 cells in the absence of CD8 T cells (Muller-Hermelink N et al. 2008).

C. The design of the experiment Fig. 1a is fundamentally different from the experiment in Fig. 8d (former Fig. 3d).

In Fig. 1a we depleted CD8 T cells to permit the growth of the transplanted tumours.

In Fig. 3d we deleted all T cells (CD4 and CD8) with a pan-anti T cell mAb to ask, whether T cells are required for efficient ICB. As expected and in line with all reports the data show that

T cell are needed for efficient ICB (Fig. 8d). This is now more clearly explained in the text on page 9, 1st paragraph.

2. Question

The rationale for the different immunotherapies used and their treatment regimes applied in the different experiments is unclear.

Answer

There are currently two different ICB therapies either already in the clinics or very promising in preclinical and clinical studies for advanced cancers.

The combination of anti-CTLA4 mAb combined with anti-PD-1 mAb is currently the most effective, clinically approved ICB combination therapy (Postow MA et al. N Engl J Med 2015).

The combination of anti-Lag3 mAb with anti-PD-L1 mAb is highly promising in clinical studies and phase III studies are under investigation. According to preclinical data the combination of anti-Lag3 mAb with anti-PD-L1 mAb is highly efficient and, when combined with adoptive T cells transfer, the most efficient one (Goding SR et al. J Immunol 2013), page 3, bottom.

To test whether immune activation with ICB induces cancer cell senescence independent of the type of antibody combination used, we performed the experiments with either one of the two ICB therapies. We treated mice either with anti-Lag3 mAb and anti-PD-L1 mAb (Fig. 1 and Fig. 4) or with anti-CTLA4 mAb and anti-PD-1 mAb, we now clarified this ICB treatment regimen in the figure legend new Fig. 8.

3. Question

The lack of statistical comparison between the tumor volumes illustrated in the spider plots in Figure 1 confuse data interpretation. Group differences that are significant should be illustrated, and tumor sizes should be represented as volume. While the individual tumor growth curves illustrate group variance and highlight outliers, growth curves with means and standard error of mean, and accompanying significance indication, should be provided to allow for direct comparison of tumor sizes between groups. Mean data from the different groups should be presented on a single plot illustrating tumor volumes. Additionally, it appears that p16 loss is much more aggressive than STAT1 loss in these cancer cells (Figure 1.b, e), suggesting alternative mechanism than the one proposed by the authors. However, it is difficult to discern as no statistics or direct comparison can be made with the plots provided.

Answer

Figure 1 was changed as proposed by the reviewer. We also calculated the tumour volumes and provided the statistics. (Please note that we give a summary curve with the slope of the tumour growth inside the spider plot, as instructed by the statistics department; ANCOVA).

Importantly, we did not work with a RT2 clone. The RT2-cancers, the STAT1-deficient RT2-cancers, the CDKN2a-deficient cancers were all derived from different cell lines that were from different primary tumours and from different mice. Therefore, all had slightly different growth dynamics *in vivo*. We addressed this in the text on page 4, bottom.

Despite the slightly different growth dynamics, 8/8 of the STAT1-deficient RT2-cancers or 19/20 the CDKN2a-deficient RT2-cancers were resistant to ICB and to senescence

induction. 9/10 of the original RT2-cancer lines were responsive to ICB and became senescent in response to ICB. This underlines the reproducibility and validity of the system.

4. Question

For Figure 1.b, the lack of p16 induction in a STAT1 dependent manner needs to be validated with biochemical evidence. The authors should at least show quantification of IF. In addition, qPCR for p16 transcript should be shown. Furthermore, β 2M is also a STAT1 target gene that is critical for immune response post ICB. The lack of ICB in Stat1^{-/-} RT2 cells could be a result of β 2M loss in these cells. STAT1-dependent efficacy of immune responses to solid tumors is not novel. The authors need to clearly demonstrate that STAT1 activation in these cancer cells results in the transcription of p16, and that this is required for sensitization to ICB.

Answer

A. [redacted]

B. As previously reported, TNF and IFN induce p16^{INK4a} primarily through posttranscriptional mechanisms and quantification of the p16 transcript does not reflect necessarily the quantity of p16 produced (Braumuller H et al. Nature 2013).

C. We showed that interferon induces β 2M only on STAT1-expressing RT2 cells and not on Stat1^{-/-} RT2 cells (New Supplementary Fig. 2d). As STAT1-positive macrophages and T cells infiltrate the cancers during ICB, β 2M is expressed on the immune cells infiltrating the Stat1^{-/-} RT2-cancers of ICB treated mice (New Supplementary Fig. 2c).

D. We agree that the statement that STAT1 is involved in the cancer control is not novel. We explicitly addressed in the introduction (page 3) that IFN- γ (and in consequence STAT1) activate multiple important mechanisms that are critical for cancer immune control, such as induction of MHC class I (and in consequence β 2M), anti-angiogenic effects and various others.

It is a key message of the manuscript to show that STAT1 is needed to activate p16^{INK4a} during the anti-tumour immune response. We provided the proof of concept as follows:

- Loss mutants and knock-out tumours of p16^{INK4a}p19^{Arf} that escape from killing cannot be controlled by immune check blockade (ICB). They fail to become senescent during ICB (New Fig. 1- 3). Importantly, p16^{INK4a}p19^{Arf}-deficient cancer cells were normally responsive to interferon- γ . Interferon- γ induced normal levels of PD-L1 or of β 2M in the p16^{INK4a}/p19^{ARF} deficient cancer cell lines and cancers (New Supplementary Figures 2b, d).
- We provided proof of concept that ICB required STAT1 in transplanted (New Fig. 1) and in endogenously growing (New Fig. 4) RT2-cancers to control the RT2-cancers, to induce senescence and to activate p16^{INK4a}.

Thus, STAT1 activation in the cancer cells was needed to induce p16^{INK4a} protein. In seven independent p16^{INK4a}p19^{Arf} loss mutants, by natural selection or CRISPER.KO, we showed that p16^{INK4a}p19^{Arf} was required for cancer control by ICB.

5. Question

The authors claim CRISPR-Cas9 p16-deficient cancer cell model is similar to the p16-deficient cells derived from tumors. However, based on the data in Figure S1 there are clear differences between these two model systems. Additionally, the methods for the establishment of the CRISPR cell line needs substantial elaboration.

Answer

CRISPR-Cas9 p16-deficient cancers were similar to the p16-deficient cells established by *in vitro* or *in vivo* selection as:

A. The CRISPR-Cas9 construct itself was highly immunogenic. Even in the CD8-depleted mice all CRISPR-Cas9 controls were rejected and only 80% of the CRISPR-Cas9 p16-deficient cancers did grow. The initial growth was also slower. But, importantly, once tumours did grow, all continued to grow and were not rescued by ICB (New Fig. 1a). Thus, the CDKN2a loss mutants were resistant to ICB, whether CDKN2a was lost by selection or deleted by CRISPR-Cas9.

B. Whether CDKN2a was lost by selection or deleted by CRISPR-Cas9, both were resistant to senescence induction, *in vitro* and *in vivo* (New Figure 1b, c, d; New Figure 3b, c).

C. Interferon induced PD-L1 and β 2-microglobulin in all p16^{INK4a}/p19^{ARF} deficient cell lines, whether CDKN2a was lost by selection or deleted by CRISPR-Cas9 (New Supplementary Fig. 2a-d). This also shows that they were responsive to interferon- γ .

D. In the materials and methods section the methods for the establishment of the CRISPR cell line were substantially elaborated, page 32.

6. Question

For Figure 1 and 4, the lack of cytotoxicity upon TNF and IFN γ treatment needs to be shown. Cell number is not an adequate measure of cytostatic effect of treatment. It is further necessary to provide Annexin V staining for the cells after treatment and before the “wash” step.

Answer

Treatment of cancer cells with TNF and IFN induces apoptosis in some cancer cells and senescence in others. The percentage of cells that become apoptotic is very variable. At the concentrations used, TNF and IFN induced apoptosis only in a small number of Rip1Tag2 cancer cells or the melanoma cell lines that we had derived from patients not responding to ICB and from patient responding to ICB.

As also addressed in the answer to reviewer 2, question number 6, and in line with recent literature (e.g. Demaria et al. Cancer Discovery, 2017) we found activation of many pro-apoptotic genes in response to TNF and IFN- γ . Yet, these cells are protected from cell death by the simultaneous activation of Bcl-2. Blocking of Bcl-2 induces apoptosis in senescent cells – also in cells after CIS (see also Demaria et al. Cancer Discovery, 2017).

Therefore, [redacted]

Annexin V staining of melanoma cells lines that are given in Fig. 10d, e, in the presence or absence of 100 ng/ml IFN-g and 10 ng/ml TNF [redacted]. Please note: We show these concentrations, as we used these concentrations of TNF and IFN throughout the experiments. At higher concentrations TNF and IFN induce apoptosis/Annexin V in larger numbers of the melanoma lines. We introduced these important aspects into the result section page 11 top, and into the discussion page 12, 3rd paragraph.

7. Question

Data in Figure 3 is largely phenomenological, with no clear indication of mechanism. IFN γ has effects on a variety of cell types including myeloid and stromal cells within the TME. The lack of p16 induction in anti-IFN γ treated mice is correlative. The authors need to provide direct evidence that this is functioning in a manner dependent on IFNR signaling in cancer cells. It needs to be clarified if anti-IFN γ treatment has the same effect in IFNR $^{-/-}$ RT2 cells.

Answer

As the different IFNR-KO mouse lines show very different immune responses (van den Broek M et al. Immunol Rev 1995) [redacted]

8. Question

In vitro and biochemical validation are needed to substantiate the claims that the effect of ICB observed is at least partially explained by IFNR-dependent activation of STAT1 in cancer cells that leads to p16 and/or p21 activation.

Answer

[redacted]

Thus we performed experiments for the reviewer, and they confirmed that senescence required the IFN- γ -IFNR-STAT1 signalling to induce senescence. Because of the complexity of the topic addressed by the reviewer (biochemical breakdown), we performed the experiments but feel that the integration of this topic would further complicate the manuscript. Moreover, the data are preliminary. Yet, based on the interesting suggestions and early results we plan to address this topic in detail in future research.

We thank all reviewers for their time and their valuable suggestions that definitely improved our manuscript. We hope that we sufficiently addressed their concerns and that they will like the version of the manuscript.

[redacted]

Reviewers' comments:

Reviewer #1 (Remarks to the Author):

The authors have addressed the comments of the reviewers

Reviewer #2 (Remarks to the Author):

Brenner-E,... ..Röcken-M, Cancer immune control needs intratumoral senescence-inducing p21CIP1 and p16INK4a pathways

Re-submission to Nature Communications

This is now the revised version of a manuscript by Röcken and colleagues that links cell-based anti-cancer immunity – beyond its well-established cytotoxic mode of action – to induction of cellular senescence. The central claim is that defective target cell senescence (e.g. by deletion of the senescence mediators p16INK4a or p21CIP1) impairs natural or immune checkpoint blockade (ICB)-exerted tumor control.

The revision addressed some, but by far not all of my concerns. Regarding the overall concept and in its course through the figure flow, the paper remains very difficult to follow. Important experimental conditions (e.g. whether or not an anti-CD8 pre-treatment was applied) can neither be found in the text nor in the figure legend. Genotypes are labeled imprecisely, although it is of pivotal importance whether a tumor is just p16INK4a-deficient, INK4a/ARF-deleted or even possesses bi-allelic gross deletions that extend to other loci beyond the CDK2NA locus. A two-antibody (Lag3, PD-L1) immune checkpoint double-blockade principle is used (claimed to be “one of the most efficient ICB combination therapies published” – which might be true with respect to some selected preclinical settings, otherwise an irritating over-statement, since anti-Lag3 is not used in the clinic), but not experimentally validated. Sometimes, the rationale behind an experiment remains poorly unexplained (e.g. in Suppl. Fig. 2a: why PD-L1 immunostaining after PD-L1 blockade in “remaining” cells?), claimed conclusions cannot be retraced from the figure (e.g. in Suppl. Fig. 2c: what are tumor cells, what is “microenvironment?”), or provides even clashing interpretations between main text and figure (e.g. senescence markers in RT2-CRISPR-p16INK4a-/- cells in Fig. 2a-c), with the general issue of presenting “remaining” tumors of very different size after ICB – hence, uncoupling overall tumor regression from remainder biology from clinical long-term outcome, which forces readers to re-integrate numerous data sets in their mind (e.g. Fig. 1a in correspondence to Fig. 1b to Fig. 2...), making this tedious and scientifically difficult. Improper referencing (such as “loss of INK4a/ARF promoting cancer stemness – not shown in the cited Ref. 25) adds to the blurry picture, out of which complex conclusions are drawn: senescence needed for full cell-based immune control, senescence defects promoting immune escape and cancer stemness. Such far-reaching conclusions are simply not sufficiently substantiated with analyses of “remaining” (for which many different reasons may apply) tumor cells at a single point in time. It is very well conceivable that many findings presented here are due to the lack of the Rb/p53 pathway-controlling INK4a/ARF locus independent of any senescence/TH1 immune cell interaction. On the contrary, this does not exclude that the authors made an important observation.

In essence, I suggest to tone down a bit, to fundamentally revise structure and phrasing, condense overall content, clearly explain experimental settings (e.g. why were CD8+ T-cells depleted?), sharpen the message figure by figure and paragraph by paragraph (eliminate overinterpretation of often less clearly discriminating datasets), include more subheadings, refrain from permanently jumping back and forth between figure panels, provide more integrative views on tumor growth, biology, molecular mechanisms and immune interaction over time, be clear on model systems (e.g. the repeatedly misleading presentation of actually INK4a/ARF-deleted tumors as “p16INK4a-/- tumors”), avoid misinterpretation of formally non-significantly different findings

as biologically equivalent, and present the entire central message as an additional, not the major principle underlying impaired outcome of INK4a/ARF-deficient tumors to ICB.

Remaining more specific concerns and comments

1. The title – as done so in the abstract – should be adapted to the re-defined statement that it's not seldomly p16INK4a but the deletion of both p16INK4a and p19ARF that disables cytokine-induced target cell senescence (CIS). Also, the genetic designation of p16INK4a-deficient RT2 tumors, e.g. in Fig. 1 – explicitly distinguishing between p16INK4a^{-/-} and p16INK4a^{-/-};p19ARF^{-/-} cells – is misleading. According to the methods section, p16INK4a-deficient cells are "CDK2NA loss variants on chromosome 4" generated as survivors of ICB exposure (Suppl. Fig. 3a – hard to confirm for the reader regarding RT2 precursor line #1, but, if so, showing equally reduced signals for both chromosomes 4 and 5), thereby possibly selecting for gross deletions that include adjacent genes (e.g. MTAP, ANRIL, p15INK4b), and most likely disrupting both alleles of CDK2NA-exon 2, thereby cancelling both p16INK4a and p19ARF expression. Likewise, "CRISPR-p16INK4a" tumors are actually (see methods) p16INK4a/p19ARF-co-deleted tumors, but their label in Fig. 1 and subsequent figures implies that only p16INK4a-deficient tumors were generated by the CRISPR approach. This is particularly confusing, because remaining "RT2 CRISPR p16INK4a^{-/-}" tumors, in contrast to "RT2 p16INK4a^{-/-} tumors", present with and without ICB exposure positive for senescence markers p21CIP (significant), HP1gamma (weak) and H3K9me3 (strong), albeit no SA-beta-gal activity (Fig. 2a-c). Later in the text, the authors explain the H3K9me3 result by their assumption that CRISPR-Cas9 transfection may cause DNA damage – but provide no evidence for this claim in Suppl. Fig. 5a, where these cells remain negative for the DNA damage response marker gamma-H2AX. If CRISPR-driven ongoing DNA damage would actually occur, I would worry even more about the validity of this CRISPR model to study senescence as a read-out.

2. The legend of Suppl. Fig. 1a does not explain "RmCD8", a CD8-depleting antibody pre-treatment according to the "methods" section. The cartoon suggests that tumors formed at a size of > 3mm within one week – while the CD8-depleting antibody should be still around at quantitative concentrations. This is very puzzling, as no CD8⁺ T-cells would be available for immune checkpoint blockade (ICB) treatment at this time. Moreover, since no functional, T-cell-based data on anti-Lag3/anti-PD-L1 ICB are shown, it is entirely unclear how this ICB cocktail exerts its function. Are the authors postulating a specific role of endogenous CD4-positive T-cells? What presumed immune target cells actually express PD1 and/or Lag3 at the time of ICB administration?

3. What do the authors intend to explain in Suppl. Fig. 2, when showing PD-L1 IF on remaining tumor cells prior to and after presumably saturating anti-PD-L1 antibody exposure? Moreover, according to the main text, Suppl. Fig. 2b, d supposedly shows an analysis of T antigen ("Tag") expression, which is not the case.

4. With no error bars shown, cells in Fig. 3c behave in growth curve analyses very similar in response to IFN-gamma/TNF-alpha – irrespective of p16INK4a (i.e. INK4a/ARF) deletion by CRISPR-Cas or not. Such data is of key importance to support the hypothesis of the authors, the differences observed are marginal, but the wording in the text is black and white ("...control cells susceptible to CIS... ..p16/p19-deficient cells resistant to CIS". While the referee did not overlook clear differences in SA-beta-gal responsiveness, the situation is odd and requires clarification, i.e. more robust biological evidence (additional senescence markers, time-course analyses etc.).

5. Irrespective of the drug-induced cell death data in Fig. 3a-c, it cannot be ignored that p19ARF-

depleted cells harbor a p53 pathway defect, which is likely to contribute to better engraftment or reduced ICB responsiveness – but not necessarily due to a senescence defect. Subtle differences in p53 activity may translate into a slightly reduced apoptotic propensity, which is hard to see in short-term drug-induced cytotoxicity assays, but becomes very relevant in tumor outgrowth experiments regarding number and nature of remaining cells.

6. If the tumor cells continue to express Large T antigen, as stated in the legend of Suppl. Fig. 4 and shown in a single case in Fig. 5, how is senescence induction possible, given the inhibition of both the Rb and the p53 pathway by Large T? The authors themselves admit that “RT2 cancers are inappropriate to carefully investigate the role of p21CIP1 induced senescence in response to ICB” – but, actually, the statement could extend to “...inappropriate to carefully investigate the role of senescence...” ...at all.

7. There is no information how Large T antigen-specific TH1 cells were generated. Again, it remains unclear to the readers why CD8+ T-cells were eliminated from the experimental setting. In turn, the actual mode of tumor cell elimination is speculative (“most likely through the induction of cytotoxic macrophages”), although deciphering the underlying immunological mechanism appears to be central for the claimed link between senescence and immune clearance.

8. What’s the long-term fate of residual, senescent RT2 tumor cells after ICB plus adoptive anti-Large T antigen TH1 transfer (AT)? How stable is this type of senescence (with apparently strong p16INK4a induction, now virtually absent in ICB-only treated RT2 tumors... [Fig. 4d]) in vivo, what’s the fraction and timing of re-progressing tumors, do the cells lose the large T antigen, and would re-treatment with ICB or AT or both again result in tumor control?

9. The lambda-myc lymphoma experiment is interesting, since it is not based on expression of the problematic, senescence-compromising Large T antigen. Here, it is unclear to the reader whether CD8+ T-cells were pre-eliminated in this model prior to a different ICB combo (now anti-PD1 plus anti-CTLA4), too. While – somewhat surprisingly – lymphomas lacking p21CIP1 present with a senescence defect, the results are confusing. Essentially, the differences observed in p21CIP1-proficient lymphomas for untreated vs. ICB (Fig. 8a) can be superimposed (including virtually identical median survival times) with the results shown for p21CIP1-deficient lymphomas in Fig. 8e, but the latter being statistically “not significant” – which, if true, is clearly due to lower sample numbers. To argue that “non-significant” means no biological difference, and concluding that this provides proof for the requirement of an intact senescence response for full ICB activity is simply not correct.

Reviewer #3 (Remarks to the Author):

The authors have successfully addressed all my criticisms. This work will be of great interest to scientists interested in senescence and onco-immunology. More broadly, it will impact on the biological understanding of a variety of tumor types in response to innovative immunotherapies as well as inform on new therapeutic regimens that combine immunotherapy with pro-senescence treatments. I recommend publication in Nature Communications.

Minor notes:

Figure 1C: as the statistics refers to the effect of treatment in triplicates from one mouse I recommend to remove p-values and leave only a descriptive stat. This in general should apply to all experiments with n=3.

Figure 7: the labels for the 3 panels should read ICB/AT instead of ICB.

Figure Suppl. 12: misspelled "Amplification" in the legend

Please check the first two sentences in the Discussion.

In general, there are quite a few misspelled words/typos in the manuscript that should be corrected by the authors or at the proof stage.

Reviewer #4 (Remarks to the Author):

The manuscript has been substantially revised to address the majority of my original considerations. The added experiments, and altered presentation and discussion of the data make the manuscript acceptable for publication.

Answer to the Reviewer

We thank the editors and reviewers for their comments that helped us to develop the statements and readability of our manuscript. Our answers to your concerns are labelled in blue. The changes in the text and figures are labelled in red.

Revision II

Manuscript NCOMMS-18-35742A: Brenner-E,... ..Röcken-M,

Cancer immune control needs intratumoral senescence-inducing p21^{Cip1} and p16^{Ink4a} pathways

Referee #2

This is now the revised version of a manuscript by Röcken and colleagues that links cell-based anti-cancer immunity – beyond its well-established cytotoxic mode of action – to induction of cellular senescence. The central claim is that defective target cell senescence (e.g. by deletion of the senescence mediators p16INK4a or p21CIP1) impairs natural or immune checkpoint blockade (ICB)-exerted tumor control.

The revision addressed some, but by far not all of my concerns.

Below, we have divided the questions of referee #2 into 25 single points that we have identified and answered all of them precisely.

Comment 1

Regarding the overall concept and in its course through the figure flow, the paper remains very difficult to follow.

Answer

We restructured the manuscript as proposed, and subdivided especially the 1st and the 2nd result section in paragraphs following exactly the experimental flow paragraph by paragraph (page 4-6).

Comment 2

Important experimental conditions (e.g. whether or not an anti-CD8 pre-treatment was applied) can neither be found in the text nor in the figure legend.

Answer

We improved the methodological details. The anti-CD8 pre-treatment is now more precisely described in the text and we provide a reference that leads to the source (page 3, end and further page 4; page 5 paragraph 2), in the legend to *Fig. 1* and the legends of the *Supplementary Figure 1* (reference Egeter O et al. 2000 *Cancer Res* 60, 1515-1520 leads to the source) and *Supplementary Figure 6*, and in the *Materials and Methods* section *Treatment of RT2-cancers in C3HeB/FeJ mice* - reference Egeter O et al. 2000 *Cancer Res* 60, 1515-1520 leads to the source. Furthermore, we improved the methodological details of the SA-β-gal staining and refer to it correctly where we show the first SA-β-gal staining (the former *Supplementary Figure 14* is now *Supplementary Figure 2*).

Comment 3

Genotypes are labeled imprecisely, although it is of pivotal importance whether a tumor is just p16INK4a-deficient, INK4a/ARF-deleted or even possesses bi-allelic gross deletions that extend to other loci beyond the CDK2NA locus.

Answer

We revised the nomenclature of our cell lines and replaced *p16^{Ink4a}*-deficient by *Cdkn2a*-deficient, RT2.*Cdkn2a*^{-/-} or RT2.CRISPR-*Cdkn2a*-cancer cells in the entire manuscript. The genotype *Cdkn2a* is now precisely labelled from the beginning of the result section on page 3, 1st paragraph.

Comment 4

A two-antibody (Lag3, PD-L1) immune checkpoint double-blockade principle is used (claimed to be “one of the most efficient ICB combination therapies published” – which might be true with respect to some selected preclinical settings, otherwise an irritating over-statement, since anti-Lag3 is not used in the clinic), but not experimentally validated.

Answer

We toned down this point, by describing that the blockade of either LAG-3 or of PD-1/PD-L1 can each enhance immune reactions against tumours, and that the combined blockade of both molecules is more effective than the blockade of either molecule alone. Moreover, we address that the combined blockade of both molecules is currently investigated in clinical trials. For this, we cite the appropriate literature (Nguyen LT & Ohashi P (2015) Nat Rev Immunol 15, 45-56). Page 4, 1st paragraph.

Comment 5

Sometimes, the rationale behind an experiment remains poorly unexplained (e.g. in Suppl. Fig. 2a: why PD-L1 immunostaining after PD-L1 blockade in “remaining” cells?), claimed conclusions cannot be retraced from the figure (e.g. in Suppl. Fig. 2c: what are tumor cells, what is “microenvironment?”),

Answer

The rationale is now answered in the legend. In case of *Supplementary Figure 2a*, what is now the new *Supplementary Figure 4a*: “To detect the PD-L1 target molecules also on cancers of mice treated *in vivo* with mAb PD-L1 (clone 10F.9G2), staining was performed with a second unlabelled rat anti-mouse mAb PD-L1 (clone MIH6) and, as second step, with a fluorescence labelled anti-rat antibody.”

In case of *Supplementary Figure 2c*, what is now the new *Supplementary Figure 4c*, in *Stat1*-deficient cancer cells the β 2-microglobulin expression cannot be induced by IFN- γ (see *Supplementary Figure 4d*). Therefore, *in vivo* only the tumour-infiltrating host cells can express β 2-microglobulin. The wording was specified in the text page 4, 2nd paragraph.

Comment 6

or provides even clashing interpretations between main text and figure (e.g. senescence markers in RT2-CRISPR-p16INK4a^{-/-} cells in Fig. 2a-c),

Answer

In *Fig. 1*, we show that the immune therapy induced a senescence phenotype in tumour cells only if they were positive for *Stat1* and for *Cdkn2a* (the gene of *p16^{Ink4a}* and *p19^{Arf}*).

In *Fig 2a-c*, we further analysed the tumour tissues of the tumours shown in *Fig. 1*. In line with *Fig. 1*, the immune therapy also induced other important senescence markers only in the tumours that regressed during ICB treatment (*pHP1 γ* , H3K9me3, and SA- β -gal). Immune therapy failed to induce the senescence markers (*pHP1 γ* , H3K9me3, and SA- β -gal) if the tumour cells were *Stat1*^{-/-} or *Cdkn2a*^{-/-}. The *Stat1*^{-/-} or *Cdkn2a*^{-/-} tumours did proliferate, were positive for the proliferation marker Ki67 and did not express one of the additionally analysed 4 senescence markers. We readapted our wording to make the point above clearer, page 4, 1st paragraph.

Comment 7

with the general issue of presenting “remaining” tumors of very different size after ICB – hence, uncoupling overall tumor regression from remainder biology from clinical long-term outcome, which forces readers to re-integrate numerous data sets in their mind (e.g. Fig. 1a in correspondence to Fig. 1b to Fig. 2...), making this tedious and scientifically difficult.

Answer

We changed our writing, replaced the word ‘remaining’ by the common expression ‘residual’ and clarified the structure as suggested. Page 4, 1st paragraph; page 7, 1st paragraph and page 13, 2nd paragraph.

Comment 8

Improper referencing (such as “loss of INK4a/ARF promoting cancer stemness – not shown in the cited Ref. 25) adds to the blurry picture, out of which complex conclusions are drawn: senescence needed for full cell-based immune control,

Answer

We deleted this aspect from the result section and corrected the mistake in the discussion section, page 12, 1st paragraph.

Comment 9

senescence defects promoting immune escape and cancer stemness. Such far-reaching conclusions are simply not sufficiently substantiated with analyses of “remaining” (for which many different reasons may apply) tumor cells at a single point in time. It is very well conceivable that many findings presented here are due to the lack of the Rb/p53 pathway-controlling INK4a/ARF locus independent of any senescence/TH1 immune cell interaction.

Answer

a) The term ‘cancer stem cell’ was deleted, and the corresponding paragraph revised, page 6, 1st paragraph.

b) In a previous manuscript (Braumüller H et al. 2013, Nature 494: 361), we carefully analysed how IFN- γ /TNF-producing T_H1 cells induce p16^{Ink4a}/p19^{Arf}-dependent senescence in Tag-expressing cancer cells (the same RT2-cancer cells as here). Tag attenuates the action of p53 but does not abrogate the effect of p53 (Casanovas O et al. 2005, Oncogene 24, 6597-6604). We wrote this more clearly in the text, page 9, 1st paragraph.

Comment 10

On the contrary, this does not exclude that the authors made an important observation. In essence, I suggest to tone down a bit, to fundamentally revise structure and phrasing, condense overall content, clearly explain experimental settings (e.g. why were CD8+ T-cells depleted?), sharpen the message figure by figure and paragraph by paragraph (eliminate overinterpretation of often less clearly discriminating datasets),

Answer

As suggested, we now revised the structure, especially of the first result part, explained the CD8-depletion, and sharpened the message to eliminate over-interpretation.

Comment 11

include more subheadings, refrain from permanently jumping back and forth between figure panels, provide more integrative views on tumor growth, biology, molecular mechanisms and immune interaction over time, be clear on model systems (e.g. the repeatedly misleading presentation of actually INK4a/ARF-deleted tumors as “p16INK4a-/- tumors”), avoid misinterpretation of formally non-significantly different findings as biologically equivalent, and present the entire

central message as an additional, not the major principle underlying impaired outcome of INK4a/ARF-deficient tumors to ICB.

Answer

As addressed above, we restructured the manuscript as proposed, and subdivided especially the 1st and the 2nd result section with new paragraphs following exactly the experimental flow. In our manuscript, we clearly and repeatedly state that senescence is needed (as a second mechanism) to control those cancer cells that escape from either natural (*Fig. 1a*, upper panel) or ICB-induced cancer cell killing (*Fig. 1a*, lower panel). We sharpened our message and state this clearly in the abstract, page 2, in the introduction, page 3, as well as at the end of the first result part, page 6, 2nd and 3rd paragraph.

Remaining more specific concerns and comments

Comment 12

1. The title – as done so in the abstract – should be adapted to the re-defined statement that it's not seldomly p16INK4a but the deletion of both p16INK4a and p19ARF that disables cytokine-induced target cell senescence (CIS). Also, the genetic designation of p16INK4a-deficient RT2 tumors, e.g. in *Fig. 1* – explicitly distinguishing between p16INK4a^{-/-} and p16INK4a^{-/-};p19ARF^{-/-} cells – is misleading. According to the methods section, p16INK4a-deficient cells are “CDK2NA loss variants on chromosome 4” generated as survivors of ICB exposure (*Suppl. Fig. 3a* – hard to confirm for the reader regarding RT2 precursor line #1, but, if so, showing equally reduced signals for both chromosomes 4 and 5), thereby possibly selecting for gross deletions that include adjacent genes (e.g. MTAP, ANRIL, p15INK4b), and most likely disrupting both alleles of CDK2NA-exon 2, thereby cancelling both p16INK4a and p19ARF expression. Likewise, “CRISPR-p16INK4a” tumors are actually (see methods) p16INK4a/p19ARF-co-deleted tumors, but their label in *Fig. 1* and subsequent figures implies that only p16INK4a-deficient tumors were generated by the CRISPR approach. This is particularly confusing, because remaining “RT2 CRISPR p16INK4a^{-/-}” tumors, in contrast to “RT2 p16INK4a^{-/-} tumors”, present with and without ICB exposure positive for senescence markers p21CIP (significant), HP1gamma (weak) and H3K9me3 (strong), albeit no SA-beta-gal activity (*Fig. 2a-c*).

Answer

a) As suggested, we changed the title into:

“Cancer immune control requires senescence induction in tumour cells”

b) We reworded our abstract.

c) In the revised manuscript, we designate the natural loss mutants as RT2.*Cdkn2a*^{-/-} and the RT2-CRISPR-*p16^{Ink4a}^{-/-}p19^{Arf}^{-/-}* as RT2.CRISPR-*Cdkn2a*-cancer cells throughout the text.

d) In the ICB-treated RT2.CRISPR-*Cdkn2a*-cancer cells p21^{Cip1}, pHP1γ and SA-β-gal were not increased by ICB; in the ICB-treated RT2.CRISPR-*Cdkn2a*-cancer cells the staining is not stronger than the staining in the natural loss RT2.*Cdkn2a*^{-/-} mutants or the *Stat1*^{-/-} cancers – please compare to the ICB-treated RT2-cancers (*Fig. 2a-c*). For H3K9me3 see answer to comment 13.

Comment 13

Later in the text, the authors explain the H3K9me3 result by their assumption that CRISPR-Cas9 transfection may cause DNA damage – but provide no evidence for this claim in *Suppl. Fig. 5a*, where these cells remain negative for the DNA damage response marker gamma-H2AX. If CRISPR-driven ongoing DNA damage would actually occur, I would worry even more about the validity of this CRISPR model to study senescence as a read-out.

Answer

We do not claim that CRISPR-Cas9 transfection causes persistent DNA damage; we even show that the CRISPR-Cas9-transfected tumour cells are negative for γ H2AX and DNA-PK (*Supplementary Figure 3*), there is no difference between RT2.*Cdkn2a*^{-/-} and RT2.CRISPR-*Cdkn2a* cancers. In the text, we addressed this point now more precisely.

Others, like the group of Crabtree have shown that CRISPR-Cas9 transfection induces also non-specific H3K9me3. This CRISPR-Cas9-induced H3K9 methylation is independent of the gene transferred, persists longer and has been the topic of several manuscripts e.g. in Nature Communications (Braun SMG et al. (2017) Nat Commun 8, 560). We carefully changed our wording to make this point clear, page 6, 1st paragraph.

Comment 14

2. The legend of Suppl. Fig. 1a does not explain “RmCD8”, a CD8-depleting antibody pre-treatment according to the “methods” section. The cartoon suggests that tumors formed at a size of > 3mm within one week –

Answer

We now provided the reference for the anti-CD8 mAb, described the procedure more precisely in the legend of *Supplementary Figure 1* and in the materials and methods section. See also our answer to Comment 2. We improved the methodological details: the pre-treatment with the anti-CD8 mAb is now more precisely explained in the text (page 3, end and further page 4; page 5 paragraph 2), in the legend to *Fig. 1* and the legends of the *Supplementary Figure 1* (with reference 40 (Egeter O et al. 2000 *Cancer Res* 60, 1515-1520) leading to the source) and *Supplementary Figure 6*, and in the *Materials and Methods* section *Treatment of RT2-cancers in C3HeB/FeJ mice* - with reference 40 (Egeter O et al. 2000 *Cancer Res* 60, 1515-1520) leading to the source.

Comment 15

while the CD8-depleting antibody should be still around at quantitative concentrations. This is very puzzling, as no CD8+ T-cells would be available for immune checkpoint blockade (ICB) treatment at this time.

Answer

Tumours are frequently controlled by CD8 cytotoxic T cells. Yet, many manuscripts have shown that tumours, including RT2-cancers are controlled by T_H1 cells and activated macrophages in the absence of CD8 T cells (van den Broek ME et al. (1996) *J Exp Med* 184, 1781-1790, Muller-Hermelink N et al. (2008) *Cancer Cell* 13, 507-518, Kang T-W et al. (2011) *Nature* 479, 547-551.) We changed our wording and explained the mechanism and the rationale for our experiments more precisely, page 6, 3rd paragraph.

Comment 16

Moreover, since no functional, T-cell-based data on anti-Lag3/anti-PD-L1 ICB are shown, it is entirely unclear how this ICB cocktail exerts its function. Are the authors postulating a specific role of endogenous CD4-positive T-cells? What presumed immune target cells actually express PD1 and/or Lag3 at the time of ICB administration?

Answer

LAG-3 and PD-1 can be expressed by CD4 T cells. As outlined in the response to “Comment 4” we specified our wording and cite the corresponding review (Nguyen LT & Ohashi P (2015) *Nat Rev Immunol* 15, 45-56). They summarize the mechanistic role of LAG-3, PD-1/PD-L1 and the effects of blocking either one of the two molecules and of blocking both molecules, page 4, 1st paragraph.

Comment 17

3. What do the authors intend to explain in Suppl. Fig. 2, when showing PD-L1 IF on remaining tumor cells prior to and after presumably saturating anti-PD-L1 antibody exposure?

Answer

Mice were treated with a rat anti-PD-L1 mAb. For the immune fluorescence analyses, we first incubated the tissue sections with a different, also unconjugated rat anti-PD-L1 mAb, and for the staining of both mAb we then used a fluorescent anti-rat antibody. This method allowed us to detect the PD-L1 expression in tumours of untreated and of treated mice. We now exactly describe this procedure in the legend to *Supplementary Figure 4* and addressed this in the *Materials and Methods* section, page 20, *Immunofluorescence staining*.

Comment 18

Moreover, according to the main text, Suppl. Fig. 2b, d supposedly shows an analysis of T antigen (“Tag”) expression, which is not the case.

Answer

In the new *Supplementary Figure 4b, d*, we show PD-L1 or β 2microglobulin on RT2-cancer cells. We corrected this error in the main text, page 4, 1st paragraph.

Comment 19

4. With no error bars shown, cells in Fig. 3c behave in growth curve analyses very similar in response to IFN-gamma/TNF-alpha – irrespective of p16INK4a (i.e. INK4a.#/ARF) deletion by CRISPR-Cas or not.

Such data is of key importance to support the hypothesis of the authors, the differences observed are marginal, but the wording in the text is black and white (“...control cells susceptible to CIS... ..p16/p19-deficient cells resistant to CIS”).

While the referee did not overlook clear differences in SA-beta-gal responsiveness, the situation is odd and requires clarification, i.e. more robust biological evidence (additional senescence markers, time-course analyses etc.).

Answer

We tuned down our wording as suggested, page 5, 2nd paragraph.

Growth curves were given as single experiments; the experiments were repeated fourtimes. We mentioned this technical point in the legend of *Fig. 3*. Additionally, we now provided for the manuscript a clearer *Fig. 3d*, where the RT2.CRISPR-*Cdkn2a* shows exponential growth and a clear distinction between the RT2.CRISPR-*Cdkn2a* and the RT2.CRISPR controls. For the referee, we show a graph summarizing four independent experiments with the same cell line + SD.

Figure for reviewer only 1 | Growth assays were performed with RT2.CRISPR-*Ctr*-control (left panel) or RT2.CRISPR-*Cdkn2a* RT2-cancer cells (right panel). Cells were cultured either with medium (Ctr) or with medium containing 100 ng/ml IFN- γ and 10 ng/ml TNF for 96 h, washed and then cultured with medium for another 3-5 days, data show the mean with SD, n = 4.

Comment 20

5. Irrespective of the drug-induced cell death data in Fig. 3a-c, it cannot be ignored that p19ARF-depleted cells harbor a p53 pathway defect, which is likely to contribute to better engraftment or reduced ICB responsiveness – but not necessarily due to a senescence defect. Subtle differences in p53 activity may translate into a slightly reduced apoptotic propensity, which is hard to see in short-term drug-induced cytotoxicity assays, but becomes very relevant in tumor outgrowth experiments regarding number and nature of remaining cells.

Answer

At this point we cannot follow the referee:

a) The *p19^{Arf}*-depleted (*Cdkn2a*-deficient) RT2-cancer cells did not engraft better than the RT2-cancer cells that expressed *p19^{Arf}* (Fig. 1a, 1st versus 3rd experimental group).

b) We directly show that the *p19^{Arf}*-depleted (*Cdkn2a*-deficient) cancer cells failed to undergo senescence (Fig. 3b).

We never claim that apoptosis is irrelevant. What we claim and what we show is that those cancer cells that were not cleared, neither by apoptosis nor by cytolysis, require the senescence inducing *Cdkn2a* and p21^{Cip1} to be controlled by the immune system.

Comment 21

6. If the tumor cells continue to express Large T antigen, as stated in the legend of Suppl. Fig. 4 and shown in a single case in Fig. 5, how is senescence induction possible, given the inhibition of both the Rb and the p53 pathway by Large T? The authors themselves admit that “RT2 cancers are inappropriate to carefully investigate the role of p21CIP1 induced senescence in response to ICB” – but, actually, the statement could extend to “...inappropriate to carefully investigate the role of senescence...” ...at all.

Answer

In our previous manuscript (Braumüller H et al. (2013) Nature 494: 361), we carefully addressed this point. We showed that the combined action of IFN- γ and TNF induces senescence in RT2-cancers in a *Cdkn2a*-dependent fashion. We also showed that treatment of RT2 mice with Tag-

T_H1 cells or of RT2 cells with IFN- γ does not suppress the expression of Tag (Supplementary Figure 5 in Braumüller et al., 2013).

Importantly, Tag impairs the p53-Rb axis, but it does not abrogate this axis (Casanovas O et al. 2005, *Oncogene* 24, 6597-6604). In line with this, we found p21^{Cip1} in RT2 tumours of sham-treated or ICB-treated tumours (Fig. 2a). Thus, Tag does not eliminate the action of p53. Yet, because of this influence of Tag on p53, RT2-cancers are inappropriate to study the role of p21^{Cip1} in response to ICB. This would mix two different influences on p21^{Cip1} that cannot be separated one from each other. We therefore used λ -MYC mice to study the role of p21^{Cip1}. We explained this at the beginning of page 9, 1st paragraph.

Comment 22

7. There is no information how Large T antigen-specific TH1 cells were generated. Again, it remains unclear to the readers why CD8+ T-cells were eliminated from the experimental setting.
Answer

We now exactly describe the generation of Tag-T_H1 cells in the Material and Methods section “Treatment of RT2 or RT2.*Stat1*^{-/-} mice.”, page 19.

The reason for the CD8-depletion is now precisely explained in the main text, page 4, 1st paragraph.

Comment 23

In turn, the actual mode of tumor cell elimination is speculative (“most likely through the induction of cytotoxic macrophages”), although deciphering the underlying immunological mechanism appears to be central for the claimed link between senescence and immune clearance.

Answer

The mode of tumour cell elimination in the absence of CD8 T cells by T_H1 cell-activated type I macrophages is well established and was the topic of several manuscripts, including early studies by the Zinkernagel group (van den Broek ME et al. (1996) *J Exp Med* 184, 1781-1790) or the Zender group (Kang T-W et al. (2011) *Nature* 479, 547-551). We rewrote the entire paragraph to make clear:

a) that cancer cell clearance by T_H1 cells and type I macrophages is an established mode of cancer cell clearance and

b) that we performed immune fluorescence to analyse whether the cell populations required for the T_H1 cell-mediated cancer clearance are also found in the tumours regressing during ICB.

Page 6, 3rd paragraph.

Comment 24

8. What’s the long-term fate of residual, senescent RT2 tumor cells after ICB plus adoptive anti-Large T antigen TH1 transfer (AT)? How stable is this type of senescence (with apparently strong p16INK4a induction, now virtually absent in ICB-only treated RT2 tumors... [Fig. 4d]) in vivo, what’s the fraction and timing of re-progressing tumors, do the cells lose the large T antigen, and would re-treatment with ICB or AT or both again result in tumor control?

Answer

a) The long-term fate of senescent RT2-cancer cells was analysed in a previous manuscript (Braumüller H et al. (2013) *Nature* 494: 361 – Fig. 4). There, we performed exactly the experiment suggested by the reviewer: senescent β -cancer cells were isolated from senescent RT2-cancers and first cultured *in vitro* (as here in Fig. 7) and then transplanted into NSG mice. For a total of 3 months, the cells did not restart growing, neither *in vitro* nor *in vivo*.

b) In mice treated with ICB and AT, the Tag expression is as strong as in the sham-treated controls (we show mRNA for four mice and immune histology from one mouse; Fig. 5). We address this on page 7, 1st paragraph.

c) We earlier showed that about half of the RT2 mice remain healthy for at least half a year, instead of dying after about 3 months (Müller-Hermelink N et al. (2008) *Cancer Cell* 13, 507-518), if Tag- T_H1 transfer (AT) treatment is started early in life. When the treatment is started later, life extension and the cure rate decline.

Comment 25

9. The lambda-myc lymphoma experiment is interesting, since it is not based on expression of the problematic, senescence-compromising Large T antigen. Here, it is unclear to the reader whether CD8⁺ T-cells were pre-eliminated in this model prior to a different ICB combo (now anti-PD1 plus anti-CTLA4), too. While – somewhat surprisingly – lymphomas lacking p21^{CIP1} present with a senescence defect, the results are confusing. Essentially, the differences observed in p21^{CIP1}-proficient lymphomas for untreated vs. ICB (Fig. 8a) can be superimposed (including virtually identical median survival times) with the results shown for p21^{CIP1}-deficient lymphomas in Fig. 8e, but the latter being statistically “not significant” – which, if true, is clearly due to lower sample numbers. To argue that “non-significant” means no biological difference, and concluding that this provides proof for the requirement of an intact senescence response for full ICB activity is simply not correct.

Answer

a) We did not delete CD8 T cells in λ -MYC mice, as the tumours grow in normal immune competent mice. We deleted CD8 T cells only to allow the engraftment of the transplanted RT2-cancer cells. This is now clearly described in the main text on page 4, 1st paragraph; page 5 paragraph 2, in the legend to *Fig. 1* and the legends of the *Supplementary Figure 1* and *Supplementary Figure 6*, and in the *Materials and Methods* section *Treatment of RT2-cancers in C3HeB/FeJ mice*.

b) Concerning the wording ‘statistically not significant’ – we now just give the p-value. We introduced p-values as referee #4 asked for this statistical comparison. We now describe precisely in the text that 20% to 30% of the p21^{Cip1}-proficient mice did not develop lymphomas and remained healthy when treated with ICB, while ICB did not rescue one single p21^{Cip1}-deficient mouse. All ICB-treated p21^{Cip1}-deficient mice died from lymphomas. We address this on page 9, 1st and on page 10, 1st paragraph.

Referee #3

The authors have successfully addressed all my criticisms. This work will be of great interest to scientists interested in senescence and onco-immunology. More broadly, it will impact on the biological understanding of a variety of tumor types in response to innovative immunotherapies as well as inform on new therapeutic regimens that combine immunotherapy with pro-senescence treatments. I recommend publication in *Nature Communications*.

Minor notes:

Comment 1

Figure 1C: as the statistics refers to the effect of treatment in triplicates from one mouse I recommend to remove p-values and leave only a descriptive stat. This in general should apply to all experiments with n=3.

Answer

The statistics were introduced in response to referee #2; therefore we think it would be good to leave them.

In the case of Fig. 1c, we analysed tumours of three individual mice and each data point represents the total of three tumour slides measurements. We changed the figure legend to point this out.

In the case of the annexin V staining of Fig. 3, we analysed three to four independent *in vitro* treatments of each individual cancer cell line and measured them via FACS. The p-value was calculated and shown, as referee #2 demanded these data analyses; the analysis of the p-value in Fig. 3 is also statistically correct, even if the number of samples is limited.

Comment 2

Figure 7: the labels for the 3 panels should read ICB/AT instead of ICB.

Answer

Done as suggested by the reviewer. We changed the labelling in the 3 graphs to ICB/AT in Fig. 7.

Comment 3

Figure Suppl. 12: misspelled “Amplifikation” in the legend

Answer

Done as suggested by the reviewer. We changed “Amplifikation” to “Amplification” in the legend of the Figure, which is now *Supplementary Figure 13*.

Comment 4

Please check the first two sentences in the Discussion.

Answer

Done as suggested by the reviewer. We thank the reviewer for this remark and changed the start of the Discussion, page 11 to 12.

Comment 5

In general, there are quite a few misspelled words/typos in the manuscript that should be corrected by the authors or at the proof stage.

Answer

Done as suggested by the reviewer. We carefully checked the text of the revised manuscript and corrected all misspelled words/typos we found.

REVIEWERS' COMMENTS:

Reviewer #2 (Remarks to the Author):

Brenner-E,... ..Röcken-M, Cancer immune control needs intratumoral senescence-inducing p21CIP1 and p16INK4a pathways

Re-re-submission to Nature Communications

This is now the second revision of a manuscript by Röcken and colleagues that links cell-based anti-cancer immunity – beyond its well-established cytotoxic mode of action – to induction of cellular senescence. The central claim is that defective target cell senescence (e.g. by deletion of the senescence mediators p16INK4a/p19ARF or p21CIP1) impairs natural or immune checkpoint blockade (ICB)-exerted tumor control via an insufficient long-term proliferative arrest in the ICB-surviving residual tumor cell population.

The authors have responded to the long list of remaining concerns, and fixed most of them. While many of the additional explanations enhance clarity, other points remain scientifically problematic. For instance, the criticized interpretation of Fig. 8e (lambda-Myc lymphomas lacking p21CIP1) as being meaningfully different from Fig. 8a is now countered by the authors in the re-revised manuscript by saying “While ICB protected 20-30% of lambda-Myc mice from lymphoma and death (Fig. 8a), all ICB-treated lambda-Myc/p21CIP1^{-/-} mice died from lymphomas between day 100 and 210 (Fig. 8e)”. The stated “20-30%” is wrong; first, there is no room for a vague range, second, it’s simply just an approx. 18% of the mice being long-term survivors (not “up to 30%”), and this low plateau is probably maintained by just 1 or 2 mice, hence, not anyhow different from the results presented in Fig. 8e with much less mice investigated there. As a consequence, the authors write a few lines later “This proves that the senescence-inducing cell-cycle regulator p21CIP1 was strictly required to prevent the transition of pre-malignant B cells into B-cell lymphomas in ICB-treated lambda-Myc mice”. This conclusion is scientifically NOT correct.

Moreover, the authors responded to the critique of their imprecise labeling and presentation of lymphomas lacking not only p16INK4a but also the p53 upstream regulator p19ARF by rephrasing the title to “Cancer immune control requires cancer-intrinsic senescence induction” – making the title much more general (no molecular focus) and bolder (“requires!”), for which I cannot see any justification. A more accurate title that also acknowledges the (limited) biological weight of the data (as just exemplified again in the previous paragraph) might rather be “Cellular senescence contributes to lasting tumor control by immune checkpoint blockade”.

We revised our manuscript to answer the remaining REVIEWERS' COMMENTS and provided a point-by-point response with our answers in blue. The changes in the main manuscript can be tracked in the Microsoft Word file.

REVIEWERS' COMMENTS:

Reviewer #2 (Remarks to the Author):

Brenner-E,... ..Röcken-M, Cancer immune control needs intratumoral senescence-inducing p21CIP1 and p16INK4a pathways

Re-re-submission to Nature Communications

This is now the second revision of a manuscript by Röcken and colleagues that links cell-based anti-cancer immunity – beyond its well-established cytotoxic mode of action – to induction of cellular senescence. The central claim is that defective target cell senescence (e.g. by deletion of the senescence mediators p16INK4a/p19ARF or p21CIP1) impairs natural or immune checkpoint blockade (ICB)-exerted tumor control via an insufficient long-term proliferative arrest in the ICB-surviving residual tumor cell population.

Answer

We thank the reviewer #2 for having now accepted the explanations on all questions of the referees in the two previous revisions. Reviewer #2 remains with two additional questions, we hope to answer them in a satisfactory manner.

The authors have responded to the long list of remaining concerns, and fixed most of them. While many of the additional explanations enhance clarity, other points remain scientifically problematic. For instance, the criticized interpretation of Fig. 8e (lambda-Myc lymphomas lacking p21CIP1) as being meaningfully different from Fig. 8a is now countered by the authors in the re-revised manuscript by saying “While ICB protected 20-30% of lambda-Myc mice from lymphoma and death (Fig. 8a), all ICB-treated lambda-Myc/p21CIP1^{-/-} mice died from lymphomas between day 100 and 210 (Fig. 8e)”. The stated “20-30%” is wrong; first, there is no room for a vague range, second, it’s simply just an approx. 18% of the mice being long-term survivors (not “up to 30%”), and this low plateau is probably maintained by just 1 or 2 mice, hence, not anyhow different from the results presented in Fig. 8e with much less mice investigated there. As a consequence, the authors write a few lines later “This proves that the senescence-inducing cell-cycle regulator p21CIP1 was strictly required to prevent the transition of pre-malignant B cells into B-cell lymphomas in ICB-treated lambda-Myc mice”. This conclusion is scientifically NOT correct.

Answer to question 1

Here reviewer #2 criticises two points that were addressed by a improved explanation in the text:

- Our statement that ‘... ICB protected 20 %-30 % of λ -MYC mice from lymphomas’, by referring to Fig. 8a.

This window results from the experiments in Fig. 8a and those in Fig. 8e. To make this point clear, we changed our phrasing in the lines 250 – 252 from:

Surprisingly, combined ICB with anti-CTLA-4 and anti-PD-1 mAb protected 20%-30% of λ -MYC mice from lymphomas, combined ICB treated mice were still healthy at > 250 days (Fig. 8a), a lifetime that has never been achieved by any other therapy in this mouse model.

to

Combined ICB with anti-CTLA-4 and anti-PD-1 mAb protected 18% (Fig. 8a) to 30% (Fig. 8d) of λ -MYC mice from lymphomas for at least 200 days. In a long-term experiment, 18% of the λ -MYC mice with a combined ICB therapy were still healthy at > 250 days (Fig. 8a), a lifetime that has never been achieved by any other therapy in this mouse model.

and in the line 269 from:

While ICB protected 20%-30% of λ -MYC mice from lymphoma and death (Fig. 8a)
to

While ICB protected 18%-30% of λ -MYC mice from lymphoma and death (Fig. 8a, d)

- Our statement that the effect of ICB-treatment depends on p21^{Cip1}:

We describe that:

In λ -MYC.p21^{Cip1}^{-/-} mice, ICB failed to protect mice from death as all mice died from lymphomas between day 100 and 210 (Fig. 8e).

Explanation:

It is not possible to compare the percentages of surviving mice between Fig. 8a and Fig. 8e as the mice have a slightly different background, due to the crossing of C57BL/6 λ -MYC mice with C57BL/6 p21^{Cip1}^{-/-} mice. The comparison can only be done within exactly the same strain. There was no significant difference between treated and untreated λ -MYC.p21^{Cip1}^{-/-} mice (Fig. 8e), while this difference was significant in C57BL/6 λ -MYC mice. Moreover, all λ -MYC.p21^{Cip1}^{-/-} mice die, whether treated or not. In consequence, we make the conclusion that cycle regulator p21^{Cip1} was strictly required to prevent the transition of pre-malignant B cells into B-cell lymphomas in ICB-treated λ -MYC mice.

Moreover, the authors responded to the critique of their imprecise labeling and presentation of lymphomas lacking not only p16^{Ink4a} but also the p53 upstream regulator p19^{Arf} by rephrasing the title to “Cancer immune control requires cancer-intrinsic senescence induction” – making the title much more general (no molecular focus) and bolder (“requires”!), for which I cannot see any justification. A more accurate title that also acknowledges the (limited) biological weight of the data (as just exemplified again in the previous paragraph) might rather be “Cellular senescence contributes to lasting tumor control by immune checkpoint blockade”.

Answer to question 2

Reviewer #2 asked us in the 2nd review to include in the title not only p16^{Ink4a} but also the p53 upstream regulator p19^{Arf}. As we did not directly analyse p19^{Arf}, it would be incorrect to include p19^{Arf} in the title. To keep the old title as far as possible and to keep its focus on cell cycle regulators (as asked by reviewer #2 in this 3rd review) we returned again from ‘requires’ to ‘needs’ (what was two times accepted by all 4 referees) and used instead of the terms p16^{Ink4a} and p21^{Cip1} the more general term ‘cell cycle regulators’. This is also justified, as we found in the melanomas severe aberrations in various interferon/STAT1-dependent genes that regulate *CDKN1A*, *CDKN2A* or their effector function.

To address both concerns of reviewer #2 and to be as close as possible to the original title accepted by reviewer #1, #3 and #4, new title therefore is:

Cancer immune control needs senescence induction by interferon-dependent cell cycle regulator pathways in tumours